



# Atmospheric composition in the European Arctic and 30 years of the Zeppelin Observatory, Ny-Ålesund

Stephen M. Platt[1], Øystein Hov[2], Torunn Berg[3], Knut Breivik[1], Sabine Eckhardt[1], Konstantinos Eleftheriadis[4], Nikolaos Evangeliou[1], Markus Fiebig[1], Rebecca Fisher[5], Georg Hansen[1], Hans-Christen Hansson[6], Jost Heintzenberg[7], Ove Hermansen[1], Dominic Heslin-Rees[6], Kim Holmén[8], Stephen Hudson[8], Roland Kallenborn[1], Radovan Krejci[6], Terje Krognes[1], Steinar Larssen[1], David Lowry[5], Cathrine Lund Myhre[1], Chris Lunder[1], Euan Nisbet[5], Pernilla B. Nizetto[1], Ki-Tae Park[9], Christina A. Pedersen[8], Katrine Aspmo Pfaffhuber[1], Thomas Röckmann[10], Norbert Schmidbauer[1], Sverre Solberg[1], Andreas Stohl[1,11], Johan Ström[6], Tove Svendby[1], Peter Tunved[6], Kjersti Tørnkvist[1], Carina van der Veen[10], Stergios Vratolis[4], Young Jun Yoon[9], Karl Espen Yttri[1], Paul Zieger[6], Wenche Aas[1], Kjetil Tørseth[1]

[1]NILU-Norwegian Institute for Air Research, PO Box 100, 2027 Kjeller, Norway
[2]Norwegian Meteorological Institute, Henrik Mohns Plass 1, 0371 Oslo, Norway
[3]NTNU-Norwegian University of Science and Technology, Department of Chemistry, Trondheim, Norway
[4]NCSR "Demokritos"-Institute of Nuclear and Radiological Sciences and Technology, Energy and Safety Environmental Radioactivity Laboratory, 15310 Athens, Greece
[5]Department of Earth Sciences, Royal Holloway, University of London, Egham, UK
[6]Department of Environmental Science, Stockholm University, 106 91 Stockholm, Sweden
[7]Leibniz-Institute for Tropospheric Research, Permoserstrasse. 15, 04318 Leipzig, Germany
[8]NPI-Norwegain Polar Institute, Fram Centre, PO Box 6606 Langnes, 9296 Tromsø, Norway
[9]KOPRI-Korea Polar Research Institute, 26, Songdo Mirae-ro, Yeonsu-Gu, Incheon, Republic of Korea
[10]IMAU-Institute for Marine and Atmospheric Research Utrecht, Utrecht University, The Netherlands
[11]Now at Department of Meteorology and Geophysics, University of Vienna, Althanstrasse 14, 1090 Vienna, Austria

*Correspondence to:* Stephen M. Platt (sp@nilu.no), Kjetil Tørseth (kt@nilu.no)

**Abstract.** The Zeppelin Observatory (78.90 °N, 11.88 °E) is located on the Zeppelin Mountain at 472 m above sea level on Spitsbergen, the largest island of the Svalbard archipelago. Established in 1989, the observatory is part of the 'Ny-Ålesund Research Station' and an important atmospheric measurement site, one of only a few in the high Arctic and as a part of several European and global monitoring programs and research infrastructures, notably the European Monitoring and Evaluation Programme (EMEP), the Arctic Monitoring and Assessment Programme (AMAP), the Global Atmosphere Watch (GAW), the Aerosols, Clouds, and Trace gases Research InfraStructure (ACTRIS), the Advanced Global Atmospheric Gases Experiment (AGAGE) network, and the Integrated Carbon Observation System (ICOS). The observatory is jointly operated by the Norwegian Polar Institute (NPI), Stockholm University and the Norwegian Institute for Air Research (NILU). Here we detail the establishment of the Zeppelin Observatory including historical measurements of atmospheric composition in the European Arctic leading to its construction. We present a history of the measurements at the observatory and review the current state of the European Arctic atmosphere, including results from trends in greenhouse gases, chlorofluorocarbons (CFCs) and hydrochlorofluorocarbons (HCFCs), other traces gases, persistent organic pollutants (POPs) and heavy metals, aerosols and Arctic haze, and atmospheric transport phenomena.



# 1 Introduction

Following early advances in aerosol measurement technology and data, Junge (1972) coined the concept of a 'global background aerosol', recommending its study at background stations as far away from anthropogenic sources as possible. The
Polar regions were prime areas for the establishment of such sites. Furthermore, a possible feedback mechanism, where decreased ice cover would decrease Earth's albedo, leading to more warming, particularly in the Arctic, an 'Arctic albedo effect', had already been described in the literature (Budyko, 1969;Schneider and Dickinson, 1974), while Hov and Holtet (1987) noted that "Theoretical calculations indicate that the growth in temperature around Svalbard could be 3 to 4 times the global average temperature increase". A third motivation for atmospheric background measurements in the Arctic followed
the 1973 oil crisis, which led to increased oil exploration in the region. Norwegian environmental research was commissioned to establish the status of the pristine Arctic environment before the advent of large-scale commercial exploitation (Joranger and Ottar, 1984).

In Norway, a growing interest in 'Arctic haze' (an observed seasonal variability of Arctic aerosol, with maximum levels around spring) led to the 'Workshop on Arctic Aerosols' (27 to 28[th] April 1977, at NILU) sponsored jointly by the U.S. Office of
Naval Research, co-chaired by NILU's director Brynjulf Ottar, and Kenneth A. Rahn of the University of Rhode Island (Ottar and Rahn, 1980). Out of this meeting grew a co-operation to establish a pan-Arctic observation programme to determine the sources, transport mechanisms, and effects of aerosols in the Arctic. Meanwhile in Sweden in the same year, the Swedish parliament accepted the proposal for a Swedish monitoring programme ('program för övervakning av miljökvalitet', PMK) one part of which was to be long-term monitoring of changes in atmospheric composition.

Following the 1977 workshop on Arctic aerosol there were three further symposia on Arctic atmospheric chemistry, on 6 to 8[th] May, 1980 (Rahn, 1981a), 7 to 9[th] May, 1984 (Rahn, 1985), and 29[th] September to 2[nd] October, 1987 (Rahn, 1989b). These Arctic air chemistry symposia provided an international framework for Arctic haze research based on long-term ground-based observations, or at least field campaigns with extended measurement programmes, and aircraft measurements. A substantial five-year Arctic measurement and research programme led by NILU also started in 1981, financed by British Petroleum Ltd.
(BP), as part of the Norwegian government's policy to allow the search for oil and gas at northerly latitudes (Ottar, 1989). The Arctic air chemistry symposia and the BP-programme at NILU also provided the international scientific support and legitimacy for Norwegian government funding to establish a global background observatory.

From the start, establishing a joint Norwegian/Swedish baseline monitoring observatory within the WMO framework was under discussion, and measurement campaigns were carried out in the Arctic to determine the ideal location for such an
observatory. Heintzenberg, (1983) and Heintzenberg et al., (1985) emphasised that regional transport of anthropogenic trace substances from populated regions in the Soviet Union and Europe should be monitored, even if much of the high Arctic lay within Soviet Union territory where atmospheric monitoring activities were impossible for most scientists. Year-round measurements were particularly required to characterise Arctic haze (e.g., 'background' vs 'baseline' levels). Specifically, Heintzenberg et al. (1985) recommended the establishment of a permanent station with instrumentation to measure particle





number concentration, light-scattering, and greenhouse gases, particularly $CO_2$. Accordingly, several research groups promoting the development of high latitude background stations devised a plan to link up to the baseline monitoring stations at the South Pole, run by the National Oceanic and Atmospheric Administration (NOAA); Cape Grim on the western cape of Tasmania, run by Commonwealth Scientific and Industrial Research Organisation (CSIRO), Australia; American Samoa, Mauna Loa, and Point Barrow, Alaska (all NOAA sites); and Alert, Canada, run by Environment Canada. The World Meteorological Organization (WMO) was to have a strong coordinating role. This, in addition to the Arctic haze symposia, Norwegian Environment agency funding, and Swedish funding via the PMK, led to the establishment of an observatory on the Zeppelin Mountain close to Ny-Ålesund (Rahn, 1989a).

Here we present historical atmospheric composition measurements in the Arctic, including the measurements used to identify Mt Zeppelin as the ideal location for an atmospheric observatory. We detail the construction of the Zeppelin Observatory, its characteristics with respect to atmospheric transport, and subsequent expansions of measurement activities. We discuss trends in aerosol physical-chemical properties, greenhouse gases, reactive trace gases, atmospheric oxidants, persistent organic pollutants (POPs) reactive trace gases, atmospheric oxidants, persistent organic pollutants, and heavy metals including mercury.

## 2 A history of atmospheric composition measurements in the European Arctic

### 2.1 The rediscovery of Arctic haze

As discussed by Garrett and Verzella (2008), the presence of visibility-reducing haze in the Arctic was noted by early explorers in the late nineteenth century and discussed by Nordenskiöld (1883). Schnell (1984b) also suggest early evidence of observations of Arctic haze during Macmillan's search for the (non-existent) Crocker Land in the Canadian Arctic in 1913. However, according to MacMillan and Ekblaw (1918), MacMillan finally accepted that an apparent land mass he believed to be Crocker Land and was attempting to reach, was a indeed a mirage or fata morgana, described by their local Inuit guide as 'mist', only after five arduous days, stating "The day was exceptionally clear, not a cloud or trace of mist. [….] had we not been out on the frozen sea for 150 miles, we would have staked our lives upon its reality. Our judgment then, as now, is that this was a mirage […]".  Arctic haze was not definitively observed during the expedition and 'mist' in this case was a term used to refer to a mirage. Nevertheless, the account does provide evidence of a term for Arctic haze in the local Inuit vocabulary of the time.

According to Schnell (1984a), Mitchell (1957) was the first to document haze over the Arctic ice cap in the contemporary scientific literature. Haze was observed by pilots of the 'Ptarmigan' weather reconnaissance flights in the 1950s from Alaska to the North Pole, which Mitchell (1957) suggested was composed of non-ice particles $< 2$ µm in diameter. Raatz (1984) reanalysed the Ptarmigan flights from 1948 to 1961, finding a maximum in the number of low visibility observations in spring.



It was further suggested that the haze had an anthropogenic origin (Holmgren et al., 1974;Radke et al., 1976), subsequently identified as Eurasian.

Measurements of Arctic haze and Arctic aerosols in the European Arctic began in the 1970s when, following a mining accident in Ny-Ålesund in 1962 and the political turmoil unleashed in Norway, the 'King's Bay Affair', Hanoa (1989), a new use was sought for Norwegian infrastructure in the settlement. The European Space Research Organization (ESRO) established a
satellite ground station in Ny-Ålesund in 1967, while Norwegian Polar Institute (NPI) started year-round activities with over-wintering staff from 1968. The establishment of environmental research activities at the former mining settlement enabled researchers at NILU to begin studying the transport of air pollutants into the European Arctic with measurements of total suspended particulate matter (TSP) at Ny-Ålesund in 1973. A high-volume sampler (500 m$^3$ of air per day) was installed on the Roald Amundsen airship mooring mast close to the Kongsfjorden shoreline (Fig. 1) in collaboration with NPI. Samples
were taken weekly on filters and analysed at NILU for TSP, as well as for elements including mercury, chromium, and zinc. Similar regional-type stations operated in Tange (Jutland, Denmark), Tveiten (South Norway), Rena (central Norway) and Skoganvarre (North Norway). Results showed the episodic transport of air pollutants into the Arctic (Rahn, 1981b and references therein.) Due to the historical and cultural significance of the Amundsen mast, operations were stopped in 1977 by a preservation order.



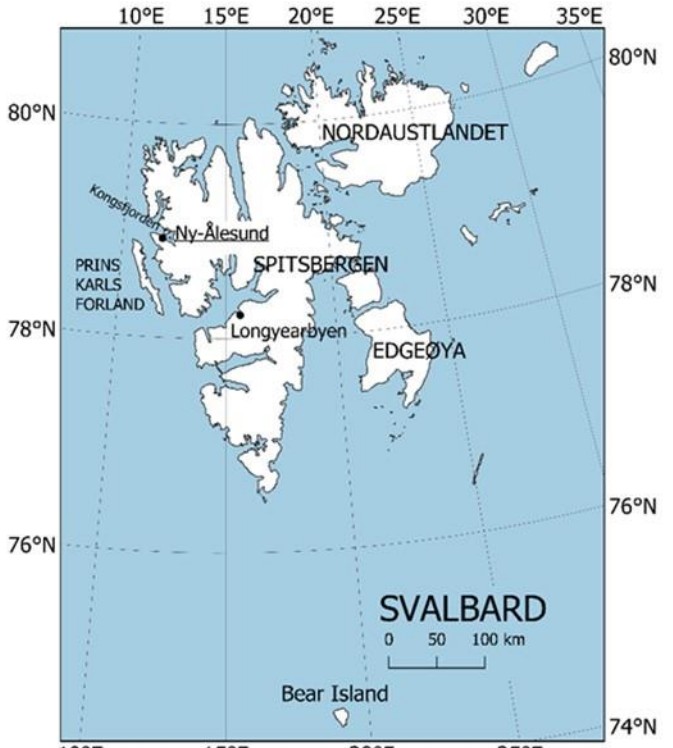

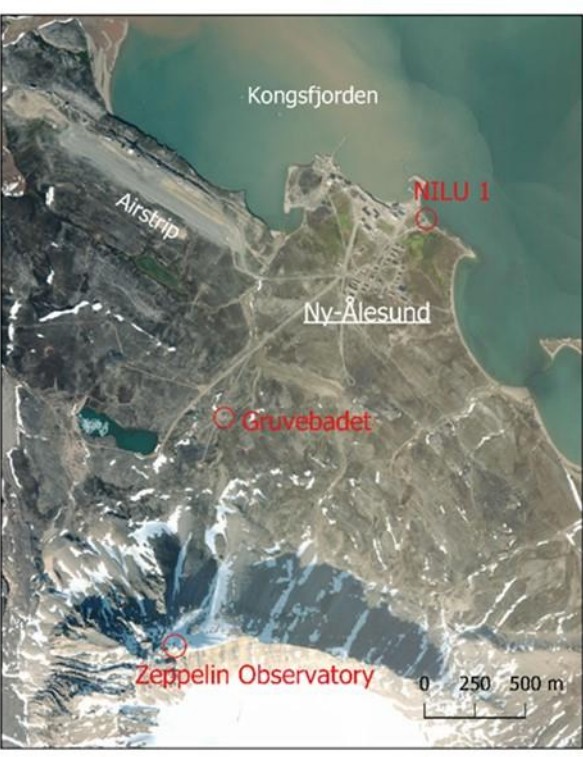

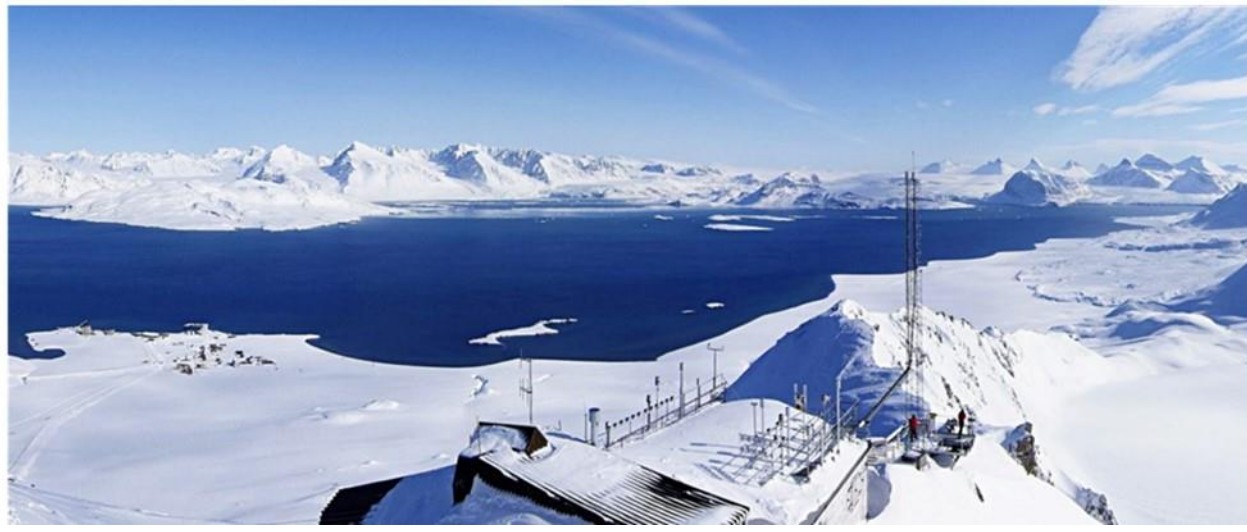

**Figure 1: Top left: map of the Svalbard Archipelago. Top right: Satellite image of Ny-Ålesund with the measurement sites at NILU-1, Gruvebadet, and the Zeppelin Observatory in red. Bottom: view of the Zeppelin Observatory looking down over the Ny-Ålesund settlement and Kongsfjorden. Top left, map data: Svalbard Kartdata, scale: 1:1 000 000, credit to: Norwegian Polar Institute. https://doi.org/10.21334/npolar.2014.63730e2e, accessed 18.06.2021. Top right, map data: ESRI Satellite, scale: variable resolution, credit to: Esri, Maxar, Earthstar Geographics, USDA FSA, USGS, Aerogrid, IGN, IGP, and the GIS User Community, https://server.arcgisonline.com/ArcGIS/rest/services/World_Imagery/MapServer, accessed 18.06.2021. Top panels produced using**





**QGIS Geographic Information System. Open Source Geospatial Foundation Project, http://qgis.osgeo.org, accessed 18.06.2021. Bottom, photo credit: Ove Hermansen, NILU.**

Aerosol composition measurements by NILU began again on July 5[th] 1977, with sulfur pollutant measurements at Bear Island and Ny-Ålesund (Fig. 2, Joranger and Ottar, 1984;Larssen and Hanssen, 1980). In Ny-Ålesund a new measurement site (NILU-1) was constructed close to the settlement shoreline with filter samples analysed for sulfate, nitrate, ammonium, chloride, magnesium, calcium, and sodium in addition to lead, cadmium and zinc until 30[th] June 1980. Additional aerosol measurements (composition, aerosol size distribution) at the site were performed by Stockholm University in 1979 and 1981 (Heintzenberg,

1980;Heintzenberg et al., 1981).

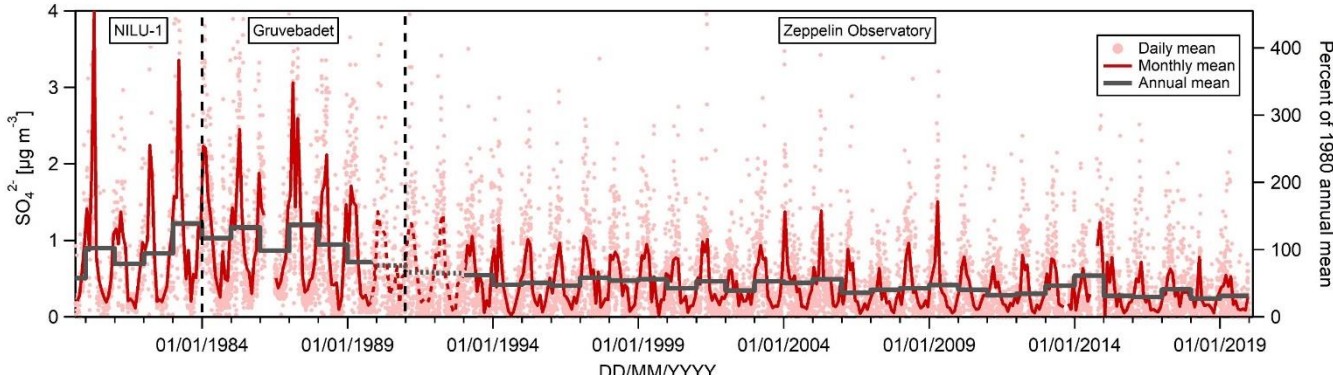

**Figure 2: Sulfate ($SO_4^{2-}$) concentrations at Ny-Ålesund (NILU-1), Gruvebadet, and the Zeppelin Observatory. Sulfate is sea-salt corrected except for 1990-1992 where total sulfate was measured, as also indicated by the dashed line. For reference, the right axis shows levels relative to the 1980 annual mean of 0.92 µg m$^{-3}$. Daily and annual means calculated only where data coverage is ≥75%**
**of total day or year, respectively.**

**2.2 Early greenhouse gas measurements in the European Arctic**

The first global background $CO_2$ monitoring programme was organised in 1956 as part of the 'International Geophysical Year' (Fritz, 1959). The remote Arctic, far from large local combustion sources, represents an ideal location for such background measurements.  Accordingly, $CO_2$ measurements at Barrow, Alaska were initiated in 1961, with Kelley (1974) finding mixing

ratios ≈2.5 ppm higher than from contemporaneous measurements at Mauna Loa, Hawaii, and with a larger annual variability. An early observed mean mixing ratio of 398 ppm at Hornsund, Svalbard in October to December 1957 is likely erroneous, due to analytical artefacts or contamination from heating oil combustion at Hornsund (Jaworowski, 1989).

Greenhouse gas (GHG) measurements around Svalbard began as part of the PMK-funded quest for a suitable location for a background station in the European Arctic. The two leading candidates for the background site were Bear Island, the site of

weather observations including radio soundings, located in the Barents Sea about halfway between the northern Scandinavian coast and Svalbard (74.52 °N, 19.02 °E), and the former mining settlement of Ny-Ålesund (78.93 °N, 11.92 °E), now converted into a site for research and monitoring following the King's Bay Affair. Heintzenberg (1983) took daily $CO_2$ grab samples at both locations in August 1981 to 1982. In addition, an intensive summer campaign at the NILU-1 site in Ny-Ålesund, August to September 1982, included 6-hourly nondispersive infrared (NDIR) $CO_2$ measurements. Results from both sites were similar



to previous observations at Barrow and Alert, Canada, with levels showing an annual variation of ≈15 ppm and short term variations ≈4 ppm (Peterson et al., 1982;Wong and Pettit, 1981). Regular measurements of $CO_2$ (using infra-red absorption spectroscopy measurement techniques) did not begin at Svalbard until the construction of the Zeppelin Observatory in 1989. Measurements of Arctic $CH_4$ began later than those of $CO_2$. The Arctic $CH_4$ mixing ratio was ≈1600 ppb in August-September 1967 at Point Barrow, Alaska (Cavanagh et al., 1969), while aircraft measurements when descending into Point Barrow showed

a mixing ratio of 1721 ppb in April 1986 (Conway and Steele, 1989). Trivett et al. (1989) observed a strong correlation between Arctic haze and mixing ratios of both $CO_2$ and $CH_4$ at the Alert station, on Ellesmere Island, Canada in 1986 demonstrating that synoptic variations in greenhouse gas mixing ratios in the Arctic were due to long-range transport of anthropogenic emissions.

No atmospheric $CH_4$ measurements are reported for Svalbard in the literature until measurements began at Zeppelin (in 1994,

by NOAA). Interestingly however, a 1920s study of natural water springs on Svalbard noted the presence of gas bubbles in numerous streams, and hydrocarbon deposits with natural gas emissions near the surface in Grønfjorden containing 97% $CH_4$ in 1926 (Orvin, 1944). By the time the Zeppelin Observatory had been constructed in 1989, the hypothesis that decomposing hydrates, which only form in the presence of such $CH_4$ seeps if located on the seafloor where pressure is relatively high, could cause run-away warming effects was already the focus of scientific study (Nisbet, 1989).

## 2.3 Early trace gas measurements in the European Arctic

When Arctic haze was identified as the result of long-range transport of pollutants into the polar region, it was clear that these transport episodes could also carry numerous other pollutants. Measurements of carbon monoxide (CO), hydrocarbons and halocarbons at Barrow around 1980 showed highly elevated concentrations in winter compared to in summer (Rasmussen et al., 1983). The first measurements of organic species in the European Arctic were carried out in summer 1982 at four locations:

Bear Island, Hopen, Longyearbyen and Ny-Ålesund and in spring 1983 at Ny-Ålesund (Hov et al., 1984). Samples were collected in stainless steel canisters that were subsequently analysed for halocarbons and non-methane hydrocarbons (NMHC) at the Atomic Energy Research Establishment, Harwell, United Kingdom. The observed fraction of alkanes was higher at Ny-Ålesund than at Barrow, a consequence of the proximity of the petroleum activity in the Soviet Union and the prevailing atmospheric transport from southeast into Spitsbergen. Elevated levels of alkenes (ethene and propene) in summer were linked

to biogenic emissions from the ocean. Even at this early stage it was concluded that "Ny-Ålesund is a good site to measure air coming off the Soviet Union and Europe. Continued sampling could provide valuable information about questions related to global climate, Arctic haze and the chemical composition of the troposphere" and that there was a "need for continued measurements of organic gases [...] at a representative Arctic site like Ny-Ålesund" (Ottar et al., 1986).

$C_2$ to $C_6$ NMHC were sampled at weather ship *M* (located at 66°N, 3°E) and Ny-Ålesund in winter to spring 1985 and in spring

1986 at Ny-Ålesund (Hov et al., 1989), away from the settlement following a snow-scooter ride out to an upwind, unpolluted site. The results showed that the sum of alkanes and alkenes at Ny-Ålesund in spring was close to half the level found just 60





km downwind of London and higher than observed at a rural site in Germany. Given these findings, it was concluded that despite being further north than Barrow, Ny-Ålesund is more influenced by transported pollutants, especially in spring. NMHCs control ozone ($O_3$) production and rates of sulfate and nitrate formation. NMHC levels at Ny-Ålesund were found to

be one order of magnitude higher in spring than in summer (Hov et al., 1984) and modelling indicated that some of the spring increase in $O_3$ is due to tropospheric formation from NMHC build-up in winter. High NMHC levels at Ny-Ålesund and other Arctic sites during the 1980s, and the growing awareness of the importance of these species for atmospheric oxidising capacity, tropospheric ozone, and acid deposition made it clear that a dedicated effort was needed "to sample and analyze such HCs at a representative station network over several years" (Hov et al., 1989). The formation of photooxidants was also a topic of

concern at lover latitudes leading to the establishment of the EUROTRAC-TOR network in Europe. TOR (Tropospheric Ozone Research) was an 8-year project under the EUREKA environment programme that started in 1987, setting up a network of surface monitoring sites for $O_3$ and precursors (Isaksen, 1988). Most of the sites were also active in EMEP.

## 3 The Zeppelin Observatory

The Zeppelin Observatory is located on the Zeppelin Mountain at 472 m above sea level on the Brøgger Peninsula, Svalbard,

Norway (Fig. 1) and is in the Northern Arctic Tundra Zone. Surrounding the Brøgger Peninsula are the waters of the 26 km long Kongsfjorden, while the peninsula itself is a mountainous, barren landscape of scree, occasional patches of thin topsoil with little to no vegetation, and plains with snow-packs or glaciers at lower altitudes. The climate at the observatory reflects its high latitude, but is moderated by the North Atlantic Current, with substantially higher temperatures than elsewhere at corresponding latitudes. The mountain itself is named after Ferdinand Graf von Zeppelin, German officer and designer of

airships, who visited the area during an expedition in 1910.

### 3.1 History and construction of the site

In the 1980s a search began for potential sites for background observations of the atmospheric chemical composition of the European Arctic. The criteria for such a site with respect to background GHG measurements were 1) Minimal local emissions; 2) weak surface exchange such that surface measurements represent the total column; and 3) no expected change in land use

over a decadal time span. Initial $CO_2$ analyses (Sect 2.2) indicated that Bear Island was a favourable location. However, there were indications of local sulfate contamination on the island, since the atmospheric sulfate levels did not drop in summer as expected. And, crucially, access would be limited to the summer months. Thus, the Norwegian settlements on Spitsbergen offered prime possibilities, and after short-term experiments at several valley sites in the Ny-Ålesund area, the Norwegian plan to establish a monitoring observatory on Mt Zeppelin close to Ny-Ålesund emerged.

The main disadvantage of a monitoring observatory at Ny-Ålesund was the potential for contamination from the settlement and from Norwegian and Soviet Union coal mining activities and power stations on Svalbard. The experience derived from the atmospheric chemical observations in the late 1970s and 1980s showed that local air pollution from the Ny-Ålesund





settlement, including traffic on land, electricity generation, waste disposal, smouldering coal heaps and traffic at sea and in the harbour meant that it was necessary to take special precautions to minimise these local impacts. By 1982 NILU had already

moved its observations from the harbour in Ny-Ålesund (NILU-1, Fig. 1) to Gruvebadet, 1.5 km outside Ny-Ålesund and close to sea level at the foot of Mt Zeppelin, to minimise local influences. However, it was found that even at Gruvebadet local impacts could be a problem during episodes of wind from some directions or stagnant air over the coastal plateau in Ny-Ålesund, significantly reducing the sampling frequency of true background Arctic air (Hov and Holtet, 1987), and hence a new location was required for such measurements.

The aim of a permanent observatory of atmospheric chemical composition in Ny-Ålesund was to establish the background, or baseline concentration levels during seasons with very little long-range transport pollution in the boundary layer. An assessment was therefore made of how to minimise local pollution impacts, however minor. Based on numerous radio soundings of the lower troposphere in Barentsburg (Spitsbergen), it was concluded that surface inversion was common during winter and spring, but usually with a depth <300 to 400 m. An observatory on Mt Zeppelin would thus remain above the

surface inversion (Hov and Holtet, 1987). It was further concluded that an observatory on Mt Zeppelin would be in stratus clouds or orographic clouds ≈20% of the time in summer, less during the rest of the year. This conclusion was based on extensive climatological tabulations of the meteorological observations in Ny-Ålesund 1971 to 1980 (Steffensen, 1982).

The final decision to build the observatory on the Zeppelin mountain was taken in 1987 by the Norwegian Ministry of Environment. In early spring 1988, a Norwegian governmental directorate (SBED-Statens bygge og eiendomsdirektorat) was

given the task to build the observatory, and the actual work on site was carried out in the summer of 1988. Access via a lift was commissioned in 1988 and the installation was carried out during the summer 1989. The total cost of the observatory and lift was 11.4 million Norwegian Kroner, funded by the Ministry of Environment. NPI is the owner of the observatory while all three partners (NPI, NILU and Stockholm University -SU) form the consortium responsible for its operation. Funding for scientific equipment and research programmes came from the Royal Norwegian Council for Scientific and Technical Research

(NTNF), later merged into the Research Council of Norway (RCN). Later funding from the Swedish EPA in 1994 allowed the construction of a roof over the arrival space for the lift and the entrance to the observatory, a necessity for safety reasons due to snow drift which at times prevented safe access to the observatory. Due to structural problems with the first building (water leaks, poor insulation, and larger snow loads than anticipated), but more importantly the need for more space for instrumentation, the building was replaced in 1999 (inaugurated May 2, 2000) with the successor of SBED, Statsbygg, as

responsible builder again, and with funding from the Norwegian Ministry of Environment. ≈33% of the investment for the new building came from the Swedish Wallenberg foundation.

The previous background measurement site, Gruvebadet (now 'Gruvebadet Atmosphere Laboratory'), remains an active site for environmental studies, including of aerosol chemical physical properties (e.g. Lupi et al., 2016;Stathopoulos et al., 2018). Other atmospheric observing platforms in Ny-Ålesund include the 'Alfred Wegener Institute/ Institut Polaire Français Paul

Emile Victor (AWIPEV) atmospheric observatory' (Neuber, 2006), NPI's Sverdrup Station Institute (where NILU operates a number of atmospheric monitoring instruments), the 'Ny-Ålesund [Japanese] National Institute of Polar Research (NIPR)





observatory' and the 'Amundsen-Nobile Climate Change Tower' (Mazzola et al., 2016). Together, these platforms are a key component of the 'Ny-Ålesund Atmosphere Flagship', a collaborative effort by researchers from the numerous institutions conducting research and monitoring at Ny-Ålesund to improve data sharing and enhance research outputs (Neuber et al., 2011).

In recent years, this cooperation has been further intensified through the Svalbard Integrated Arctic Earth Observing System (SIOS; https://sios-svalbard.org/) as a Norwegian-led European infrastructure initiative addressing ongoing changes in the Arctic. All international partners listed above, and other international institutions active in Svalbard, are members of this new network.

## 3.2 Atmospheric transport aspects

In the 1980s much was learned about the meteorological conditions leading to the episodic nature of atmospheric aerosol loadings in the Arctic. If the atmospheric processes are assumed to be nearly adiabatic, possible source areas of Arctic air pollution at the ground level are confined to regions with almost the same temperature as the Arctic itself (Iversen, 1984, 1989a, b). Hence, Svalbard generally offers a pristine Arctic environment for environmental monitoring, where anthropogenic influence is very small. However, the Zeppelin Observatory is located only ≈2 km from the Ny-Ålesund settlement (near sea

level), and an important question is whether local emissions can be transported up to the mountain, influencing measurements. The local wind field, strongly influenced by Kongsfjorden and surrounding topography is complicated and thus the winds measured at Ny-Ålesund and at the Zeppelin Observatory can be quite variable (Beine et al., 2001). Katabatic winds coming down from the Kongsvegen glacier, wind channeling in the fjord and the thermal land-sea breeze circulation are all important (Esau and Repina, 2012). Nevertheless, winds blowing directly from the settlement to the observatory are rare and the Zeppelin

Observatory is mostly isolated from emissions in Ny-Ålesund by the frequent presence of temperature inversions below 500 m altitude (Dekhtyareva et al., 2018). Possible exceptions are only the rather infrequent periods with northerly flows (Beine et al., 2001). Consequently, local emissions from Ny-Ålesund have a much stronger influence on chemical measurements near the sea level than at the Zeppelin Observatory, where local pollution episodes are difficult to detect at all (Dekhtyareva et al., 2018). The clearest (but still relatively small) influence was demonstrated for the emissions of cruise ships visiting Ny-Ålesund

(Eckhardt et al., 2013;Dekhtyareva et al., 2018), for which plume rise may be an important mechanism in transporting the exhaust to higher altitudes. A ban on heavy fuel oil use close to the Svalbard coast, however, has reduced ship traffic emissions considerably in Ny-Ålesund since 2015. In summary, the Zeppelin Observatory is representative of the larger-scale conditions in the Svalbard area, and long-range rather than local-scale transport is the dominant mechanism by which pollution reaches the observatory.

With respect to long-range transport of air masses, the location of the Zeppelin Observatory on the western coast of the Svalbard archipelago is important. The West Spitsbergen Current, the northernmost branch of the North Atlantic Current, brings relatively warm ocean waters and keeps the sea largely free of ice even in winter, in contrast to the east side of Svalbard. Air masses arriving at the observatory from the Greenland Sea and the Norwegian Sea are consequently relatively warm,





whereas air masses arriving from the Barents Sea and the Arctic Ocean are much colder, particularly in winter.

Correspondingly, exposure of the air to open sea water versus sea ice depends strongly on where the air is coming from.

In terms of interpreting the chemical composition of the air at the Zeppelin Observatory, we are mostly interested in where the arriving air had recent contact with the surface, where both natural and anthropogenic emissions primarily occur. Figure 3 shows an ≈5-year climatology of the 'footprint emission sensitivity' (based on 50-day backward simulations with the FLEXPART particle dispersion model, Stohl et al., 2005). The FLEXPART footprint is a 2D data field showing the sensitivity

of the receptor (here the Zeppelin Observatory) to emissions at the surface (the 'source/receptor relationship') for all grid cells in the domain, accounting for horizontal/vertical transport, chemical reactions, and where applicable, particle wet and dry deposition. I.e., for a given flux of a component in one grid cell the quantity reaching the observatory is known/modelled.

The simulations were done for a black carbon tracer, for which dry and wet deposition were accounted for. This reduces the emission sensitivity backwards in time. While the details of the footprint emission sensitivity maps depend on the lifetime of

the model tracer used, the maps clearly indicate where air masses arriving at the observatory had recent surface contact. For comparison, very similar results are shown in Fig. 1 of Hirdman et al. (2010) for a passive tracer. Figure 3 (right panel) shows that in summer the emission sensitivity is mostly restricted to ocean areas and does not extend deeply into the continents. Transport modeling thus further supports earlier conclusions by e.g., Iversen (1984) that transport is only from regions of a similar potential temperature, a consequence of the so-called 'polar dome' that prevents warmer continental air masses from

entering the Arctic lower troposphere (Stohl, 2006). In contrast, during the Arctic haze season (defined here as the period December to March, Fig. 3, left), transport of emissions takes place particularly from Northern Europe and Siberia, as illustrated by the elevated emission sensitivities there. Similar findings are documented by Potential Source Contribution Function (PSCF) modelling of equivalent black carbon observations (eBC: black carbon calculated from absorption measurements) for the cold and warm periods in Eleftheriadis et al. (2009).

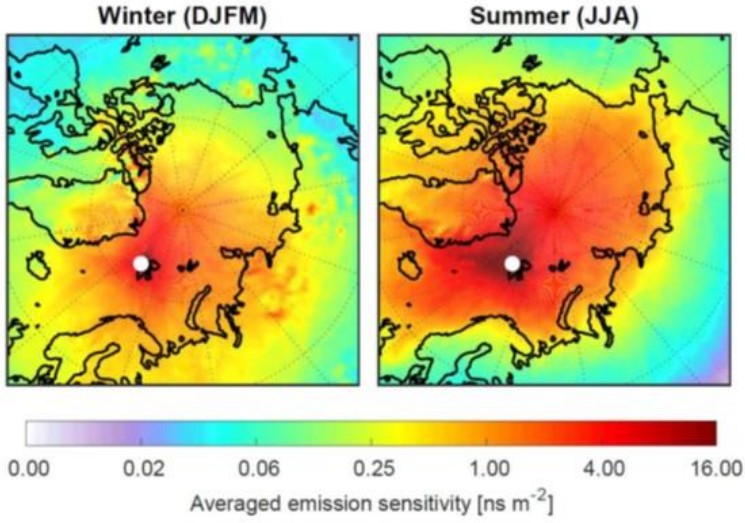




**Figure 3: Footprint emission sensitivity for a black carbon tracer obtained from FLEXPART 50-day backward calculations based on operational meteorological analyses for the period February 2014 to November 2018. The left panel shows the Arctic haze period (December, January, February, March), the right panel the summer period (June, July, August). The unit of the footprint is nanoseconds (ns) m$^{-2}$, demonstrating that longer residence time of an air mass over the surface leads to higher sensitivity to emissions. The concentration change at the receptor is the product of flux × sensitivity.**

For aerosols such as sulfate and black carbon, more efficient scavenging in summer is also an important factor shaping seasonal variations. This is likely a result of the transition from ice-phase cloud scavenging to the much more efficient warm cloud scavenging and the appearance of drizzle in the summer boundary layer in the Arctic (Browse et al., 2012). The relative contributions of seasonal variations in transport, scavenging and changes in emissions are still debated, since models have problems reproducing the observed seasonal cycles of aerosols at the Zeppelin Observatory and at other Arctic stations (Eckhardt et al., 2015). However, the seasonality of atmospheric transport, particularly transport from major source regions in Northern Eurasia, certainly plays an important role (Stohl, 2006;Freud et al., 2017). It has also been noted that at the Zeppelin Observatory the transition from Arctic haze conditions to the much cleaner summer conditions can occur very rapidly (within a few days). At the same time there is usually a shift in the aerosol size distribution from dominant accumulation mode towards smaller Aitken mode particles (e.g. Tunved et al., 2013) indicating a very different origin of the chemical load observed.

Transport modelling can also be used to investigate the sources of air pollutants measured at the Zeppelin Observatory. As an example, we have used aethalometer measurements of eBC. The instrument and dataset has been described earlier in Eleftheriadis et al. (2009). We have sorted the aethalometer data into the top (90%) and bottom (10%) of the aerosol absorption coefficient and show the footprint emission sensitivities for these deciles in Fig. 4, left and right panels, respectively, both for the Arctic haze period (Fig. 4, top) and for summer (Fig. 4, bottom). We see that in winter the lowest eBC is transported almost exclusively from the North Atlantic sector, where there are few eBC sources and where scavenging in frontal systems is efficient. In contrast, the highest eBC concentrations are transported over the Arctic Ocean (where there is little scavenging in winter) and the high values of emission sensitivities extend deeply into Siberia and Eastern Europe. I.e., when the polar front is located north of the main pollutant source regions, the pollution concentrations in the Arctic boundary layer are low. When the polar front is south of important pollution sources, e.g., in northern Russia, the pollution levels in the Arctic boundary layer can be much higher. This confirms earlier suggestions that these are the major source regions of eBC measured at Zeppelin Observatory (Eleftheriadis et al., 2009;Hirdman et al., 2010).

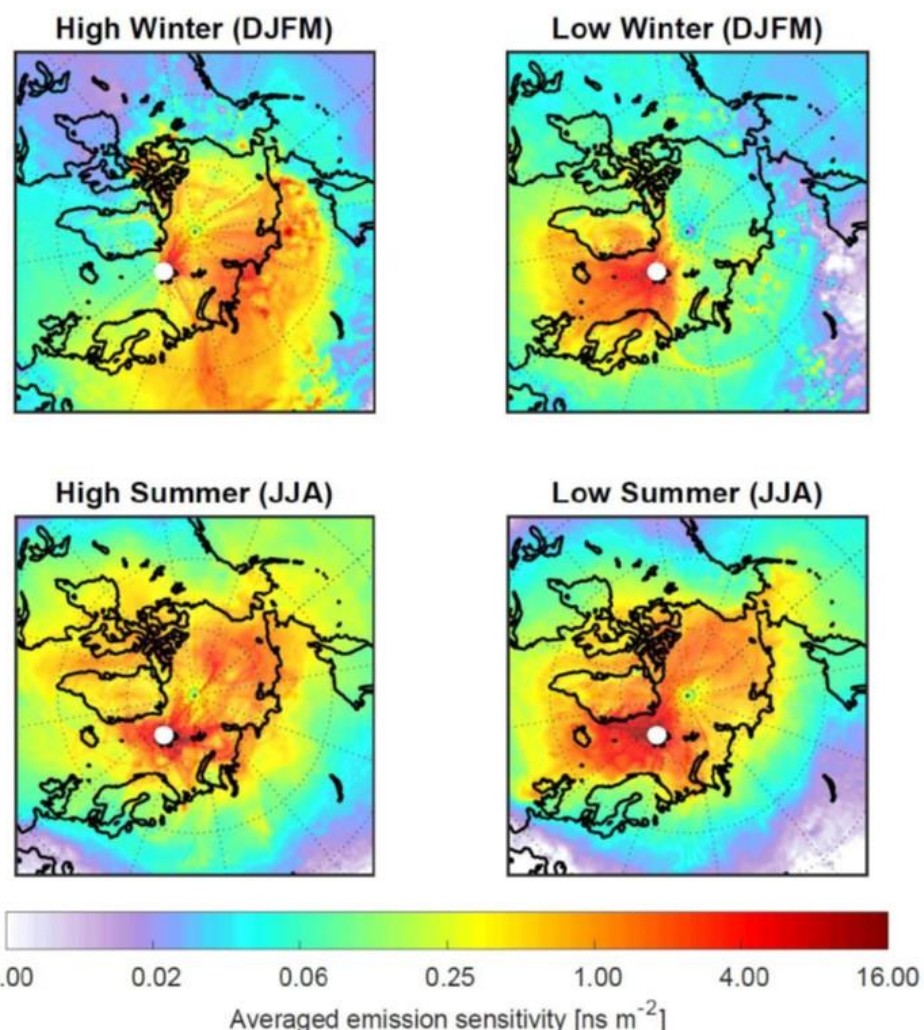

**Figure 4: FLEXPART footprint emission sensitivities for a black carbon tracer for the 10% highest (left) and 10% lowest (right) values of the measured aerosol absorption coefficient (equivalent black carbon), based on data for the years 2014 to 2017.**

In summer, the lowest eBC concentrations are again associated with transport from the North Atlantic but also with transport from the Arctic Ocean, where scavenging in stratus clouds is efficient. The transport for the highest eBC concentrations in summer does not occur over the North Atlantic Ocean and the emission sensitivities again extend into Siberia, albeit much less

extensively than in winter. While we chose eBC as an example, similar results are obtained for most other aerosols, e.g., sulfate (Hirdman et al., 2010) and gases, e.g., methane (Pisso et al., 2016). Particularly intensive pollution episodes can be observed at the Zeppelin Observatory when direct transport from the continent occurs during periods of intensive biomass burning there (Stohl, 2006;Eckhardt et al., 2007;Stohl et al., 2007).



## 4 Instrumentation and measurements

The unique location of the Zeppelin observatory with strong institutional support for the science and operations, makes the Zeppelin observatory an ideal platform for measurements of numerous atmospheric constituents, including for monitoring and field campaigns, and participation in international measurement programmes, as detailed here.

### 4.1 Aerosol chemical composition

Measured aerosol chemical constituents at Zeppelin include levels of inorganic ions and inorganic species, and primary

biological aerosol particles (PBAP) and other carbonaceous species including eBC, and online measurements of non-refractory species (species such as organic aerosol and ammonium sulfate/nitrate that vaporise rapidly at ≈600 °C under vacuum), (Table 1). Heavy metals and mercury, also particulate species, are discussed separately in Sect 4.6.

**Table 1: Aerosol composition/inorganic species measurements at the Zeppelin Observatory, listed chronologically by measurement starting year. See table footnotes for full lists of abbreviations.**

| From | Parameter | Instrument/ Sample[1] | Responsible institution[2] | Comments |
|---|---|---|---|---|
| 1989 | Inorganic ions, Total sulfur | Filter 3-pack, ICPMS | NILU | Total sulfate since 1989, NH4+, NO3- , Ca2+, K+, Cl-, Na+ since 1993. Daily 2001-2002, weekly June-December 2003, hourly April 2003-June 2005. Open filter face. |
| 1998 | Equivalent black carbon (eBC) | Aethalometer | NCSR Demokritos | Magee AE31, absorption, λ=[370;470;520;590;660;880;950] nm |
| 2002 | eBC | PSAP | | λ= 525 nm 2002 to 2013. PSAP with automatic filter change λ=525 nm 2012 to 2016. |
| 2006 | Elemental carbon/ organic carbon (EC/OC) | TOA | SU, NILU | Leckel filter sampler, weekly, similar to NIOSH (2006 to 2012); EUSAAR2 (2009 to present) |
| 2008 | Pollen | Pollen trap | Bjerkenes Centre | Yearly |
| 2010 | Dimethyl sulfide (DMS) | GC-PFPDDMS | KOPRI | |
| 2014 | eBC | MAAP | SU | ThermoFisherScientific Inc., Model5012, λ= 637 nm |
| 2015 | eBC | Aethalometer | NILU NCSR Demokritos | Magee AE33, absorption λ=[370;470;520;590;660;880;950] nm |
| 2017 | Organic tracers | HVS, UHPLC Orbitrap ESI- | NILU | Weekly, tracers of biomass burning, primary biological aerosol particles, biogenic secondary organic aerosol |
| 2017 | EC/OC | HVS, TOA | NILU | In PM10. Weekly, EUSAAR2 2017-present |
| 2019 | Aerosol composition | ToF-ACSM | NILU | Non-refractory, species vaporising below ~600℃. |
| 2019 | Refractory BC (rBC) | SP2-XR | PSI | Refractory, not vaporising at ≈600℃ |





| 2019 | Inorganic species/ aerosol deposition | VTDMA | SU | |
|------|----------------------------------------|-------|-----|---|
| [1]ICPMS: inductively coupled plasma mass spectrometry; HVS: high volume sampler; TOA: thermal optical analysis; GC-PFPD: gas chromatography equipped with a pulsed flame photometric detector; PSAP: particle soot absorption photometer; MAAP: multi angle absorption photometer; UHPLC: ultra-high-pressure liquid chromatography; ESI-:electrospray ionization in negative mode; ToF-ACSM: time-of-flight aerosol chemical speciation monitor; SP2-XR: single particle soot photometer-extended range; VTDMA: volatility tandem differential mobility analyser. [2]NILU-Norwegian Institute for Air Research; NCSR Demokritos-Institute of Nuclear and Particle Physics; NTNU-Norwegian University of Science and Technology; SU-Stockholm University; KOPRI-Korea Polar Research Institute; PSI-Paul Scherrer Institute. | | | | |


The main inorganic anions ($SO_4^{2-}$, $NO_3^-$, $Cl^-$) and cations ($NH_4^+$, $Ca^{2+}$, $Mg^{2+}$,$K^+$, $Na^+$) in air are sampled daily using a 3-stage filter pack for both gaseous and particulate-bound components (noting that species such as ammonium nitrate partition between particle and gas phases). The first stage is an aerosol filter (Zeflour Teflon 2 µm pore, 47 mm diameter, Gelman Sciences), followed by an alkaline potassium hydroxide (KOH) impregnated cellulose filter (Whatman 40) for $HNO_3$, $SO_2$, $HNO_2$, HCl,

and other volatile acidic substances. $HNO_3$ and $SO_2$ react with KOH producing nitrate and potassium sulfite. Oxidising species in air such as ozone, are believed to convert most of the sulfite to sulfate during sampling. The final, oxalic acid-impregnated, filter (Whatman 40) absorbs alkaline species such as $NH_3$. The filter pack method is biased in separating gaseous nitrogen compounds from aerosols and therefore the sum (i.e. $NO_3^- + HNO_3$ and $NH_3 + NH_4^+$ in µg nitrogen (N)) is reported. The filter pack has no fixed size cut off, but the effective size cut-off is ≈10 µm, except for episodes with a high sea salt, mineral dust or

bioaerosol content, when larger particles have been observed.

After samples are collected, they are shipped to NILU's laboratory for analysis. The filters are put into test tubes with extraction solvents. The aerosol filters are extracted in Milli-Q water using ultra sonic treatment to obtain complete extraction. Alkaline filters are extracted in a 0.3% hydrogen peroxide solution to oxidize any remaining sulfite to sulfate. The acid impregnated filters are extracted in 0.01 M $HNO_3$. The ions are analysed using ion chromatography, whereas $NH_3$ collected on the acidic

filter is determined as $NH_4^+$ using an AutoAnalyzer.

Trends in inorganic ions are evaluated according to the Mann-Kendall Test/Sen's slope. Measurements of inorganic ions and total sulfur were an initial focus of atmospheric composition measurements in the Arctic and on Svalbard (section 2.1). They are therefore some of the first measurements recorded at the observatory, and even when excluding prior measurements at Gruvebadet and Ny-Ålesund, the 30-year time series from the Zeppelin Observatory are among the longest in the world. The

data are reported to the Norwegian national monitoring programme e.g. Aas et al. (2019) and to EMEP (Tørseth et al., 2012). Collection of aerosol filter samples for subsequent analysis of elemental carbon/ organic carbon (EC/OC) in aerosol via a whole air inlet, with an effective cut-off of ≈40 µm (Karlsson et al., 2020), began by SU in 2006 (Hansen et al., 2014). Samples are collected at weekly intervals onto pre-heated (800C for 30 min) quartz filters (Munktell & Filtrak GmbH, diameter 47 mm, grade T293) using a low-volume aerosol filter sampler (38 L min$^{-1}$, Leckel Sequential Sampler SEQ 47/50, Leckel GmbH,

Germany). EC/OC was quantified initially according to a thermal optical analysis (TOA) protocol similar to NIOSH (transit



time 800 seconds, sometimes 780 seconds in early cases, same temperature ramps as NIOSH), and later from 2009 (three years of overlap) according to the EUSAAR-2 temperature program (Cavalli et al., 2010).

In parallel to the EC sampling via the whole-air inlet, sample collection for analysis of EC/OC and organic tracers in PM$_{10}$ began in 2017 using a high-volume sampler with a PM10 inlet operated at a flow rate of 40 m3 hr$^{-1}$ and a filter face velocity of 72.2 cm s$^{-1}$. Aerosol particles are collected on pre-fired quartz-fiber filters (PALLFLEX Tissuequartz 2500QAT-UP; 150 mm in diameter) for one week, and according to the quartz fiber filter behind quartz fiber filter (QBQ) set up for an estimate of the positive sampling artefact of OC (Turpin et al., 1994;McDow and Huntzicker, 1990). The filters are shipped to NILU for thermal-optical analysis (TOA), using the Sunset Lab EC/OC Aerosol Analyzer operated according to the EUSAAR-2 temperature program (Cavalli et al., 2010) and using transmission for charring correction. The instrument's performance is regularly intercompared as part of the joint EMEP/ACTRIS quality assurance and quality control effort. The EC/OC data in PM$_{10}$ are reported as part of the Norwegian national monitoring programme (Aas et al., 2020).

From the same PM$_{10}$ filters, tracers for Biomass burning (BB, monosaccharide anhydrides), biogenic secondary organic aerosol (BSOA) precursors (2-methyltetrols), and primary PBAP (ugars and sugar-alcohols) are quantified using ultra high-performance liquid chromatography (UHPLC) connected to an orbitrap mass spectrometer (Q-exactive plus) operated in the negative electrospray ionization (ESI-) mode (Dye and Yttri, 2005;Yttri et al., 2021). Separation was performed using two columns (2 mm × 2.1 mm × 150 mm HSS T3, 1.8 µm, Waters Inc.). Species are identified based on retention time and mass spectra of authentic standards. Isotope-labelled standards are used as internal recovery standard. The limit of detection ranges from 1–10 pg m-3. A high-resolution time-of-flight aerosol chemical speciation monitor (HR-ToF-ACSM, Aerodyne), measuring non-refractory organic aerosol, sulfate, nitrate, ammonium, and chloride, has been in operation at the Zeppelin Observatory since 2016.

There are several ongoing parallel aerosol absorption/ eBC measurements ongoing at the Zeppelin Observatory. SU operates a custom-built particle soot absorption photometer (PSAP) which was accompanied by a multi-angle absorption photometer (MAAP) in 2014. NCSR Demokritos has operated a seven wavelength aerosol absorption photometer (Magee Scientific, AE31 aethalometer) at Zeppelin since the 1990s (Eleftheriadis et al., 2009) and since 2015, NILU and NCSR Demokritos have jointly operated a newer model seven wavelength aerosol absorption photometer (Magee Scientific, AE33 aethalometer) with automatic 'two spot' compensation of the filter loading effect.

### 4.2 Aerosol physical properties

Aerosol physical properties have been measured at Zeppelin Observatory from the start in 1989, providing one of the longest time series of aerosol optical and physical properties from the polar regions. All in-situ instrumentation sample from a whole-air inlet (combined aerosol and cloud particles). The inlet system follows the GAW guidelines for aerosols sampling (Kazadzis, 2016) and has a cut-off of ≈40 µm (Karlsson et al., 2020). All aerosol properties are sampled at low RH conditions due to the strong temperature gradient between ambient and indoors, and no active drying is needed.



The first aerosol physical observations started with continuous nephelometer measurements of aerosol light scattering. The total particle number concentration is measured by condensation particle counters (CPCs, TSI Inc., Model 3025 and 3010).

The size distribution of sub-micron particles is recorded by using a custom-built closed-loop differential mobility particle sizer (DMPS) system since 2000 (see e.g. Tunved et al., 2013). The system has been continuously improved and now measures the particle size distribution from around 5 to >800 nm mobility diameter with a synchronised twin-DMPS system. 'DMPS 2a' (Fig. 5) measures at 5 to 57 nm with a smaller differential mobility analyser (DMA) and 'DMPS 2b' (Fig. 5) measures at 20-809 nm with a larger DMA. Both DMPS use a CPC (TSI Inc., USA, Model 3010) behind the DMA and a CPC (TSI Inc., USA,

Model 3010) for measuring the total aerosol particle concentration. Coarse mode aerosol has been continuously recorded since spring 2018 with an optical particle size spectrometer (OPSS, Fidas 200 E, Palas GmbH), which is situated on the measurement platform of Zeppelin Observatory. At the same time, measurement programmes have been added from other institutions notably in South Korea and Japan (Table 2).

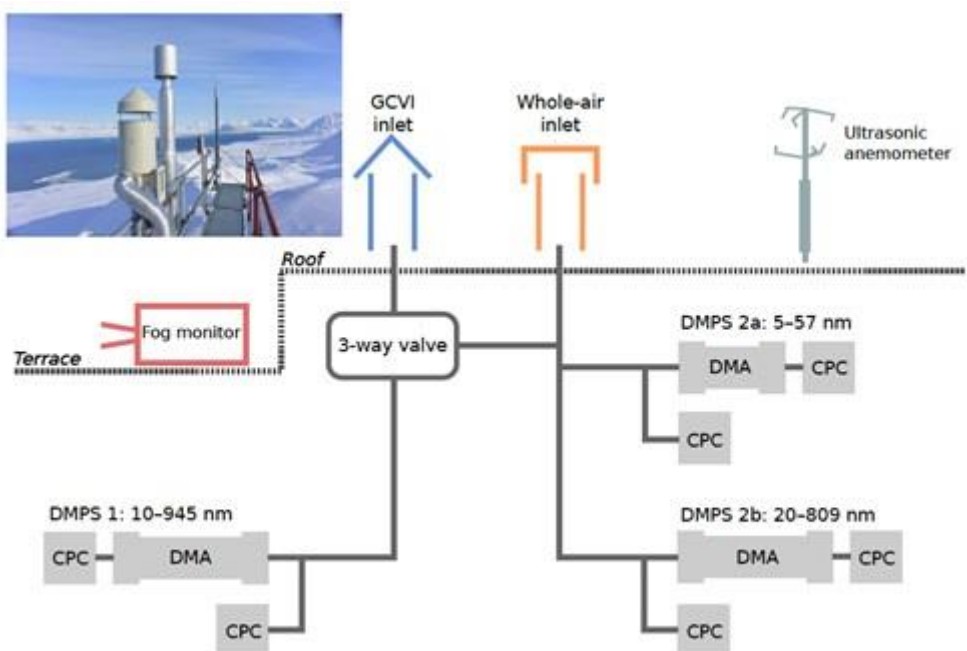

**Figure 5: Schematic illustration of the experimental set-up at the Zeppelin Observatory. The diagram shows how the whole-air inlet (orange) and the ground-based counterflow virtual impactor (GCVI) inlet (blue) are connected to the differential mobility analysers (DMAs) and condensation particle counters (CPCs). The 3-way valve switches the sample flow to the instruments on the left-hand side from the GCVI inlet to the whole-air inlet when there is no cloud to be sampled. Cloud sampling is activated if the visibility drops below 1 km (measured by a visibility sensor (not pictured) next to the GCVI inlet). Auxiliary measurements from a fog monitor**
**and an ultrasonic anemometer have also been included in the data analysis. Figure from (Karlsson et al., 2021).**

**Table 2: Aerosol physical property measurements at the Zeppelin Observatory and measurement owner (institute), listed chronologically by measurement starting year. See table footnotes for full lists of abbreviations.**

| From | Parameter | Instrument/ Sample[1] | Responisble Institution[2] | Comments |
|------|-----------|----------------------|---------------------------|----------|
|      |           |                      |                           |          |





| 1989 | Aerosol light scattering | Nephlometer | SU | 1989 to 1997: custom built λ=550 nm; 1997 to present TSI model 3563 λ= [450; 550;700] nm |
|---|---|---|---|---|
| 1989 | Particle number | CPC | SU | TSI models 3025 and 3010 |
| 2000 | Particle number size distribution | DMPS | SU | Synchronised twin DMPS system. Lower size using small differential mobility analyser (DMA, length 0.053 m, outer radius 0.033 m, inner radius 155 0.025 m) larger particles with large DMA (length 0.28 m, outer radius 0.033 m, inner radius 0.025 m). |
| 2007 | Cloud condensation nuclei | CCNC | KOPRI | Supersaturation: 0.2, 0.4, 06, 0.8, and 1.0% |
| 2013 | Fog | Fog monitor | NIPR | Droplet Measurement Technologies Inc., USA, Model FM-120 |
| 2015 | Ice nucleating particles | Aerosol sampler | NIPR | Sources, compositions, and concentrations of aerosol particles acting as ice nucleating particles under mixed-phase cloud conditions |
| 2016 | Particle number size distribution | Nano SMPS | KOPRI GIST | $D_P$ >3 nm |
| 2016 | Particle number size distribution | DMPS | NILU | $D_p$ 10 to 800 nm |
| 2017 | Cloud Particles | Hawkeye | NIPR | |
| 2018 | Aerosol light scattering | Nephlometer | SU | Ecotech, backscatter λ= [450;525;635] nm |
| 2018 | Cloud droplets | CVI, OPS | SU | Cloud droplet sampling via CVI, including particle size, FIDAS-PALAS Optical Particle Sizer |
| 2019 | Particle number size distribution | APS | NILU | Aerodynamic diameter 0.5 to 20 µm. Light-scattering intensity 0.3 to 20 µm. |

[1]CPC: condensation particle counter; DMPS: differential mobility particle sizer; PFR: Precision filter radiometer; SMPS: Scanning Mobility Particle Sizer; CVI: counterflow virtual impactor; OPS: optical particle sizer; APS: aerodynamic Particle sizer.
[2]SU-Stockholm University; NILU-Norwegian Institute for Air Research; KOPRI-Korea Polar Research Institute; PSI-Paul Scherrer Institute; PMOD- Physikalisch-Meteorologisches Observatorium Davos; WRC-World Radiation Center; NIPR-National Institute of Polar Research [Japan]; GIST- GwangJu Institute of Science and Technology [Korea]


To further investigate links between aerosol composition, physical properties and the Arctic climate, a major effort in long term observation of the interaction between the aerosol and clouds began at the Zeppelin Observatory in 2018. As the observatory is often in low level clouds it is a unique site in the Arctic. General observations of the total aerosol are complemented by a similar instrumental set up of 'cloud residuals', i.e., those particles which have been involved in cloud

formation as CCN or ice nucleating particles INP. CCN and INP are sampled through a special inlet, the 'counter-flow virtual impactor' (CVI), separating the droplets or ice crystals from other non-activated particles in the cloud. These cloud droplets or crystals are then dried and the cloud residual is measured by the set of aerosol instruments connected to the CVI (Fig. 5). The cut-off size is typically 6 to 7 µm aerodynamic diameter (Karlsson et al., 2020). With this set-up the Zeppelin Observatory is now one of the first global aerosol observatories with semi-continuous in-situ cloud sampling.



### 4.3 Climate gases

Climate gases/precursors with mixed biogenic/anthropogenic sources to the atmosphere currently measured at Zeppelin Observatory are: $CO_2$, $CH_4$, CO, nitrous oxide ($N_2O$), reactive VOCs (e.g. ethane, propane, see also Sect. 4.4) and chloro- or bromo-methane. Purely anthropogenic gases include chlorofluorocarbons (CFCs), hydrochlorofluorocarbons (HCFCs), hydrofluorocarbons (HFCs) and halons. Additional information on $CO_2/CH_4$ is also provided by measurements of isotopic composition. These compounds have variously been adopted as part of national monitoring programmes, while the analysis techniques deployed have often allowed for measurements of other compounds not part of the monitoring programmes.

Stockholm University began $CO_2$ measurements in 1989 as part of the Global Atmosphere Watch (GAW) programme using infrared measurements. For 2001 to 2012 $CH_4$ was measured using a GC-FID system with an inlet 2 m above the observatory roof with a precision of $\pm 3$ ppb at hourly resolution, determined via calibrations to working standards calibrated to NOAA reference standards. Through the same inlet, CO was measured at 20-minute intervals with a mercuric oxide detector (GC-HgO) calibrated to NOAA standards. Since April 2012, $CH_4$, $CO_2$ and CO at Zeppelin have been measured using a cavity ring-down spectroscope (CRDS, Picarro G2401) at 1-minute resolution with a sample inlet 15 m above the observatory roof. The CRDS is measured daily against target gases and calibrated every 3 weeks against working standards, which are calibrated to NOAA reference standards. For both measurement regimes, sampling was done through a Nafion drier to minimise any water correction error in the instruments.

As part of the harmonisation of historic concentration measurements within the INGOS project (INGOS, 2016), the full time series from August 2001 to 2013 was re-processed and archived in the ICOS Carbon portal (Colomb et al., 2018). All original data were reprocessed with improved software, recalculating all measurements from the previous 12 years. This new software facilitates quality assurance and control and detection of measurement errors. For example, although the Zeppelin Observatory is located far from local sources, there are nevertheless occasional large baseline excursions in the mixing ratios due to long-range transport (Stohl et al., 2013;Stohl et al., 2007b). Hence, at least 75% of calculated back trajectories within $\pm$ 12 hours of the sampling day must be from a clean sector (i.e. not from Europe, North America, or Russia), before the data are considered as background mixing ratios (Myhre et al., 2020). The old data were also analysed against new reference standards using new improved instrumentation. All other working standards are linked to these through comparative measurements. Hence, calibrations factors for the first 12-year period were also recalculated during the reprocessing.

The Zeppelin Observatory is now recognised as an ICOS class 1 site for observations of carbonaceous greenhouse gases. I.e. the Zeppelin Observatory fulfils all the core criteria outlined by ICOS required for contribution to a harmonised high-quality global dataset to quantify the exchange of carbon between the surface ocean and the atmosphere, ocean acidification, and interior ocean carbon transport and storage (Yver-Kwok et al., 2020). All data from the EMEP/ICOS measurements are available at ebas.nilu.no (EBAS), the ICOS carbon portal (Colomb et al., 2018), and reported annually e.g. Myhre et al. (2020). Annual trends in atmospheric trace mixing ratios are calculated for clean background data according to Simmonds et al. (2006)





whereby the change in atmospheric mixing ratio of a species as a function of time is fitted to an empirical equation combining Legendre polynomials and harmonic functions with linear, quadratic, and annual and semi-annual harmonic terms.

$\delta^{13}C_{CH4}$ measurements (the shift in the carbon-13, $^{13}C$, fraction in methane compared to the Vienna Pee Dee Belemnite

reference standard, VPDB) were initiated by INSTAAR lab (Institute of Arctic and Alpine Research, University of Colorado) in 2001, who also began $\delta D_{CH4}$ (shift in deuterium, D, compared to Vienna Standard Mean Ocean Water, VSMOW) measurements in 2003. The latter programme was cancelled in 2010. Parallel $\delta^{13}C_{CH4}$ measurements by NILU along with new $\delta D_{CH4}$ measurements began in 2012. In 2017, $\delta^{13}C_{CH4}$ measurements were adopted as part of the Norwegian National Monitoring. The NILU isotope samples are collected in 1 L steel or aluminium canisters at the same air inlet as $CH_4$. Two

samples per week are sent to the Greenhouse Gas Laboratory at Royal Holloway University of London. The $CH_4$ mole fraction is measured using a CRDS (Picarro 1301), while $\delta^{13}C_{CH4}$ analysis is carried out using a modified gas chromatography isotope ratio mass spectrometry system for all samples (Trace Gas and Isoprime mass spectrometer, Isoprime Ltd.) with 0.05‰ repeatability. All measurements for the canisters are made in triplicate. See e.g., Nisbet et al. (2019) for more details. $\delta D_{CH4}$ measurements are performed at University of Utrecht on flask samples using a continuous-flow isotope ratio mass spectrometry

(CF-IRMS) technique with a precision of 2.3‰ (Brass and Röckmann, 2010). A high-resolution (2 minute) instrument for $CH_4$ isotopes ('CH4 Isotope Monitor for $\delta13CH4$ and $\delta CH3D$', Aerodyne) was installed in 2018 by SU, with a precision (30-minute averaging) of 0.1‰ and 3‰ for $\delta^{13}C_{CH4}$ and $\delta D_{CH4}$, respectively.

For 2001 to 2010, measurements of a wide range of HCFCs and HFCs (e.g. HCFC-141b, HCFC-142b, HFC-134a), methyl halides ($CH_3Cl$, $CH_3Br$, $CH_3I$) and halons (e.g. H-1211, H-1301), see Table 3, were measured with an adsorption-desorption

gas chromatograph system (ADS-GCMS) as part of the Advanced Global Atmospheric Gases Experiment (AGAGE) network (Prinn et al., 2008). Many compounds, CFCs and others, were measured with this system, but did not meet AGAGE standards for precision due to unsolved instrumental problems e.g., possible electron overload in detector (for the CFCs), influence from other species, detection limits ($CH_3I$, $CHClCCl_2$) and unsolved calibration problems (as for $CH_3Br$). Thus, since September 2010, the ADS-GCMS was replaced by an online GC-MS (Medusa). The Medusa can be used to measure hydrocarbons (e.g.

Benzene, ethane, n-butane, n-pentane, propane, toluene, for ACTRIS) including the halogenated compounds previously measured by the ADS-GC-MS at the ppt level (Miller et al., 2008) and is calibrated to AGAGE reference standards.

A GC-electron capture detection system (ECD) was used for $N_2O$ with a high time resolution of 15 minutes until 2017. $N_2O$ at Zeppelin is now measured at <1 minute resolution with a mid-IR CRDS (Picarro G5310) which is calibrated against ICOS reference standards (NOAA scale). Instrument data are submitted to ring tests and measurement control / calibration following

ICOS protocols. The high time resolution data are also compared to weekly flask samples sent to NOAA ESRL Global Monitoring Laboratory, Boulder, Colorado.



**Table 3: Measurements of trace gases at the Zeppelin Observatory, the atmospheric constituent(s) measured and measurement responsible institutes, listed chronologically by measurement starting year. See table footnotes for full lists of abbreviations**

| From | Parameter | Instrument/ Sample[1] | Responsible Institution[2] | Comments |
|------|-----------|------------------------|-----------------------------|----------|
| 1988 | Carbon dioxide ($CO_2$) | NDIR | SU | Phased out 2013 |
| 1989 | Ozone ($O_3$) | UV Abs | NILU | |
| 1989 | Non-methane hydrocarbons | Steel cannister, GC-FID | NILU | Daily, phased out 1999 |
| 1994 | Greenhouse gases | Glass bottles | NOAA | From 1994: $CO_2$, $\delta^{13}C_{CO2}$, $\delta^{18}O_{CO2}$, methane ($CH_4$), $\delta^{13}C_{CH4}$, hydrogen ($H_2$), from 1997: Nitrous oxide ($N_2O$), sulfur hexafluoride ($SF_6$), bottles shipped to NOAA laboratory |
| 1994 | Reactive nitrogen ($NO_Y$) | CLD | | CRANOX system coupled to gold converter for $NO_Y$ to NO. Phased out 1997. |
| 1994 | Oxides of nitrogen ($NO_X$) | CLD | | Blue light conversion of $NO_2$ to NO. Phased out 1997 |
| 2001 | Halogenated compounds[3] | ADS-GC-MS | NILU | Daily, Phased out 2011 |
| 2001 | Methane ($CH_4$) | GC-FID | NILU | Custom built GC-FID, phased out 2012 |
| 2001 | Carbon monoxide (CO) | GC-HgO | NILU | Phased out 2012 |
| 2010 | Halogenated compounds[4] | Medusa GC-MS | NILU | |
| 2010 | Non-methane hydrocarbons | Medusa GC-MS | NILU | Benzene, ethane, n-butane, n-pentane, propane, toluene |
| 2012 | Methane isotopic ratio ($\delta^{13}C_{CH4}$, $\delta D_{CH4}$) | Steel cannister, GC-IRMS CF-IRMS | NILU RHUL UU | $\delta^{13}C_{CH4}$ for national monitoring measured at RHUL with GC-IRMS, $\square D_{CH4}$ measured at UU with CF-IRMS |
| 2012 | Greenhouse gases | CRDS | NILU | Carbon dioxide ($CO_2$), methane ($CH_4$), carbon monoxide (CO). ICOS after 2014. |





| 2016 | Dinitrogen monoxide (N$_2$O) | GC-ECD, CRDS | NILU | GC-ECD until 2018, CRDS since 2017 under ICOS |
|---|---|---|---|---|
| 2018 | Methane isotopic ratio (δ$^{13}$C$_{CH4}$, δ$^{12}$C$_{CH4}$, δ D$_{CH4}$) | CH$_4$ Isotope Monitor (Aerodyne) | SU | INTERACT: Distinguishing Arctic CH$_4$ sources to the atmosphere using inverse analysis of high frequency CH$_4$ $^{13}$CH$_4$ and CH$_3$D measurements (IZOMET) |

[1]Non-dispersive infrared radiometer; ADS-GC-MS=adsorption-desorption system gas chromatography with mass spectrometry; GC-FID=gas chromatography with a flame ionization detector; CD=chemiluminescence detector; GC-HgO= gas chromatography with mercuric oxide (HgO) reaction tube; Medusa GC-FID/ECD; GC-IRMS= gas chromatography with isotope ratio mass spectrometer; CF-IRMS= continuous-flow isotope ratio mass spectrometry; CRDS=cavity ring-down spectrometer (Picarro).

[2]SU-Stockholm University; NILU-Norwegian Institute for Air Research; RHUL=Royal Holloway University of London; UU= University of Uttrecht

[3]Chlorofluorocarbons: CFC-11, CFC-113, CFC-115, CFC-12 (not within AGAGE required precision, but part of AGAGE quality assurance programme); halons: H-1211, H-1301, hydrochlorofluorocarbons: HCFC-141b, HCFC-142b, HCFC-22; hydrofluorocarbons: HFC-125, HFC-134a, HFC-152a; bromomethane, chloromethane, dichloromethane, sulfur hexafluoride, tetrachloroethene, trichloroethane, trichloroethene, trichloromethane.

[4]Chlorofluorocarbons: CFC-11, CFC-113, CFC-115, CFC-12; halons: H-1211, H-1301, H-2402, hydrochlorofluorocarbons: HCFC-141b, HCFC-142b, HCFC-22; hydrofluorocarbons HFC-125, HFC-134a, HFC-143a, HFC-152a, HFC-227a, HFC-23, HFC-236fa, HFC-245fa, HFC-32, HFC-365mfc, HFC-4310mee; perfluorocarbons: PFC-116, PFC-14, PFC-218, PFC-318; bromomethane, chloromethane, dibromomethane.


### 4.4 Non-methane hydrocarbons

As part of the EUROTRAC-TOR project (see Sect. 2.3), manual NMHC sampling in steel canisters was initiated at the Zeppelin Observatory when the observatory opened in September 1989 (Hov et al., 1989) (Table 3). The canister samples were collected 2 to 3 times a week with a filling time of 10 to 15 mins and then shipped to NILU's laboratory for chemical analyses. From 1989 to 1991, samples were analysed for nine C2 to C5 NMHCs. From 1992 this was extended to 26 species including aromatic compounds and C6 to C7 alkanes (Solberg et al., 1996a). In some of the following years the samples were collected every day during the spring to capture the strong decline in concentration levels in that season.


In 1992, a pilot measurement programme on light hydrocarbons, aldehydes and ketones was initiated within EMEP (Solberg et al., 1995). A cooperation with the ongoing TOR project was established implying that the monitoring data was reported to





both programmes. The aim of this programme was to collect VOC data at rural European background sites as a support to the modelling activities within EMEP.

As part of this pilot programme, regular sampling of aldehydes and ketones started at the Zeppelin Observatory (and nine other EMEP sites) in April 1994. The carbonyl sampling was done with DNPH adsorption tubes exposed for 8 hours during daytime on the same dates as the NMHC sampling (Solberg et al., 1996a, and refrences therein). This was probably the first routine monitoring programme of carbonyls in the world. When the EUROTRAC-2 program ended, the national funding of the VOC measurements at Ny-Ålesund ended and thus the monitoring of light hydrocarbons, aldehydes and ketones ceased by the end of 1999. In September 2010, an online GC-MS (Medusa) was installed at the Zeppelin Observatory for continuous CFC and HCFC monitoring (see previous section). In 2003, as part of the GAW program, NOAA started scattered measurements of NMHC with glass flasks at the Zeppelin Observatory. In 2006 the sampling of the NOAA flasks was done once a week.

## 4.5 Persistent organic pollutants

NILU's first director, Brynjulf Ottar, hypothesised that some semi-volatile chlorinated hydrocarbons exhibited a potential to undergo reversible atmospheric deposition, making such pollutants prone to a long-term transfer from global source areas in warmer regions and into the Arctic (Ottar, 1981). Measurement campaigns at Ny-Ålesund of a range of such pollutants, now recognised as persistent organic pollutants (POPs), were performed by NILU from 1981 to 1984 (Oehme and Stray, 1982;Oehme and Manø, 1984;Oehme and Ottar, 1984;Pacyna and Oehme, 1988). These early campaigns, combined with air mass back-trajectories, were pivotal in terms of documenting the potential for polychlorinated biphenyls (PCBs) and various organochlorine pesticides (e.g. DDT) to undergo long-range atmospheric transport to the Arctic (Pacyna and Oehme, 1988;Oehme, 1991). Following 8 years without any further measurements, a new sampling campaign was carried out in 1992 (Oehme et al., 1995). An important objective of the latter campaign at Ny-Ålesund was to prepare for regular monitoring of POPs under AMAP (Oehme et al., 1996). Measurements of legacy POPs at Zeppelin, with the aim of an improved understanding of long-range transport of POPs and their spatial and temporal variability, has been a part of the Norwegian national air monitoring programme since 1993. The list of POP compounds included in the monitoring programme is continuously expanded and now also covers POP-like chemicals of emerging concern (POP-CECs). POP data from the monitoring programme are reported to EMEP, and AMAP, and aggregated data are also made available for use by the Global Monitoring Plan (GMP) of the Stockholm Convention on POPs through a data sharing arrangement.

Long-term POP monitoring is based on well-established high-volume active air sampling (HV-AAS) methodology (Bidleman and Olney, 1974). Air is pumped ($\approx$25 m3 h$^{-1}$) through a sampling unit containing a glass fiber filter for particle-bound POPs and two polyurethane foam (PUF) plugs as an adsorbent for gas-phase (volatile) POPs. The sampling interval is 24 to 72 hours, based on a crucial balance of detection and breakthrough of the individual compounds and the interest in studying atmospheric source-receptor relationships using atmospheric transport models, e.g., to track the origin of air masses during interesting episodes (Eckhardt et al., 2007). For some emerging semi volatile organic compounds not retained by PUF (e.g., per- and polyfluoroalkyl substances, PFAS), the PUF plugs are replaced by a PUF/XAD/PUF sandwich. Air samples for more volatile





organic pollutants such as cyclic volatile methyl siloxanes, cVMS are collected at 72-hour intervals using a solid-phase extraction low-volume active air sampler (SPE-LV-AAS) at a flow of ~1 $m^3 h^{-1}$. From 2011 to 2019, this sampler contained

an ENV+ sorbent (hydroxylated polystyrene divinylbenzene copolymer) but was replaced by an ABN adsorbent in 2019 as the cVMS isomers were shown to degrade/transform on the ENV+ sorbent (Krogseth et al., 2013). This highlights the need for continual development of sampling methodologies (Warner et al., 2020).

POPs are also measured in two international passive air sampling (PAS) networks, the Global Atmospheric Passive Sampling (GAPS) network (Pozo et al., 2006) and the monitoring network (MONET)-Europe (Klánová et al., 2009), alongside

occasional PAS campaigns (Halvorsen et al., 2021;Halse et al., 2011) to support the EMEP programme (Tørseth et al., 2012). Passive air samples and active filter samples from the Zeppelin observatory also contribute to the Norwegian Environmental Specimen Bank (ESB) since 2014. The ESB contains and stores environmental samples from different matrices across Norway and acts as an archive for future research on currently unrecognised environmental contaminants, with the goal of supporting future environmental contaminant control strategies (e.g. Giege and Odsjö, 1993). The passive air sampling (PAS) is done on

either a polyurethane foam (PUF) disk or XAD-resin adsorbent, yielding time-weighted averages over the exposure period of 30 days to one year. Data from the PAS networks are reported to GMP and have been crucial for a global spatial coverage of POP data.

**Table 4: Measurements of persistent organic pollutants (POPs) and other environmental contaminants at the Zeppelin Observatory listed chronologically by measurement starting year and responsible institutions. See table footnotes for lists of abbreviations.**

| From | Parameter | Instrument/ Sample[1] | Responsible Institution[2] | Comments |
|---|---|---|---|---|
| 1993 | HCH | HV-AAS | NILU | HCH=α/γ-Hexachlorohexane |
| 1993 | HCB, CD | HV-AAS | NILU | HCB=Hexachlorobenzene; CD=cis/trans-Chlordane |
| 1994 | DDT, PAHs | HV-AAS | NILU | DDT=o,p'/p,p'-Dichlorodiphenyltrichloroethane; PAH=Polycyclic aromatic hydrocarbons |
| 1994 | Heavy metals | HV-AAS ICPMS | NILU | Pb, Cd, As, V, Ni, Cu, Co, Mn, Zn, Cr, Al, Fe, Sn |
| 1994 | Gaseous elemental mercury | CVAFS AFS | NILU NTNU | CVAFS Replaced in 2000 by automated AFS system (Tekran2537) |
| 2000 | Lead-211 | HV-AAS | FMI | 3 samples per week |
| 2001 | PCBs[3] | HV-AAS | NILU | PCB=Polychlorinated biphenyls |
| 2004 | POPs | PUF-PAS | EC | PCBs, PBDEs, HCHs, DDTs, CD, endosulfans, Heptaclor, Heptachlor Epoxide, Dieldrin |
| 2006 | PBDEs[4] HBCDs PFAS[5] | HV-AAS | NILU | PBDE= Polybrominated diphenyl ethers; PFAS= Ionic Per- and polyfluorinated alkyl substances; HBCD= α-, β-, γ-Hexabromocyclododecanes |
| 2007 | Speciated mercury | AFS | NILU NTNU | Tekran Mercury 1130, 1135 and 2537 |





| 2009 | POPs | PUF-PAS | RECETOX | Polyaromatic hydrocarbons (PAH), PCBs, DDTs, HCHs, HCB, PeCB |
|------|------|---------|---------|---------------------------------------------------------------|
| 2013 | cVMS CPs | LV-AAS HV-AAS | NILU | cVMS=D4,D5,D6-Cyclic volatile methylsiloxanes; CPs= Short/medium-chained-chlorinated paraffins |
| 2013 | Volatile POPs | XAD-PAS/ XAD | NILU | e.g. HCB, siloxanes |
| 2017 | vPFAS[6] nBFRs[7] OPFRs[8] Phthalates[9] Dechloranes[10] | | NILU | vPFAS= Volatile PFAS; nBFRs=novel brominated flame retardants (nBFRs); OFPRs= Organophosphorous flame retardants |

[1]PUF-PAS: polyurethane foam passive air sampler; HV-AAS: high volume active air sampler; CVAFS: cold vapour atomic fluorescence spectroscopy; AFS: atomic fluorescence spectrometry; ICPMS: inductively coupled plasma mass spectrometry; XAD- registered trademark of Dow Chemical company, comprises a polystyrene copolymer resin; 'UFO' is a reference to the shape of the sampling system; PAS: passive air sampling

[2]NILU-Norwegian Institute for Air Research; NTNU-Norwegian University of Science and Technology; FMI-Finnish Meteorological Institute; EC-Environment Canada; RECETOX is a research center at the Masaryk University Faculty of Science

[3]PCB-18, 28, 31, 33, 37, 47, 52, 66, 74, 99, 101, 105, 114, 118, 122, 123, 128, 138, 141, 149, 153, 156, 157, 167, 170, 180, 183, 187, 189, 194, 206, 209. Data available before 2001 are classified as uncertain due to possible local contamination.

[4]PBDE-28, 47, 49, 66, 71, 77, 85, 99, 100, 119, 138, 153, 154, 183, 196, 206, 209

5PFPeS, PFHxS, PFHpS, PFOS, PFOSlin, PFNS, PFDS, PFHxA, PFHpA, PFOA, PFNA, PFDA, PFUnDA, PFDoDA, PFTrDA, PFTeDA, PFHxDA, PFODcA, PFOSA, 4:2 FTS, 6:2 FTS, 8:2 FTS, PFBS

[6]4:2 FTOH, 6:2 FTOH, 8:2 FTOH, 10:2 FTOH, N-EtFOSA, N-EtFOSE, N-MeFOSA, N-MeFOSE

[7]ATE (TBP-AE), α-, β, γ/δ-TBECH, BATE, PBT, PBEB, PBBZ, HBB, DPTE, EHTBB, BTBPE, TBPH, DBDPE

[8]TEP, TCEP, TPrP, TCPP, TBP, BdPhP, TPP, DBPhP, TnBP, TDCPP, TBEP, TCP, EHDP, TXP, TIPPP, TTBPP, TEHP

[9]MP, DEP, DPP, DAIP, DIBP, DBP, BBzP, DHP, DEHP, DcHP, DPHP, DINP

[10]Syn-DP, anti-DP, Dec-601, Dec-602, Dec-603, Dec-604, Dba

## 4.6 Heavy metals and mercury

Sample collection of heavy metals including mercury was initiated in 1994 as part of the Norwegian national monitoring programme (Table 4), and data is reported to EMEP and AMAP (Hung et al., 2010). Air samples of HMs (Pb, Cd, As, V, Ni,

Cu, Co, Mn, Zn, and Cr) are collected on paper disc filters (Whatman 41) using a high volume air sampler. An impactor is used as sample inlet to discriminate against particles >2 to 3 µm. The air flow is kept constant at 70 m3 h$^{-1}$ and one 48 h sample is collected weekly. Through 25 years of sample collection, different techniques have been applied to digest the filters. Between 1994 and 2000, filters were digested using nitric acid in closed polytetrafluoroethylene (PTFE) containers at 150°C for 6 to 8 hours. Between 2000 and 2012, microwave digestion was applied using nitric acid and hydrogen peroxide. From 2012 and

onwards, UltraClave microwave digestions were applied using diluted nitric acid. The metals (Pb, Cd, Cu, Zn, Cr, Ni, Co, Mn and As) have been analysed using different inductively coupled plasma mass spectrometry (ICP-MS) instruments (Berg et al., 2004;Berg et al., 2008;Aas et al., 2020). Trends are evaluated by the non-parametric 'Mann-Kendall Test' applied to the annual mean concentrations (Gilbert, 1987), while the Sen's slope estimator is used to quantify the magnitude of the trends.



Gaseous elemental mercury (GEM) species have been monitored using a combination of manual and automated sampling techniques. Between 1994 and 2000, manual measurements were performed based on mercury amalgamation with gold. GEM is sampled by drawing air at a flowrate of 0.7 L min$^{-1}$ through quartz glass tubes containing gold coated quartz glass pieces. Air is drawn through the trap using a pump and the air volume is measured using a volume meter. The gold traps were returned to NILU and analysed by thermal desorption and cold vapour atomic fluorescence spectroscopy (CVAFS, e.g. Brosset, 1987). Samples were collected during 24 h periods once a week.

Automated measurements were initiated in 2000 using a Tekran 2537 Hg vapor analyser detailed in Aspmo et al. (2005). Briefly, ambient air is sampled at 1.5 L min$^{-1}$ through a Teflon filter via a heated sampling line. A Sodalime (NaOH/Ca(OH)$_2$) trap is mounted in-line before the instrument filter. Hg in the air is pre-concentrated for 5 minutes by amalgamation on two parallel gold cartridges, which alternate between collection and thermal desorption, followed by AFS (atomic fluorescence spectrometric) detection. The instrument is auto-calibrated every 25 hours using an internal Hg permeation source, with accuracy verified during routine site audits that include manual injections of Hg from an external source (Aspmo et al., 2005). The detection limits are comparable for both manual and automated methods, at 0.1 ng m$^{-3}$.

Speciated mercury measurements were performed on campaign basis several times particularly during spring (Aspmo et al., 2005;Berg et al., 2003;Sommar et al., 2007), however from 2007 automated mercury speciation using the Tekran Mercury 1130, 1135 and 2537 speciation system was initiated by the Norwegian University of Science and Technology (NTNU). Sample collection and analysis is in detail described elsewhere (e.g. Landis et al., 2002;Steffen et al., 2008). In summary, air is pulled into the analyser through a Teflon coated elutriator and an impactor designed to remove particles > 2.5 µm at flow rates of 10 L min$^{-1}$. The sample air flows over a KCl coated quartz denuder to trap gaseous organic mercury (GOM) and then over a quartz particulate filter to trap particulate bound mercury (PBM). GOM and PBM accumulate for 1 to 2 hours followed by consecutive thermal desorption and AFS by the Tekran 2537, as for gaseous elemental mercury.

## 4.7 Surface ozone

As part of the EUROTRAC project TOR (Tropospheric Ozone Research) and EMEP, continuous monitoring of surface ozone was initiated at Ny-Ålesund in October 1988, then down by the Kongsfjorden shoreline (NILU-1, Fig. 1). The ozone monitor was moved to the Zeppelin Observatory upon opening in September 1989. Surface ozone has been monitored continuously except for the period 15$^{th}$ June 1999 to 31$^{st}$ January 2000 when the station was completely rebuilt, and the ozone monitor had to be taken temporarily down to Gruvebadet.

Standard UV monitors have been used since the start in 1989. The instruments have been replaced by new monitors at various times and since 1997 each monitor shift has been done according to a quality assured procedure including pre- and post-calibrations and intercomparisons. The very first monitor replacement was done in September 1994 (though there is no available documentation of the QA procedures for that instrument shift). According to the logbook in 1994, the monitor was brought to NILU's laboratory for inspection because the monitor was unstable, and it was replaced by a new monitor. Thus, the data from the last period before the replacement in 1994 are more uncertain.





The world calibration centre for surface ozone (WCC-EMPA) has carried out audits of the Zeppelin Observatory in 1997, 2001, 2005 and 2012 and all audits concluded that the on-site ozone monitor provided good and adequate results when compared with WCC-Empa's travelling standard that in turn are traceable to a Standard Reference Photometer

(https://www.empa.ch/web/s503/wcc-empa). In the first audit in 1997, it was commented that for very low ozone levels (< 20 ppb) the instrument was outside tolerance. Such low levels occur only during certain episodes in spring in connection with low ozone episodes (LOEs) linked to rapid destruction of ozone by halogen radicals over the Arctic Ocean as discussed in more detail in Sect 5.7.

**4.8 Reactive nitrogen**

Reactive nitrogen species, PAN, peroxypropionyl nitrate (PPN), $NO_x$ ($NO+NO_2$), NOY, and the $NO_2$ photolysis rate $J_{NO2}$ were measured at the Zeppelin Observatory in 1994 to 97 (Beine et al., 1996;Krognes and Beine, 1997;Beine et al., 1997;Beine et al., 1999;Beine and Krognes, 2000;Solberg et al., 1997). Together with measurements of light hydrocarbons, carbonyls, and surface ozone, this constituted a rather unique suite of observational data for an Arctic location at that time and was used to evaluate atmospheric chemistry in detail.

NO and $NO_2$ were measured separately, using a high-sensitivity chemiluminescence detector with a $3\sigma$ detection limit of 0.9 ppt and 2.6 ppt at 1 h average for NO and $NO_2$, respectively (Beine et al., 1996;Beine et al., 1997). $NO_2$ was measured as NO following broad band photolysis by a xenon arc lamp between 350 to 410 nm. Measurements of $NO_Y$ were made with a 'Correct Analysis of $NO_x$' (CRANOX) instrument consisting of a gold converter coupled to a chemiluminescent NO analyser (TECAN CLD 770). $NO_Y$ was converted to NO by a converter constructed at the UEA (University of East Anglia). After

conversion from $NO_Y$, the NO was measured by the chemiluminescence produced during reaction of NO and $O_3$. The CLD had a $2\sigma$ detection limit of 50 ppt and was calibrated on a weekly basis. More details could be found in Solberg et al. (1997).

To support research into chemistry of reactive nitrogen compounds in the arctic, PAN and PPN were measured at the Zeppelin Observatory in 1994 to 1996 using gas chromatography with an electron capture detector (GC-ECD, 10 mCi Ni-63 electron source and packed column, Carbowax 400 on a Chromosorb W-HP support). The instrument sampled automatically every 15

minutes, and results were calibrated and aggregated to 1-day averages. Calibration of the GC-ECD was based on a liquid standard of PAN in hexane. NILU initiated and coordinated an extensive project for interlaboratory comparison of calibration of liquid PAN standards (Krognes et al., 1996). The calibrated standard was transported to the observatory packed in dry ice and stored in a normal freezer at the site. 10 L of pure synthetic air is filled into a Tedlar bag, and 5 µL of the standard solution (nominally 10 or 100 µg mL$^{-1}$ PAN in hexane). The instrument sampled from the bag (in its normal 15-minute cycle) for

approximately two hours. Due to thermal decomposition in the bag at room temperature, the concentration of PAN decays quickly over this period. This decay was plotted and extrapolate back to the time of the standard injection, to find the instrument response to the known initial concentration in the bag. The resulting scaling factor covers the detector response and the systematic loss in the separation column (due to adsorption and thermal decomposition). Despite the complex process and the numerous error sources, the calibration factor was found to be constant over the 3-year campaign period.



The entire data set has ≈100000 chromatograms, all initially interpreted automatically by HP Chemstation software, then inspected manually to discard outliers and correct peak detections and baselines where appropriate. The practical detection limit was of the order of 10 pptv for individual samples. During summer, the PPN concentrations were close to this detection limit. Peaks were visible, but the percentage of good samples fell below a quality control criterium of 50%, and no concentration could be reported (Krognes et al., 1996;Beine and Krognes, 2000).

## 5 Results and discussion

### 5.1 Aerosol chemical composition

Time series of OC and EC in 2019 resemble each other in winter, spring, and late fall, suggesting similar sources/source regions and/or thorough mixing during long-range atmospheric transport (Fig. 6). The evolution of the time series in winter and spring is like that of other long-range transported aerosol species, such as $SO_4^{2-}$, and is a part of Arctic haze. In summer, the OC level is equally high and occasionally higher than the level observed during Arctic haze, whereas the EC level decrease substantially. Annual concentrations of OC (75 ng C m$^{-3}$) and EC (12 ng C m$^{-3}$, year 2019) at the Zeppelin Observatory are 8 to 9 times lower than on the Norwegian mainland, where levels are the lowest in regional background Europe (Yttri et al., 2007). By accounting for positive sampling artefacts of OC, an overestimation of approximately 25% is avoided. The resulting OC corrected for positive sampling artefact should be considered a conservative estimate of the OC level at Zeppelin Observatory, as the negative sampling artefact of OC is not accounted for.

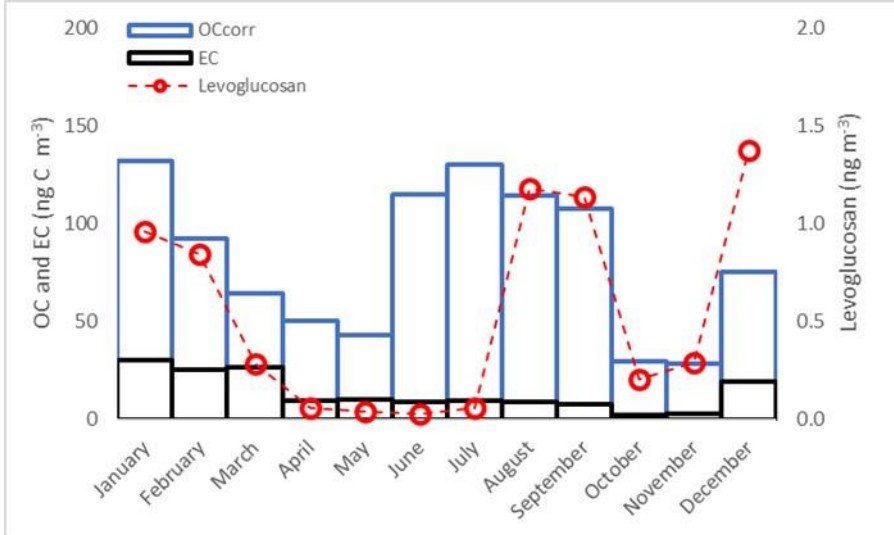

**Figure 6: Monthly means of elemental carbon, organic carbon corrected for the positive sampling artefact, and levoglucosan in the PM$_{10}$ size fraction at Zeppelin in 2019.**

The EC levels and fractions are higher in winter, spring, and late fall (heating season) due to increased emissions from combustion of fossil fuel and biomass for heating purposes (Yttri et al., 2014) and lower biogenic emissions. Increased OC





levels in summer align with those seen for areas in Scandinavia with little anthropogenic influence and where increased levels are explained by biogenic sources, i.e., biogenic secondary organic aerosol (BSOA) and primary biological aerosol particles (PBAP). Increased BSOA tracer (2-methyltetrols) levels are seen in the first part of summer at Zeppelin Observatory, whereas PBAP tracers (e.g. arabitol and mannitol) are more abundant in late summer and early fall (not shown). Levoglucosan is

elevated both in the heating season and in late summer/early autumn (Fig. 6), mirroring increased emissions from residential woodburning in the heating season and boreal wildfires in the heating season (Stohl et al., 2007a;Yttri et al., 2014). Hence, a mixture of both primary and secondary carbonaceous aerosol from natural sources explains the increased level observed for June to October, although the classification of wildfires as a natural source can be questioned, as anthropogenic activity explain the majority of cases in Europe when natural vegetation ignites (Winiwarter et al., 1999).

A conversion factor (CF) of 1.9 to 2.2 is suggested for conversion of OC to organic matter (OM) for the aged aerosol (Turpin and Lim, 2001), which complies well with a remote site like the Zeppelin Observatory where most aerosol particles are long-range transported. An annual mean OM concentration of 165 ng m$^{-3}$ (CF =2.2) is noticeably less than for other aerosol species from natural sources such as sea salt aerosol (749 ng m$^{-3}$) and mineral dust (525 ng m$^{-3}$), and even non-sea salt SO$_4^{2-}$ (270 ng m$^{-3}$, Fig. 2).

Annual mean concentrations of inorganic ions such SO$_4^{2-}$ are generally lower than the levels on the Norwegian mainland, e.g., at Birkenes in South Norway (Aas et al., 2019), a site with some of the lowest levels of particulate matter in Europe, reflecting the remote location of the Zeppelin Observatory. However, the observatory frequently experiences individual sulfate pollution episodes exceeding those seen on the mainland in a given year, e.g., an episode with a concentration of 5.1 µg m$^{-3}$ in 2018 (Aas et al., 2020), while for some years mean sulfur dioxide (SO$_2$) is actually higher than on the Norwegian mainland. These

episodes of sulfur pollution occur due to the arrival of air masses from Russia (Aas et al., 2020).

Determination of background trends in inorganic ions at Zeppelin is a crucial component of the Norwegian environmental monitoring programme, used to measure the effectiveness of the 1999 Gothenburg Protocol (GP) to abate acidification. The objective of the original 1999 GP was to reduce European emissions of sulfur by 63% in 2010 compared to 1990, and nitrogen oxides and ammonia by 41% and 17%, respectively. In 2012, the protocol was revised, with new emissions targets for 2020

with 2005 as the base year. The current abatement targets for inorganic atmospheric species in the European Union, with Norwegian targets in parenthesis, are SO$_2$: 59% (10%), NO$_x$: 43% (23%), ammonia: 6% (8%). SO$_2$ and sea salt corrected SO$_4^{2-}$ at Zeppelin have decreased by 75% and 44%, respectively between 1990 and 2019. This is significantly lower than the reductions seen on the Norwegian mainland, e.g. SO$_2$: 95%, sea salt corrected SO$_4^{2-}$: 74% at Birkenes, South Norway (Aas et al., 2020), likely reflecting varying source regions or changes in transport patterns. No significant trend is observed for the

2005 to 2019 period using the Mann Kendall statistic (Aas et al., 2020). Nevertheless, the decreasing levels of sulfur species at the Zeppelin Observatory reflect the success of the Gothenburg Protocol.





## 5.2 Aerosol physical properties

Most previous studies on CCN and cloud properties in polar regions are based on short term campaigns, carried out predominantly in polar day season. Only a handful of studies cover seasonal cycles and interannual variability e.g. (Jung et al.,
2018), allowing longer term observations of the natural variability and trends. The change in long-range transport of aerosols vary strongly during the year due to change in radiation flux in the northern hemisphere, the movement of the polar front and with that seasonal variability in the general circulation. The long-range transport of aerosols primarily from the Eurasian continent is slowly increasing during the fall and winter to a major peak during March to May which is observed as very high concentrations of aged particles (See Fig. 7).

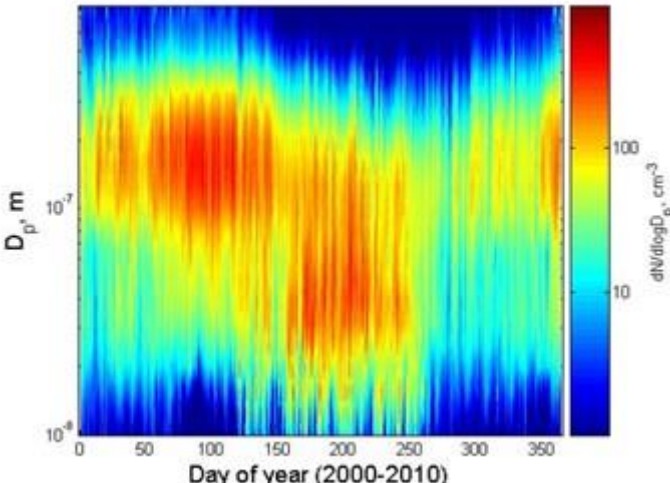


**Figure 7: Spectral plot of daily average aerosol number size distributions, March 2000 to December 2010. Units on x-axis as day of year.**

With a strong decrease in long-range transport related events, the aerosol population changes drastically during the summer. The particles are an order of magnitude smaller in size but appear in similar number concentrations. Particle nucleation events
followed by particle growth are observed frequently during the light period and dominate the particle concentration during the summer (see Fig. 8 and Tunved et al., 2013).

When converting number into mass, the same pattern occurs as is observed in the aerosol chemistry measurements. The winter period is strongly dominated by anthropogenic long distant transported aerosol, which is followed by the Arctic haze period, while the summer observations show very low atmospheric concentrations mostly influenced by natural sources related to
gaseous emission from the sea and local sources.


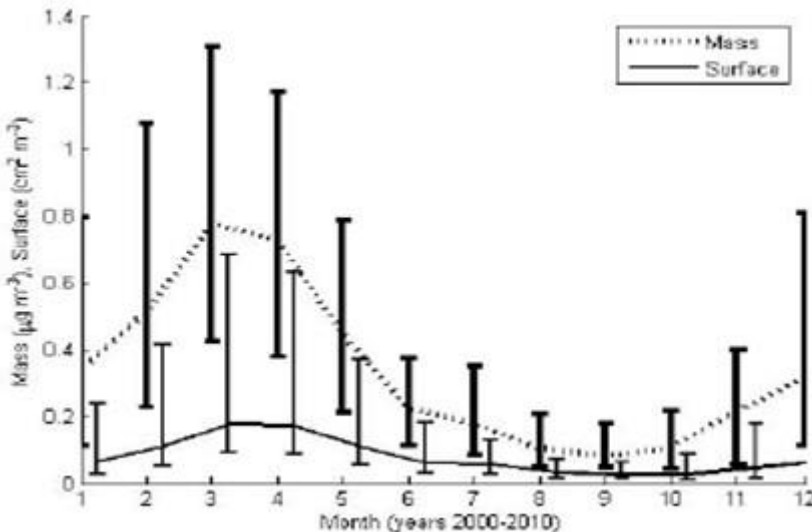

**Figure 8: Annual average variation of integrated surface and mass, March 2000 to March 2010. Mass data calculated from aerosol number size distribution assuming a density of 1 g cm$^{-3}$. Error bars show 25 to 75th percentile ranges.**

Long-term observations at high time resolution provide the opportunity to study the influence of different atmospheric conditions, e.g., precipitation and wet deposition, during transport. In an investigation of 10 years of data, Tunved et al. (2013) demonstrated a strong dependence on precipitation during the dark winter period. Strong decreases in accumulation mode particles were seen with just a few mm of precipitation slowly leveling off with larger precipitation amounts (> 40 mm) eventually almost obliterating the aerosol (See Fig. 9, left). During the sunlit period, the first decrease in accumulation mode

particles is followed by a strong increase in Aitken mode particles explained by new particle formation followed by subsequent growth (Fig. 9, right). These observations show the influence of different processes and their dependence on sunlight and precipitation.

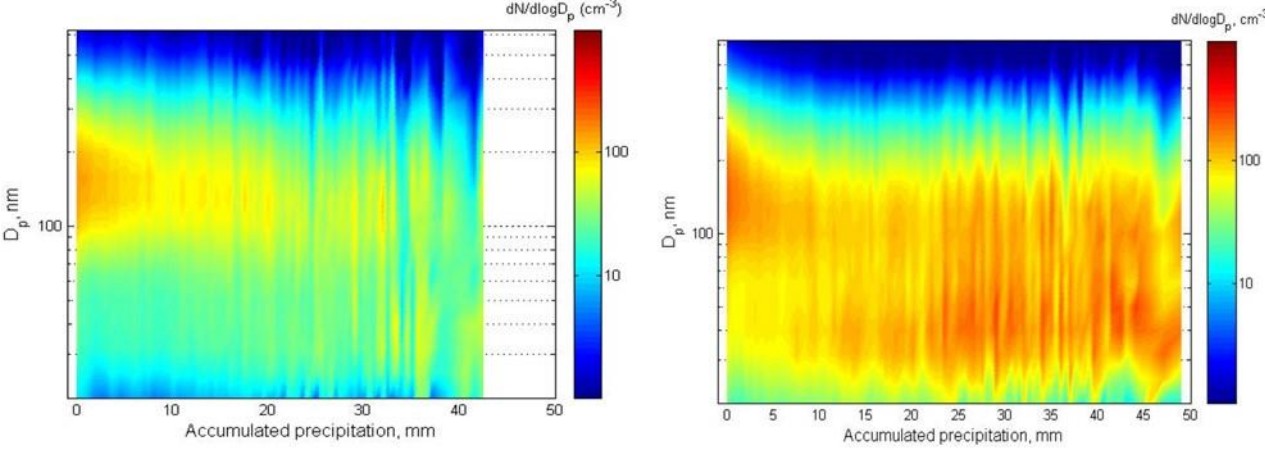





**Figure 9: Evolution of aerosol number size distribution (dN/ dlog D$_p$) as a function of accumulated precipitation along 240 h**
**trajectories. Left: for the dark period (October–February, right for the sunlit period (March to September). All data from 2000 to**
**2010.**

Long-term support and excellent facilities combined with very high-quality routines and standards are needed to establish long

term trends. 20 years of light scattering data from nephelometer measurements at Zeppelin Observatory facilitate determination

of trends of light scattering properties (see Fig. 10 and Heslin-Rees et al. 2020). All optical and physical properties are

measured a low relative humidity (RH) conditions to keep aerosol observations comparable. However, one should keep in

mind that the ambient light scattering coefficients are larger due to the hygroscopicity of aerosol particles and successive

water-uptake at elevated RH. The so-called 'light scattering enhancement' is much more pronounced in the Arctic compared

to more continental sites (Zieger et al., 2010;Zieger et al., 2013). An increase in both scattering and backscattering indicates

an increase of particle concentrations or an increase in particle size which is supported by a decreasing Ångstrøm exponent,

showing a shift to larger particles in the particle size distribution. The increase in particle concentration is seen through the

year, most likely corresponding to an increased contribution from larger particles. High time resolution data, show that

scattering depends on transport over open water, which indicates a larger contribution from sea spray. (Heslin-Rees et al.,

2020) argue the observed long-term changes are due to changes in circulation, i.e., increased frequency of long-range transport

from the open northern Atlantic. However, new particle formation (NPF) events at Zeppelin Observatory have been shown to

be anti-correlated with sea ice extent indicating a dependence on more open sea (Dall'Osto et al., 2017).

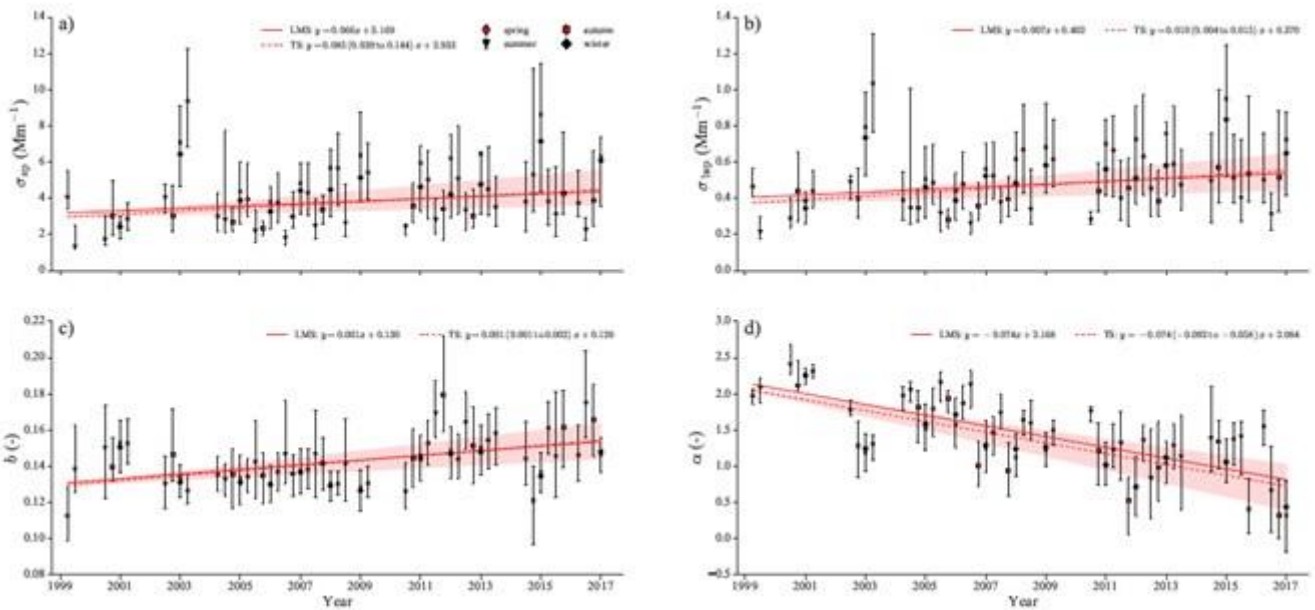

**Figure 10: Long-term trends of the seasonal medians for a) the particle light scattering coefficient (wavelength, λ = 550 nm) b) the**
**particle light backscattering coefficient (λ = 550 nm) c) the hemispheric backscattering fraction (λ = 550 nm) d) the scattering**
**Ångström exponent (λ1 = 450 nm,_ λ2 = 550 nm). The seasonal medians are denoted by their respective symbols. The error bars**
**denote the length of the 25th and 75th percentile values. The seasonal mean is given by the cross. The solid and dashed red lines**





**represent the least mean square (LMS) and Theil-Sen slope (TS) of the seasonal medians, respectively. The red shaded area denotes the associated 90% confidence interval of the TS slope. Fig. taken from Heslin-Rees et al. (2020).**

The observed physical properties together with the main chemistry so far indicate a decrease of long-range transport during the winter and an increasing contribution of sea spray. Naturally driven NPF dominates the summertime Arctic atmospheric

aerosol, even though the detailed physiochemical process pathway is not known, but object to ongoing research. Furthermore, it remains difficult to distinguish the direct influence of ongoing climate change as well as the related strong changes in sea ice. As an example of the changes occurring in wider the region, according to Pavlova et al. (2019) et al., based on long term monitoring of sea ice in Kongsfjorden initiated by NPI in 2003, over the last decade only the northern part of inner Kongsfjorden freezes whereas before 2006 sea ice usually extended into the central fjord.

The atmospheric particle life cycle is directly linked with the life cycle of clouds. The aerosol is modified in number and chemistry by cycling through the clouds and the cloud droplet number and cloud radiative properties depend on the aerosol size and chemistry. Changes in sources, i.e., number, size and chemistry may have a significant influence on the radiation balance and thus on how the Arctic climate develops.

**5.3 Climate gases**

The atmospheric mixing ratios of $CO_2$ and $CH_4$, the two most important anthropogenic greenhouse gases are shown in Fig. 11. In 2019 the annual average $CO_2$ mixing ratio at the Zeppelin Observatory was 411.9 ppm, an increase of 2.46 ppm compared to the previous year, close to the long-term trend of 2.5 ppm per year (Table 5) and ≈15% since 1989 (357 ppm). It should be noted that the growth rate is exponential, highlighting the challenge in meeting emissions reductions needed to meet the Paris

Agreement goal of keeping the global annual average temperature increases below 2°C. The $CO_2$ mixing ratio at the Zeppelin Observatory is slightly higher than the global average mixing ratio, e.g., 409.3 ppm at the Zeppelin Observatory vs. 407.8 ppm globally in 2018 (WMO, 2020), since northern hemisphere $CO_2$ emissions are higher. However, $CO_2$ at Zeppelin is lower than observed at more continental sites such as Birkenes, Southern Norway (416.1 ppm, Myhre et al., 2020).





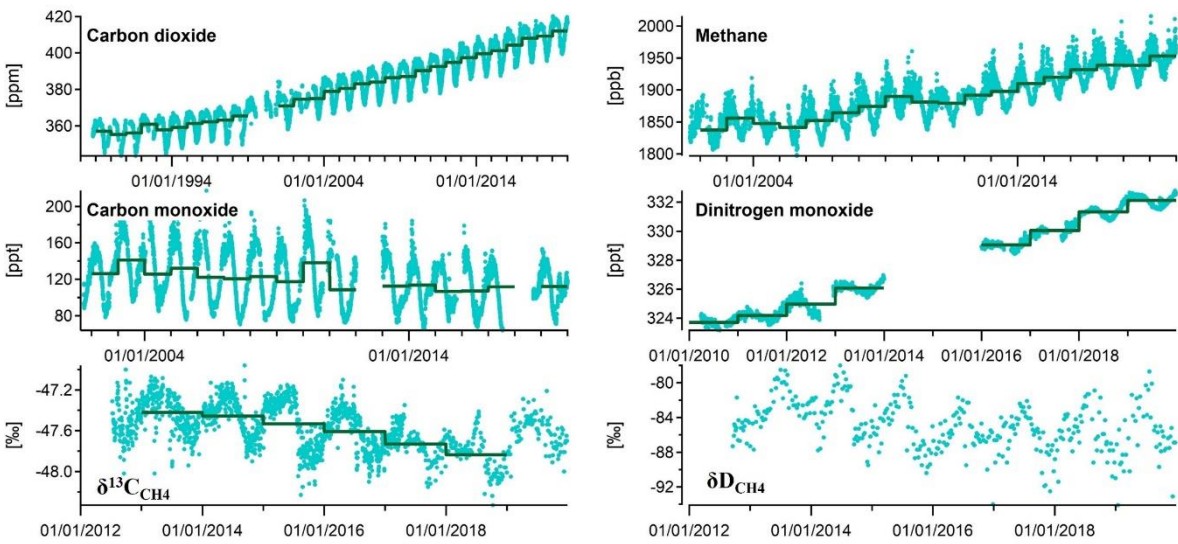

**Figure 11: Daily (markers) and annual (solid lines) atmospheric mixing ratios of carbon dioxide ($CO_2$), methane ($CH_4$), carbon monoxide (CO), and dinitrogen monoxide ($N_2O$), and isotopic shifts in the carbon and deuterium in methane of carbon 13($\delta^{13}C_{CH4}$) and deuterium ($\delta D_{CH4}$) at the Zeppelin Observatory. Daily and annual means calculated only where data coverage is ≥75% of total day or year, respectively.**

**Table 5: Selected greenhouse measured at the Zeppelin Observatory, their chemical formulas, global warming potentials (GWP, $CO_2$=1), mean mixing ratios in 2019, fitted trends and trend fit parameters (error and $R^2$). measured from 2001.**

| Compound | Chemical formula | GWP[1] | Mean mixing ratio[2] [ppb] | Trend[2,3] [ppb yr⁻¹] |
|---|---|---|---|---|
| Carbon dioxide | $CO_2$ | 1 | $411.9\times10^3$ | $2.5\times10^3$ |
| Methane | $CH_4$ | 32 | 1952.9 | 14 |
| Carbon monoxide | CO | 115 | 115.4 | -1.2 |
| Dinitrogen monoxide | $N_2O$ | 332 | 332.1 | 0.98 |

[1]Global warming potentials for a 100 year time horizon according to Montzka et al. (2011) and (Hodnebrog et al., 2013), where available

[2]From (Myhre et al., 2020)

[3]Following (Simmonds et al., 2006):

$$f(t) = a + b \cdot N \cdot P_1\left(\frac{t}{N}-1\right) + \frac{1}{3} \cdot d \cdot N^2 \cdot P_2\left(\frac{t}{N}-1\right) + \frac{1}{3} \cdot e \cdot N^3 \cdot P_3\left(\frac{t}{N}-1\right) + c_1 \cdot cos(2\pi t) + s_1 \cdot sin(2\pi t),$$

where *f(t) is* the change in atmospheric mixing ratio of a species as a function of time over $N$ months, $a$, $b$, $d$, and $e$ are fit parameters with $a$ defining the average mole fraction, $b$ defining the trend in the mole fraction and $d$ defining the acceleration in the trend. Coefficients $c_1$ and $s_1$ define the annual cycles in the mole fraction and $P_i$ are the Legendre polynomials of order $i$.

As well as the annual variation in $CO_2$ (a result of the larger landmass vegetation in the Northern hemisphere), short term inter-annual variations arise due to variability in emissions and sink strengths caused by anthropogenic activity and plant $CO_2$ uptake and release which are a function of numerous climatic factors. Local influence is minimal (Sect. 3.2). In 2017 to 2018 the annual increase in $CO_2$ at the Zeppelin Observatory was as low as 1.6 ppm compared to 4.1 ppm at Birkenes, as the Zeppelin





Observatory received above normal transport of air masses from the North Atlantic and within the Arctic (Myhre et al., 2020). The standard deviation in $CO_2$ increases at Zeppelin for the last 5 years was 0.89 ppm, or 36% of the 2018 to 2019 increase,

enough to obscure even moderate short-term changes in anthropogenic $CO_2$ emissions. For example, the expected drop in anthropogenic $CO_2$ due to the COVID-19 pandemic related lock-down measures in 2020 of around 4 to 7% (Le Quéré et al., 2020) is below the level of natural inter-annual variability in the $CO_2$ trend.

The $CH_4$ mixing ratio is also clearly increasing over time. After a brief pause, since 2005 daily mean $CH_4$ mixing ratios at Zeppelin increased by an average $5.9 \pm 0.3$ ppb yr$^{-1}$ (Platt et al., 2018), and by 14 ppb for 2018 to 2019, reaching a record level

of 1952.9 ppb. For comparison, the global mean $CH_4$ mixing ratio in 2019 was 1869 ppb (WMO, 2020), reflecting considerable latitudinal variation in $CH_4$ due to the uneven spatial distribution of sink strength and sources and its relatively short lifetime (approximately 11 years).

The resumption of an increasing $CH_4$ trend in ~2005 was seen globally and was unexpected and threatens to move the Paris Agreement 2ºC goal out of reach by increasing the overall need for abatements via internationally determined contributions,

which had not previously assumed an increasing global $CH_4$ mixing ratio (Nisbet et al., 2019). Furthermore, the GWP for $CH_4$ has been revised upwards from 28 to 32 (Etminan et al., 2016), i.e. a 25% stronger forcing.

At the same time as the mixing ratio has increased, $\delta^{13}C_{CH4}$ has shifted, by about $-0.03‰$ yr$^{-1}$ (Fig. 11), suggesting a change in the balance of the sources and sinks of methane. Due to the Zeppelin Observatory's remote location, $\delta^{13}C_{CH4}$ is only minimally perturbed by anthropogenic emissions: Thonat et al. (2019) report synoptic changes up to $-0.2$ ‰ in $\delta^{13}C_{CH4}$ due to

wetland influences, an order of magnitude higher than anthropogenic emissions (<0.02 ‰, excepting some long-range transport episodes), the influence of which are diminished by biomass burning (≈+0.01 ‰) and also by the fractionating effects of the two major sinks (≈ +0.01 ‰). Therefore, the location of the Zeppelin Observatory in principal allows the study of emissions from vulnerable (climate sensitive) hydrocarbon $CH_4$ reservoirs in the Arctic including thawing terrestrial and subsea permafrost, seabed cold seepage (e.g. fuelled by decomposing gas hydrates, GH), as well as from biomass burning, since the

potential for synoptic variations due to localised bio/geogenic emissions is relatively higher than for other sites.

Based on $\delta^{13}C_{CH4}$, Nisbet et al. (2019) suggest that ruminant and/or mid-latitude wetland emissions are largely responsible for the increased $CH_4$ levels since 2007, since they are strongly negative compared to the ambient value ($\delta^{13}C_{CH4}$= 56.7‰ for both, compared to $-47‰$ to $-53‰$ for fossil fuels (France et al., 2016)), while increases from wetland emissions are consistent with atmospheric inversion modelling. However, there are other changes in the $CH_4$ budget which may explain the isotopic shift,

and Nisbet et al. (2019) also suggest that the ongoing increase in $CH_4$ and negative shift in ambient $\delta^{13}C_{CH4}$, is compatible with four non-mutually exclusive hypotheses: 1) Increases in very negative biogenic emissions (e.g. wetlands or ruminants); 2) Increased fossil fuel emissions accompanied by a negative shift in their mean $\delta^{13}C_{CH4}$ (although depending on changes in other sources a shift in their $\delta^{13}C_{CH4}$ is not necessary); 3) Changes in the removal rate via reaction with OH; 4) Decreases in biomass burning, combined with increases in both fossil and biogenic emission of roughly equal magnitude as suggested by

(Worden et al., 2017) and independently from budget considerations by Jackson et al., 2020. Hypothesis 3 is still an unlikely



candidate to explain the isotopic shift, though recent work shows that some influence is possible since there are indications that ambient CO/ $NO_X$ is affecting the OH sink (Dalsøren et al., 2016).

Thompson et al. (2017) included methane observations from the Zeppelin Observatory in a high latitude (>50 N) inversion, finding posterior emissions generally both higher and more variable than prior estimates from inventories. The main increase

in $CH_4$ emission compared to prior emissions was in Western Siberian wetlands, with a top down flux pf 19.3 to 19.9 Tg yr$^{-1}$ compared to e.g. only 4.9 Tg yr$^{-1}$ from the LPX-Bern bottom-up inventory (Stocker et al., 2014). A large, anomalous increase was seen for Western Siberian $CH_4$ in 2007, linked to high temperatures in the same year. This underscores the potential role of high latitude wetlands as a climate feedback. Note also that wetland emissions likely include a significant fraction of permafrost $CH_4$, due to co-location, and indirect effects including increased leaching of organic carbon into soil and changes

in hydrology.

In 1987, the Montreal protocol was signed with the aim of stopping emissions of stratospheric ozone depleting substances, at that time mainly chlorofluorocarbons (CFCs), by improving technology and developing replacement compounds with lower ozone depleting potential. The main sources of these compounds were related to foam blowing, aerosol propellants, refrigeration, solvents, and the electronics industry. The largest production of the CFCs was around 1985 and maximum

emissions were around 1987. The first-generation substitutes to CFCs included the hydro chlorofluorocarbons (HCFCs), also included in the Montreal Protocol, followed by the hydrofluorocarbons (HFCs).

Halogenated hydrocarbons have been measured at the Zeppelin Observatory since 2001 as part of the AGAGE program (Table 3). Fig. 12 shows the concentrations of the CFCs, HCFCs, and HFCs measured there. The trends for most major CFCs are all negative e.g. -1.79, -2.51, and -0.64 ppt yr$^{-1}$ for CFC-11, CFC-12 and CFC-113, respectively, see **Error! Not a valid**

**bookmark self-reference.** 6. For CFC-115 the trend is still slightly positive, +0.02 ppt yr$^{-1}$, likely a consequence of its extremely long atmospheric lifetime (> 1000 years). Meanwhile, Fig. 12 also shows that the mixing ratios of the HCFCs, now almost phased out, have either peaked or the growth rate is slowing. The 2016 Kigali amendment to the Montreal Protocol aims to phase out the HFCs, though this is too recent to impact the levels seen at the Zeppelin Observatory. Mixing ratios of many HFCs are increasing rapidly (Fig. 12).





**Figure 12:** Daily (dots) and yearly (solid lines) mean halogenated compounds measured at the Zeppelin Observatory with the online adsorption-desorption system gas chromatograph with mass spectrometry with flame ionization detector (ADS-GCMS-FID, green, 2001-2011) and the Medusa GCMS (blue, for 2010 to 2019). Note the higher variability for the ADS-GCMS (Many compounds including CFCs did not meet AGAGE precision requirements, see Sect. 4.3). Daily and annual means calculated only where data coverage is ≥75% of total day or year, respectively. See also Table 6 for information on chemical formulas and compound names.





**Table 6: Halogenated compounds measured at the Zeppelin Observatory, their chemical formulas, global warming potentials (GWP, CO₂=1), mean mixing ratios in 2019, fitted trends and trend fit parameters (error and R²). For compounds measured only from 2010 (see also Table 3), the uncertainty in the trend is higher than for compounds measured from 2001.**

| Compound | Chemical formula | GWP[1] | Mean mixing ratio[2] [ppt] | Trend[2,3] [ppt yr⁻¹] | error | r2 |
|---|---|---|---|---|---|---|
| CFC-11 | $CCl_3F$ | 4660 | 228.1 | -1.79 | 0.008 | 0.99 |
| CFC-113 | $Cl_2FC\text{-}CClF_2$ | 13900 | 70.1 | -0.64 | 0.002 | 0.99 |
| CFC-115 | $ClF_2C\text{-}CF_3$ | 7670 | 8.75 | 0.02 | 0.001 | 0.73 |
| CFC-12 | $CHCl_2F_2$ | 10200 | 505.1 | -2.51 | 0.025 | 0.98 |
| H-1211 | $CBrClF_2$ | 1750 | 3.37 | -0.065 | 0.0003 | 0.995 |
| H-1301 | $CBrF_3$ | 7800 | 3.39 | 0.02 | 0.0004 | 0.776 |
| H-2402 | $CBrF_2CBrF_2$ | 1470 | 0.41 | -0.007 | 0.0001 | 0.961 |
| HCFC-141b | $C_2H_3FCl_2$ | 782 | 25.7 | 0.53 | 0.02 | 0.971 |
| HCFC-142b | $CH_3CF_2Cl$ | 1980 | 23.2 | 0.54 | 0.011 | 0.987 |
| HCFC-22 | $HCF_2Cl$ | 1760 | 255.7 | 5.81 | 0.031 | 0.997 |
| HFC-125 | $CHF_2CF_3$ | 3170 | 32.3 | 1.65 | 0.006 | 0.999 |
| HFC-134a | $CH2FCF3$ | 1300 | 114.8 | 5.14 | 0.009 | 0.999 |
| HFC-143a | $CH_3CF_3$ | 4800 | 23.9 | 1.53 | 0.004 | 0.997 |
| HFC-152a | $CH_3CHF_2$ | 506 | 10.5 | 0.43 | 0.011 | 0.965 |
| HFC-227ea | $CF_3CHFCF_3$ | 39 | 1.76 | 0.12 | - | 0.998 |
| HFC-23 | $CHF_3$ | 12400 | 33.2 | 1.04 | 0.003 | 0.998 |
| HFC-236fa | $CF_3CH_2CF_3$ | 242 | 0.2 | 0.01 | - | 0.985 |
| HFC-245fa | $CHF_2CH_2CF_3$ | 8 | 3.53 | 0.21 | 0.001 | 0.997 |
| HFC-32 | $CH_2F_2$ | 5 | 25.24 | 2.15 | 0.007 | 0.999 |
| HFC-365mfc | $CH_3CF_2CH_2CF_3$ | 804 | 1.31 | 0.07 | - | 0.981 |
| HFC-4310mee | $C_5H_2F_{10}$ | 1650 | 0.3 | 0.01 | - | 0.936 |
| PFC-116 | $C_2F_6$ | 10000 | 4.91 | 0.089 | 0.0003 | 0.996 |
| PFC-14 | $CF_4$ | 6630 | 86.1 | 0.893 | 0.1109 | 0.995 |
| PFC-218 | $C_3F_8$ | 8900 | 0.69 | 0.014 | 0.0001 | 0.976 |
| PFC-318 | $c\text{-}C_4F_8$ | 9540 | 1.8 | 0.057 | 0.0002 | 0.995 |
| Bromomethane | $CH_3Br$ | 2 | 6.78 | -0.161 | 0.0056 | 0.885 |
| Carbon_tetrachloride | $CCl_4$ | 1730 | 78.02 | -0.954 | 0.0182 | 0.935 |
| Chloromethane | $CH_3Cl$ | 12 | 508 | -0.373 | 0.2145 | 0.871 |
| Dichloromethane | $CH_2Cl_2$ | 9 | 58.89 | 1.927 | 0.0613 | 0.934 |
| Sulfur_hexafluoride | $SF_6$ | 23500 | 10.14 | 0.291 | 0.0004 | 0.999 |
| Sulfuryl fluoride | $SO_2F_2$ | 4090 | 2.53 | 0.102 | 0.0009 | 0.993 |
| Trichloroethane | $CH_3CCl_3$ | 160 | 1.71 | -1.807 | 0.0083 | 0.999 |
| Trichloroethene[4] | $C_2HCl_3$ | | 0.16 | -0.017 | 0.0035 | 0.396 |
| Trichloromethane | $CHCl_3$ | 16 | 12.2 | 0.242 | 0.022 | 0.691 |

[1]Global warming potentials according to Montzka et al. (2011) and (Hodnebrog et al., 2013), where available





---

[2]From (Myhre et al., 2020)

[3]Following (Simmonds et al., 2006):

$$f(t) = a + b \cdot N \cdot P_1\left(\frac{t}{N} - 1\right) + \frac{1}{3} \cdot d \cdot N^2 \cdot P_2\left(\frac{t}{N} - 1\right) + \frac{1}{3} \cdot e \cdot N^3 \cdot P_3\left(\frac{t}{N} - 1\right) + c_1 \cdot cos(2\pi t) + s_1 \cdot sin(2\pi t),$$

where $f(t)$ is the change in atmospheric mixing ratio of a species as a function of time over $N$ months, $a$, $b$, $d$, and $e$ are fit parameters with $a$ defining the average mole fraction, $b$ defining the trend in the mole fraction and $d$ defining the acceleration in the trend. Coefficients $c_1$ and $s_1$ define the annual cycles in the mole fraction and $P_i$ are the Legendre polynomials of order $i$.

[4]Larger uncertainties due to low concentrations and instrument memory effects

[5]Larger uncertainties for 2001-2010 from the ADS-GCMS instrument

Trends for the CFCs and HCFCs at the Zeppelin Observatory demonstrate the remarkable success of the Montreal Protocol. Not only are these compounds destructive to the stratospheric ozone layer, they are also potent greenhouse gases, strongly absorbing infrared radiation in the part of the spectrum where other GHGs have only low absorption (the so-called 'atmospheric window') and with very long atmospheric lifetimes, up to thousands of years. CFC-12,-13, and, HFC-23, and HFC-12 have global warming potentials more than 10000 times higher than $CO_2$ (Halogenated hydrocarbons have been measured at the Zeppelin Observatory since 2001 as part of the AGAGE program (Table 3). Fig. 12 shows the concentrations of the CFCs, HCFCs, and HFCs measured there. The trends for most major CFCs are all negative e.g. -1.79, -2.51, and -0.64 ppt yr⁻¹ for CFC-11, CFC-12 and CFC-113, respectively, see **Error! Not a valid bookmark self-reference.** 6. For CFC-115 the trend is still slightly positive, +0.02 ppt yr⁻¹, likely a consequence of its extremely long atmospheric lifetime (> 1000 years). Meanwhile, Fig. 12 also shows that the mixing ratios of the HCFCs, now almost phased out, have either peaked or the growth rate is slowing. The 2016 Kigali amendment to the Montreal Protocol aims to phase out the HFCs, though this is too recent to impact the levels seen at the Zeppelin Observatory. Mixing ratios of many HFCs are increasing rapidly (Fig. 12).

**Figure 12: Daily (dots) and yearly (solid lines) mean halogenated compounds measured at the Zeppelin Observatory with the online adsorption-desorption system gas chromatograph with mass spectrometry with flame ionization detector (ADS-GCMS-FID, green, 2001-2011) and the Medusa GCMS (blue, for 2010 to 2019). Note the higher variability for the ADS-GCMS (Many compounds including CFCs did not meet AGAGE precision requirements, see Sect. 4.3). Daily and annual means calculated only where data coverage is ≥75% of total day or year, respectively. See also Table 6 for information on chemical formulas and compound names.**





). Thus, according to Goyal et al. (2019) measures implemented under the Montreal Protocol will have avoided 3 to 4 °C Arctic warming and ≈1 °C global average warming by 2050 (a ~25% mitigation, the most successful abatement so far implemented). However, given the high GWPs of these compounds it is crucial to monitor changes in levels over the coming decades.

**5.4 Non-methane hydrocarbons**

On average, the seasonal cycle in the sum of the NMHC peaks in late winter at a level comparable to the levels in Southern Scandinavia (Solberg et al., 1996a). Alkenes (ethene and propene) are an exception to this and with a less pronounced seasonal cycle indicative of more nearby emissions of these species in summer, presumably linked to natural releases from biogenic activity in the oceans in particular at the sea ice edge (Solberg et al., 1996a). Meanwhile, carbonyls were measured only in 1994, 1995, 1996, and 1998 and only during parts of the year, so it is harder to evaluate the seasonal cycle with confidence. The data indicated either small variation through the year or a peak in May for some species. The ratio of the carbon-based sum of carbonyls to the carbon-based sum of NMHC indicated peak values of around 50% in summer and minimum values around 10% in March and October while it was not sufficient data in winter to calculate this ratio. This seasonal pattern in the carbonyl:NMHC ratio agreed with measurement data from the EMEP background stations at the European mainland (Solberg et al., 1996a). For example, a north European campaign measuring NMHC concentrations at five EMEP sites including Ny-Ålesund was carried out during spring of 1993. Decreasing concentrations from March to June were observed at all sites. The highest concentrations of hydrocarbons were found in air masses coming in from the southwest to southeast, indicating long-range transport from continental Europe and the U.K. The measured concentrations were compared with model calculations covering Europe and the agreement indicated that the European VOC emission inventory was quite well estimated (Hov et al., 1997).

Time series of NMHCs as measured on the Zeppelin Observatory during 1989 to 2020 are shown in Fig. 13 linking the grab samples in the 1990s with the continuous monitoring in the 2000s. The grab samples in glass flasks made by NOAA are also included in the figure. Mean NMHC levels in the Arctic (and in Europe) have decreased significantly since 1989 (Fig. 14). The percentage change over the entire period 1989 to 2020 defined as: $100 \times [X_{2020}/X_{1989} - 1]$, where X is the mixing ratio, is given in Table 7. Due to the infrequent sampling, the strong seasonal cycle and the long period of missing data these numbers will be very uncertain but could be taken as an indication of the trends for the different species. This shows highly different trends with no change in ethane as opposed to decreases of the order of 60 to 80 % for *n*-butane, n-pentane and benzene. These differences are seen also if applying a simple TheilSen trend calculation by season as shown in Fig. 14. The TheilSen statistics gives insignificant trends for ethane in all seasons, whereas trends of 1 to 2 %/year, corresponding to approximately 30 to 60% decrease over the entire 1989 to 2019 period depending on season are found for other species. The NMHC profile, i.e., the relative mix of the individual hydrocarbons, is useful for the study of the 'low ozone episodes' (LOEs) as discussed in Sect. 5.7. The change in NMHC profile during LOEs clearly indicated that the influence of other oxidants besides OH, namely halogen species.







**Figure 13: Non-methane hydrocarbon mixing (NMHC) ratios at the Zeppelin Observatory from 'grab samples' (2 to 3 steel flask samples of 20 minutes, per week, shipped and analysed at NILU's laboratory) for 1989-2000 (blue), averaged NMHC from the Medusa GCMS (grey dots) and yearly averaged NMHC from the Medusa GCMS for 2010 to 2019). Daily and annual means calculated only where data coverage is 75% of total day or year, respectively.**


Median ethane at Zeppelin Mtn

Median n-butane at Zeppelin Mtn

**Figure 14: Seasonal Theil-Sen slopes (red) and confidence intervals (red, dashed) based on monthly median concentrations (blue) of ethane and *n*-butane at the Zeppelin Observatory.**



**Table 7: Non-methane hydrocarbons measured at the Zeppelin Observatory, their chemical formulas, atmospheric lifetime, mean mixing ratios in 2019, fitted trends and the estimated percentage change over the entire period 1989-2019 as defined as $100 \times [X_{2019}/X_{1989} - 1]$ where significant.**

| Compound | Chemical formula | Lifetime[1] [~ days] | Mean mixing ratio [ppt] | Trend[2] [ppt yr$^{-1}$] | Total Change [%] |
|---|---|---|---|---|---|
| Benzene | $C_6H_6$ | 17 | 61.11 | -2.7 | -81 |
| Ethane | $C_2H_6$ | 78 | 1602.45 | - | - |
| Propane | $C_3H_8$ | 18 | 454.71 | -1.5 | -45 |
| *n*-Butane | $C_4H_{10}$ | 8 | 140.9 | -2.4 | -71 |
| *n*-Pentane | $C_5H_{12}$ | 5 | 43.77 | -2.2 | -65 |

[1]Lifetimes in approximate (~) days according to (Hewitt, 2000)
[2]Following (Markwardt, 2009) seasonal trends are calculated according to a non-linear least square fit using:
$$c(t) = [a_0 + a_1(\sin(2\pi(t - a_2)))] \cdot \exp[a_3(t - t_0)]$$
where $c(t)$ is the concentration at time $t = 2019$ (in years) and $t_0$=1989. The coefficients $a_0, a_1, a_2, a_3$ represent a simple seasonal cycle with mean concentration $a_0$, amplitude $a_1$, and seasonal phase displacement $a_2$ that change exponentially over time with $a_3$, defining positive ($a_3 > 0$), negative ($a_3 < 0$) growth rate or no trend ($a_3 = 0$).

## 5.5 Persistent organic pollutants

Hung et al. (2016) and (Wong et al., 2021) summarise temporal trends for legacy POPs at the Zeppelin Observatory, and three other AMAP stations; Alert, Canada; Pallas, Finland; and Stórhöfði, Iceland. They show that most POPs listed for control under the Stockholm Convention (SC), e.g., hexachlorohexanes (HCHs), polychlorinated biphenyls (PCBs), dichlorodiphenyltrichloroethanes (DDTs) and chlordanes, were declining slowly at all Arctic sites. The decline was largely suggested to reflect reduced primary emissions during the last two decades and the increasing importance of secondary emissions from environmental reservoirs. Slow declining trends for these POPs signifies their persistence and slow degradation in the Arctic environment, resulting in detectable levels despite being banned for decades in many countries (Ma et al., 2011). However, not all legacy POPs show a steady, continuous decline in air concentrations at Zeppelin over the entire monitoring time period as shown in Fig. 15, A notable example is HCB (hexachlorobenzene) which declined during the 1990's (from an annual mean concentration of 95 pg m$^{-3}$ in the beginning of 1990s to 55 pg m$^{-3}$ in the beginning of 2000), prior to regulation under the Stockholm Convention on POPs, but then started to increase until a few years ago (to an annual mean concentration of 85 pg m$^{-3}$ in 2014 to 2016), i.e. after HCB became regulated under the SC. Two main hypotheses have been put forward to explain this late increase: 1) as a result of increasing primary emissions and 2) enhanced re-volatilisation of HCB from previously contaminated surface reservoirs, potentially modulated by increasing temperatures due to a warming climate (e.g. Ma et al., 2011).





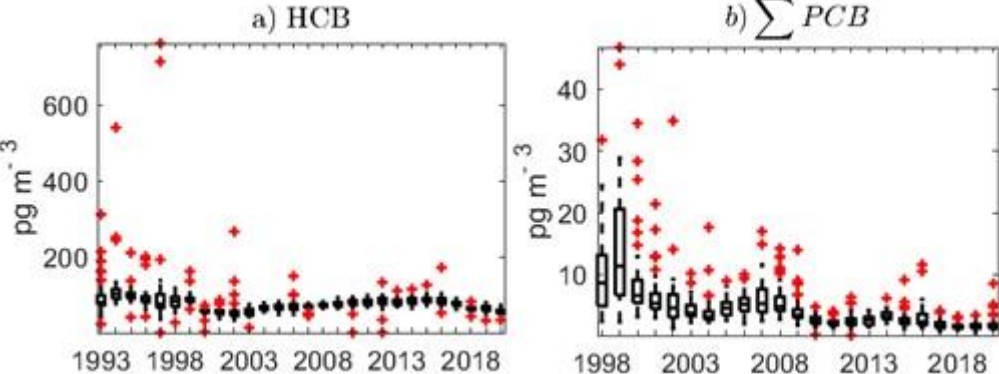

**Figure 15: Box-plot time series (box top and bottom: 75th and 50th quartiles, respectively, horizontal bar: median, outliers: red crosses) of a) Hexachlorobenzene (HCB) from 1993 to 2019, and b) the sum of 7 polychlorinated biphenyls (PCBs -28, 52, 101, 118, 138, 153, 180) from 1998 to 2019, at the Zeppelin Observatory.**

Similar to the analysis for the eBC in Sect. 3.2, we used FLEXPART to estimate footprint emission sensitivities for periods of

945 high and low HCB concentrations (>80th and <20th percentiles of concentration, respectively) in December to March 2014 to

2017, i.e. during the Arctic haze period, and also when HCB was especially elevated compared to other years. The ratio of

each of these footprint sensitivities to the average (Fig. 16) yields a qualitative description of geographic areas linked to high

and low levels of haze-time HCB seen at the Zeppelin Observatory. The patterns are distinctively different. FLEXPART

predicts that haze periods with higher concentrations of HCB in air at Zeppelin are mainly associated with transport of air

masses from Asia and, albeit to a lesser extent, from Greenland and the Arctic ocean north of Mt Zeppelin (Fig. 16a). In sharp

contrast, we attribute periods with lower concentrations of HCB (Fig. 16b) to transport from ocean areas south of Mt. Zeppelin

and the North American continent (e.g., Alaska). Together, these model predictions show that Asian HCB emissions largely

explain the elevated concentrations of HCB observed during December-March at the Zeppelin Observatory. Additionally,

secondary emissions of HCB from ice-covered areas in the high Arctic may, to some extent, have contributed during periods

with elevated concentrations (Fig. 16a) and large ice-free ocean areas are associated with low concentrations, which suggests

that secondary emissions from these regions are of limited significance.



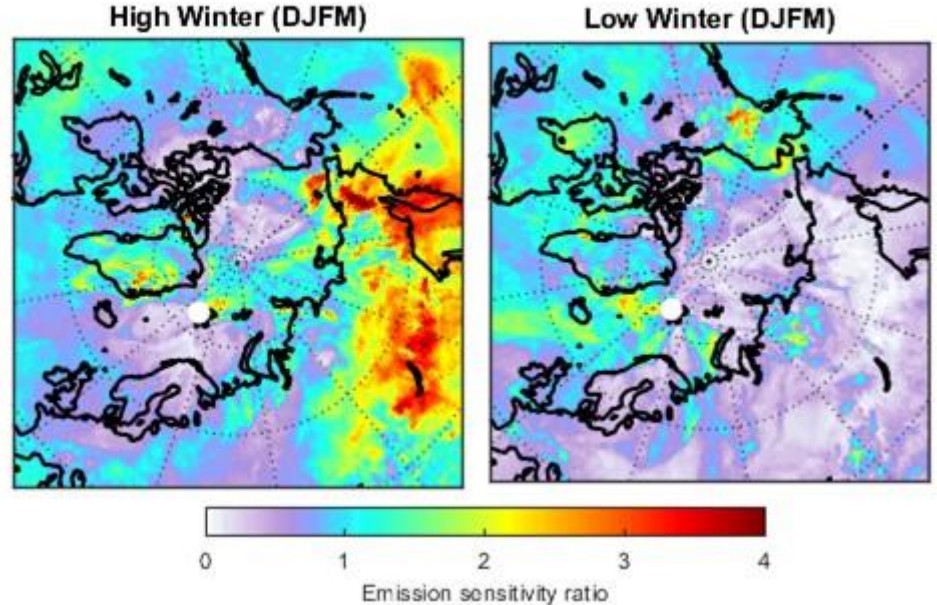

**Figure 16: The ratio of the hexachlorobenzene (HCB) emission sensitivity at Zeppelin Observatory during high (>80th percentile, left) and low (<20th percentile, right) HCB levels compared to average HCB levels, during the Arctic haze period (March to December) in 2014 to 2017. This ratio yields a qualitative description of source regions of HCB (higher emission sensitivity ratio, red, left) and lower HCB emitting regions (higher ratios, red, right), compared to average.**

### 5.6 Heavy metals and mercury

Being elements, metals cannot be broken down into less toxic substances in the environment. Although some metals are essential nutrients at low concentrations, heavy metals can be toxic even in small quantities and are present at high levels in regions remote from most anthropogenic sources, such as the Arctic. Through the 1998 Protocol on Heavy Metals under the UNECE Convention on Long-Range Transboundary Air Pollution (CLRTAP), governments are taking measures to minimize and prevent emissions of Cd, Pb and Hg by regulating their predominant anthropogenic sources; waste incineration, combustion, and industrial processes. According to the European Environment Agency (EEA), emission of these elements are reduced by 55, 87 and 61% respectively since 1994 (https://www.eea.europa.eu/data-and-maps/indicators/eea32-heavy-metal-hm-emissions-1/assessment-10).

A strong seasonal signal is observed for most of the heavy metals with a maximum in winter and minimum in summer (Fig. 17), driven by major weather systems. In winter and spring, a high-pressure system over Siberia pushes the Arctic Front southwards and sensitivity to major polluted areas increases (Fig. 3), including to smelters on the Kola Peninsula (Berg et al., 2004). The signal is most pronounced for the so-called anthropogenic elements Pb, Cd and As, typically associated with long-range transport, and less pronounced for Ni, Cu, Co and Zn, elements with Arctic sources from non-ferrous smelters on the Kola-peninsula (Laing et al., 2014a), and Cr, Mn and V that also have a natural component from soil or sea salt. Similar





seasonality has previously been observed at Zeppelin (Berg et al., 2004), Alert, Canada (Gong and Barrie, 2005), Kevo, Finland (Laing et al., 2014a) and over the Russian Arctic coast (Shevchenko et al., 2003).

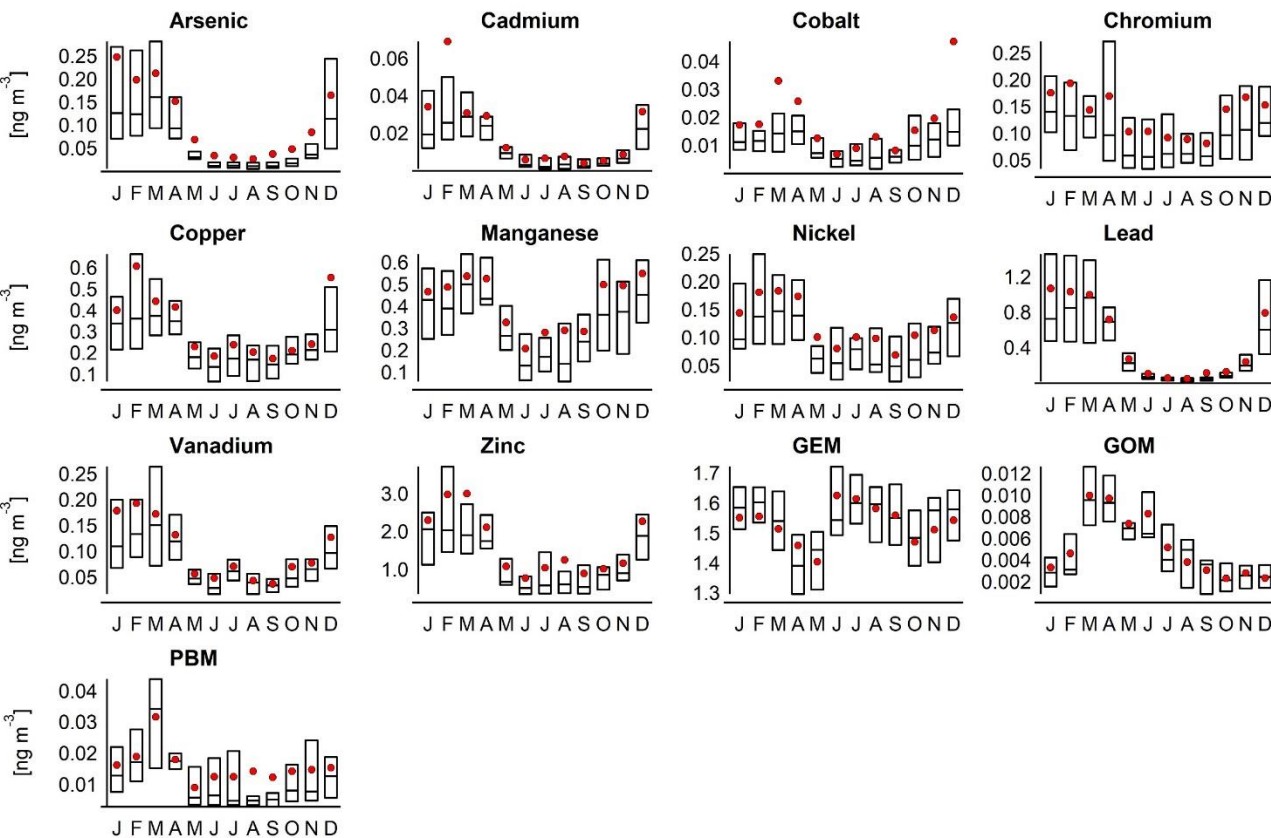

**Figure 17: Boxplots (box top and bottom: 75th and 50th quartiles, respectively, horizontal bar: median) and monthly averages (red) of heavy metals, gaseous elemental mercury (GEM), gaseous organic mercury (GOM), and particle-bound mercury (PBM) concentrations at the Zeppelin Observatory. Heavy metals and GEM from the period 1994 to 2019, and PBM and GOM from 2007 to 2018.**

We estimated the annual trends on a monthly basis (i.e., comparing the same month in consecutive years) using Sen's slope, yielding different magnitudes and signs for the slope (Table 8). Generally, the steepest decreasing trends for most elements are during the winter months, whereas trends are less homogenous during summer months with both increasing and decreasing trends. For Mn increasing trend is observed for all months but December and September.

Decreasing annual trends are observed for As ($-3.8\%$ yr$^{-1}$), Cd ($-2.8\%$ yr$^{-1}$), Cu ($-0.9\%$ yr$^{-1}$), Pb ($-4.6\%$ yr$^{-1}$) and V ($-3.8\%$ yr$^{-1}$), though not significant for Cu. Increasing annual trends are observed for Mn ($1.9\%$ yr$^{-1}$) and Cr ($2.7\%$ yr$^{-1}$), however the trend is significant only for Mn. The annual trend is close to unchanged for Zn ($0.1$ yr$^{-1}$), Co ($0.6\%$ yr$^{-1}$) and Ni ($0.2\%$ yr$^{-1}$). These annual trends are in line with observations from other Arctic long-term sites (Gong and Barrie, 2005;Laing et al., 2014b),





whereas a previous trend study from Zeppelin for the time period 1994-2003 showed no significant trends for any element except Ni (Berg et al., 2004). Interestingly, these upward trends are not seen at Birkenes indicative of a unique Arctic phenomenon.

A study by Weinbruch et al. (2012) examining composition and source of aerosols at Zeppelin found that sea salt, aged sea salt, silicates and mixed particles are the main constituents of particles at Zeppelin. They also found that the fly ash abundance is not correlated with air masses crossing industrialised regions in Central and Eastern Europe, Scandinavia or Russia, indicating a significant reduction of long-range transport of HMs to Svalbard. The HM trends observed are non-monotonic for all elements, and in for Ni and Zn and to some degree Cr, it appears the trend has changed direction and is now increasing.

Though Ni was decreasing for the first 10 years of measurements, Ni concentrations are now even higher than when the measurements were initiated in 1994. According to the European Environment agency European emissions of Ni, Zn, Cr to air have steadily decreased since 2007 by more than 50%, which may indicate that Ni, Zn and Cr observed at Zeppelin have sources of more local origin.

Mercury (Hg) is a pollutant of particular concern that has a complicated biogeochemical cycle involving atmospheric transport,
deposition to land and water surfaces, re-volatilisation and uptake by plants (Selin, 2009). Hg can exist in many different chemical forms and conversion between these forms through oxidation. Methylation results in toxic methylmercury that bioaccumulates and biomagnifies through the food web (Selin, 2009). Building on the 1998 Protocol on Heavy Metals, the Minamata Convention on Mercury (MC) was adopted in 2013 and entered into force in 2017. MCM is a global treaty under UNEP to protect human health and the environment from the adverse effects of Hg. The major content of this treaty includes
a ban on new Hg mines, the phase-out of old Hg mines, control measures on air emissions, and international regulation of the informal sector for artisanal and small-scale gold mining.

In the atmosphere, mercury is characterised by a variety of chemical and physical forms, however the most abundant is gaseous elemental mercury (GEM) with an atmospheric lifetime of 0.5 to 1 year (Schroeder and Munthe, 1998). At Zeppelin, the mean gaseous elemental mercury (GEM) concentration, combining manual and automated sample collection methods, is $1.5 \pm 0.24$
1015    ng m$^{-3}$. The measurement time series is one of the longest GEM time series worldwide (Fig. 18).



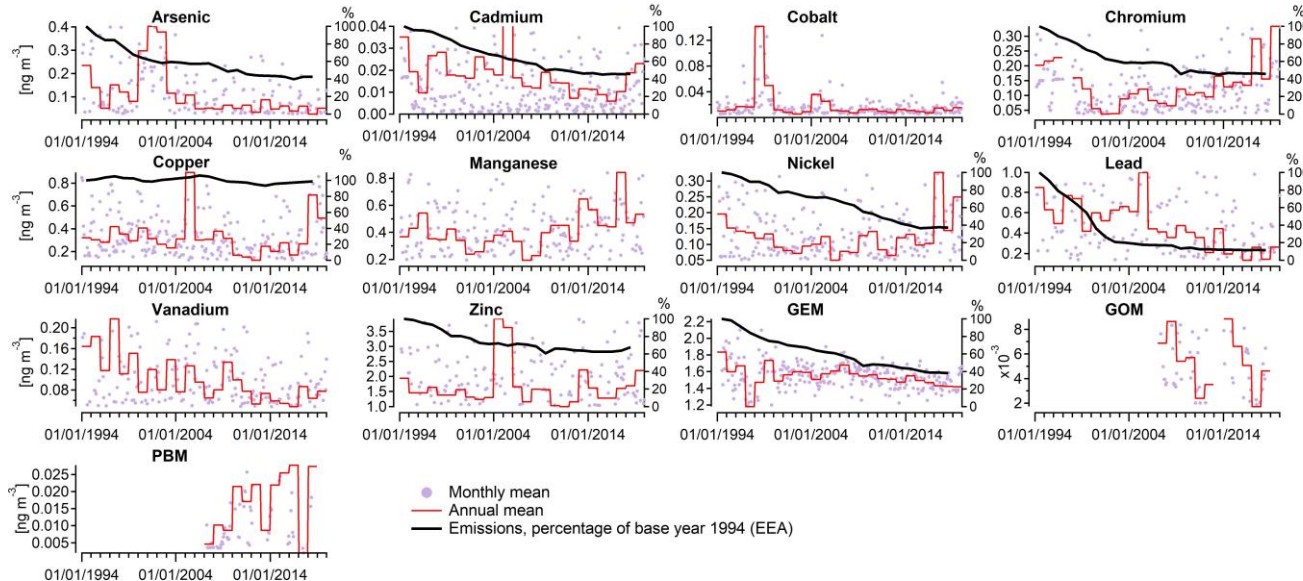

**Figure 18: Monthly (violet) and annual (red) mean concentrations of heavy metals, gaseous elemental mercury (GEM), gaseous organic mercury (GOM), and particle-bound mercury (PBM) at the Zeppelin Observatory. Estimated emissions as a percentage of base year 1994 according to the European Environment Agency (EEA, black) are shown on the right axes where available. Monthly and annual means calculated only where data coverage is 75% of total month or year, respectively.**

GEM is subject to oxidation chemistry and converted into the operationally defined gaseous oxidised mercury (GOM, also commonly called RGM) and particulate bound mercury (PBM also known as PM or PHg). Hg in the Arctic undergoes large-scale rapid conversion from GEM to GOM and PBM in the spring during the 'atmospheric mercury depletion events' (AMDE) (Schroeder and Munthe, 1998;Berg et al., 2003;Steffen et al., 2008). These chemical reactions are associated with sea-ice through surface bromine reactions (Steffen et al., 2008;Steffen et al., 2015). Previous trend analysis (Mann Kendall Statistic, Sen's slope) on shorter sub sections of the time series do not show any significant trends, neither for combined offline and online sampling e.g. Berg et al. (2004) for 1994 to 2002, Berg et al. (2008) for 1994 to 2005, nor for purely online sampling e.g. Berg et al. (2013) for 2000 to 2010. However, combining the complete data series from 1994 to 2019 a decreasing annual trend of 0.55% yr⁻¹ was observed (0.6% yr⁻¹ when considering only online sampling).

Decreasing trends in long-term GEM concentrations have been reported for many ground-based sites in Europe, North America and Asia in the range of 1.3 to 2.7% yr⁻¹ as summarised in (Lyman et al., 2020). The declines are smaller at Arctic sites compared to temperate locations (Cole and Steffen, 2010;Ebinghaus et al., 2011), and the concentrations are declining more slowly at Zeppelin than Alert (Cole et al., 2013). This is likely due to summertime Hg emission from the ocean and meteorological effects resulting from climate change (Cole and Steffen., 2010). The main source of high mercury concentrations at the Zeppelin observatory originates from continental Europe (Hirdman et al., 2010a). Though Hg emissions reductions in Europe have declined by 61% since 1994, the Hg concentration in air is only reduced by 14% during the same





period. Furthermore, emissions from East Asia including China, contribute to the global background levels of mercury, compounding the European emissions reduction signature in the observations (Streets et al., 2019).

The seasonal variation of GEM at the Zeppelin observatory displays high concentrations in winter/summer and low
concentrations in spring/autumn (Fig. 17). This is in contrast to the pattern at temperate northern latitudes with highest concentrations in winter and lowest concentrations in summer (Sprovieri et al., 2016;Temme et al., 2007), which are mainly attributed to primary anthropogenic mercury emissions from coal combustion for domestic heating (Temme et al., 2007;Weigelt et al., 2015). Global Hg models have so far not been able to test this hypothesis as current anthropogenic mercury emission inventories have no seasonal resolution and are kept constant throughout the year (Holmes et al., 2010;Song et al.,
2015;Horowitz et al., 2017). The seasonal pattern at the Zeppelin Observatory is strongly influenced by ADMEs taking place through fast oxidation mechanisms initiated by photochemistry involving halogens derived from heterogeneous reactions on hygroscopic sea salt aerosols (Steffen et al., 2015). AMDEs cause the springtime low GEM concentrations, and the summertime high is caused by either re-emission of previously deposited GEM during spring or from GEM volatilisation from the ocean (Hirdman et al., 2009;Berg et al., 2013).

Concentrations of GOM and PBM at Zeppelin are low for most parts of the year but are elevated during spring and summer, again due to AMDEs, though still lower compared to other Arctic sites (Lindberg et al., 2002;Steffen et al., 2008). This is likely because Zeppelin is located relatively far from where the AMDEs take place and most of the Hg species have already deposited before being captured at Zeppelin (Steen et al., 2011). Trends in speciated Hg measurements have been investigated in the most recent AMAP Hg assessment, and it was found that trends for GOM are declining for the months from February
through September with no significant trend for the remainder of the year. On the other hand, trends for PBM are increasing for the months January through May and in November, while decreasing for September, October, and December. The shift in speciation from GOM to PBM in spring suggests an influence of changing Arctic conditions on AMDEs.

**Table 8: Annual trends (%) as calculated from Sen's slope on annual and (individual) monthly means. Significant trends at 95% confidence level are given in bold.**

|         | As       | Cd       | Cr   | Co       | Cu   | Pb       | Mn      | Ni   | V        | Zn      |
|---------|----------|----------|------|----------|------|----------|---------|------|----------|---------|
| **Annual** | -3.8     | -2.8     | 2.7  | -0.6     | -0.9 | -4.6     | 1.9     | 0.2  | -3.8     | 0.1     |
| **Jan.**   | **-5.6** | **-5.4** | -0.3 | **-2.0** | -2.5 | **-5.3** | -0.5    | -1.8 | **-5.2** | -2.3    |
| **Feb.**   | **-5.3** | **-1.9** | 0.3  | -1.5     | -2.0 | **-4.9** | 0.5     | -1.3 | **-3.6** | -1.4    |
| **Mar.**   | **-4.9** | **-2.8** | 2.4  | -1.3     | -1.2 | **-4.8** | 0.0     | -2.0 | **-6.9** | -1.4    |
| **Apr.**   | -1.1     | 1.2      | 0.2  | -0.9     | -0.7 | **-2.5** | 1.3     | -0.7 | **-2.7** | -0.1    |
| **May.**   | **-1.2** | -1.7     | 1.0  | -0.5     | -0.9 | **-3.4** | 1.4     | -0.2 | -1.1     | -1.2    |
| **Jun.**   | -0.4     | 0.2      | 0.9  | 1.3      | 1.0  | **-1.8** | **4.0** | 1.5  | -0.5     | 0.2     |
| **Jul.**   | -0.6     | 0.5      | 1.2  | 0.4      | 1.8  | -0.7     | 2.2     | 0.4  | 0.2      | **4.9** |
| **Aug.**   | **-1.8** | -0.8     | -1.3 | -1.4     | -1.4 | **-3.4** | 1.6     | -1.9 | -2.9     | 0.1     |





| | | | | | | | | | | |
|---|---|---|---|---|---|---|---|---|---|---|
| **Sep.** | -0.9 | 0.0 | 0.2 | -0.3 | **-3.4** | -0.3 | -1.1 | 0.0 | 0.7 | -1.9 |
| **Oct.** | -0.4 | 0.8 | 2.2 | 0.4 | 0.7 | -0.6 | **3.1** | **4.1** | 2.0 | 2.4 |
| **Nov.** | **-1.4** | -1.9 | 2.9 | 0.5 | -1.8 | **-3.5** | 2.8 | 1.2 | -0.1 | -0.4 |
| **Dec.** | **-2.7** | **-2.2** | 2.4 | -0.4 | **-2.0** | **-3.0** | 1.9 | 0.4 | -2.3 | -1.0 |

## 5.7 Surface ozone

Ozone is an important species in the troposphere with implications for global warming and the atmospheric chemistry in general, and knowledge of ozone in the Arctic is of particular interest for assessing the overall chemical state of the background atmosphere in the northern hemisphere. The Zeppelin Observatory has one of the longest continuous surface ozone time series in the Arctic with its 30-year history.

As mentioned by Zhou et al. (2017) no clear and consistent trends for baseline ozone in the northern hemisphere have been found. The most recent ozone trend evaluation for the Zeppelin Observatory and 26 other remote sites was conducted by Cooper et al. (2020) as a follow-up of the TOAR project (Tropospheric Ozone Assessment Report). They looked at trends over various time periods; 2000 to 2017, 1995 to 2017 and the full record, i.e. 1989 to 2017 for the Zeppelin Observatory, and found an increasing trend but the significance is weak (p <0.10) when analysing the full period starting in 1989. They estimated an increase in mean ozone of 5 % from the start to 2017 for the Zeppelin Observatory. For the other time periods, there was no significant trend based on their data selection (monthly anomalies in mean concentration).

Several projects have been dedicated to process studies of ozone in the Arctic troposphere (e.g. POLARCAT, TOPSE etc.), but fewer studies have focused on long-term trends. Results from POLARCAT revealed that in spring and summer, anthropogenic emissions from Europe are found to contribute significantly to ozone in the lower troposphere over the eastern Arctic (Law et al., 2017;Wespes et al., 2012). This is consistent with the atmospheric transport studies by e.g Stohl (2006) and Pisso et al. (2016) as discussed in more detail in Sect. 3.2.

$O_3$ has slowly increased at Zeppelin Observatory from an annual average of ~62 ppt in 1990 to ~70 ppt in 2019, with a smooth trend indicating a modest increase from the start to 2003-2006 followed by a flat or slightly decreasing levels, Fig. 19. $O_3$ has a marked seasonal cycle with maximum values during the haze period in spring (March) and minimum values in summer (July).

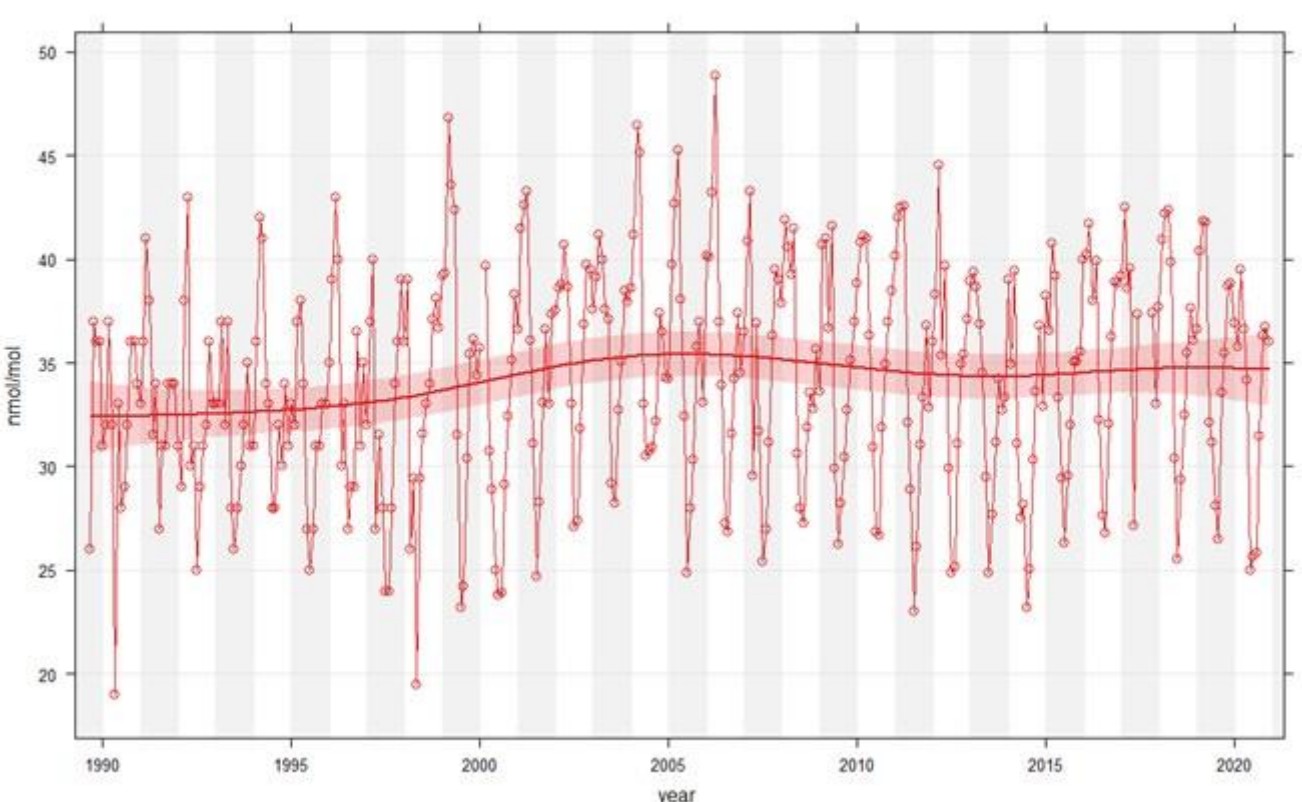

**Figure 19. Monthly median concentrations of O₃ at the Zeppelin Observatory during 1989 to 2020 [nmol mol⁻¹]. Superimposed on the data is a smooth trend function (based on a generalised additive model (GAM) as provided by the SmoothTrend function in the R library SmoothTrend). Together with the trend is the 95 % confidence interval shown as a band.**

Figure 20 shows the Theil-Sen slope for each 3-months season separately together with the monthly ozone median values.

This indicates increasing ozone levels in autumn, winter and spring and decreasing levels in summer, but the trend is only

statistically significant ($p < 0.05$) in winter. The winter trend is clearly driven by a strong increase from 1990 to 2000 and small

changes after that, in accordance with the findings of Cooper et al. (2020). Particularly strong variability is found in spring

which reflects the combined effect of a peak in the northern hemispheric tropospheric baseline ozone in this season combined

with episodes of very low ozone (LOEs) in the Arctic as discussed in more detail below.



Median $O_3$ at Zeppelin Mtn

**Figure 20. Theil-Sen slopes and confidence intervals for monthly median $O_3$ concentrations [nmol mol$^{-1}$] at the Zeppelin Observatory**
**during 1989 to 2020 for four seasons separately.**

Already in the 1980s it was discovered that LOEs occur at the surface every spring in the Arctic (Bottenheim et al., 1986;Barrie

et al., 1988). The ozone monitoring at the Zeppelin Observatory showed that the LOEs were frequent also at that location and

altitude (472 m asl), and the co-located monitoring of NMHCs offered a good opportunity to evaluate the atmospheric

chemistry behind the LOEs (Solberg et al., 1996), since the build-up of NMHCs in the northern atmosphere cold season is

relevant to the spring peak in tropospheric ozone seen at most rural background sites in central and northern Europe (Roemer,

2001).

It was soon proposed that self-catalytic reactions involving halogen radicals (Br and Cl) played an essential role for the LOEs

(Barrie et al., 1988;Bottenheim et al., 1990;Hausmann and Platt, 1994). By the early 1980s Berg et al. (1983) found elevated

levels of particulate bromine levels in spring at Ny-Ålesund and Barrow but noted that heterogeneous reactions were required



to release and activate the particulate Br into gaseous form. Campaign measurements of NMHC at Alert confirmed that halogen reactions were indeed taking place (Jobson et al., 1994).

The measurements at the Zeppelin Observatory revealed that the changes observed in the NMHC profile (the relative distribution of NMHC species) during the LOEs could not be explained by standard OH chemistry, whereas it was consistent with significant levels of Cl radicals in the Arctic atmosphere. Furthermore, particularly low levels of acetylene during LOEs

indicated atmospheric oxidation initiated by Br radicals as well, since acetylene is particularly reactive with respect to Br (Solberg et al., 1996b). Within the EU research project ARCTOC, extensive field campaigns were carried out during 1995 and 1996 at Ny-Ålesund leading to the detection and quantification of essential components of the halogen self-catalytic reactions, such as Br, Cl, BrO and ClO.

Initially, the occurrence of LOEs was regarded an isolated phenomenon only of importance for the Arctic tropospheric ozone

budget. Then, measurements at Ny-Ålesund and Alert in 1998 revealed that these episodes were strongly associated with the deposition of particle-bound mercury in the Arctic (Lu et al., 2001). Their data showed that the halogen radicals involved in the LOEs also lead to a rapid transformation of long-lived gaseous elemental mercury (GEM) to total particulate-phase mercury (TPM) that was subsequently effectively deposited to the surface (Sect. 5.6). This established an important link between the LOEs and the input of Hg to the Arctic biosphere in spring. Furthermore, links between climate change and the occurrence of

the LOEs in the Arctic have been proposed and (Koo et al., 2014) found correlations between so-called 'teleconnection patterns', i.e. weather patterns in other regions like the Western Pacific, and the frequency of LOEs in the Arctic.

### 5.8 Reactive nitrogen

Reactive nitrogen species in the atmosphere play a key role in many issues linked to atmospheric pollutants, e.g., acidification, aerosol formation and photochemical ozone episodes. The unique polar environment with low temperatures and prolonged

periods with little solar radiation means that the behaviour of reactive nitrogen differs in important ways from that seen at lower latitudes. Peroxyacetyl nitrate (PAN) has been suggested as a main reservoir of oxidised nitrogen species in the Arctic atmosphere, particularly in winter, since the chemical lifetime of PAN is strongly dependent on temperature. Furthermore, an Arctic wintertime PAN reservoir has been proposed as a contributing source to the springtime peak in tropospheric ozone in the background northern hemisphere. If PAN is accumulated at high latitudes during winter, the rising temperatures in spring

could lead to PAN being decomposed back to $NO_2$, a main ozone precursor in the background atmosphere.

PAN was measured during 1994 to 1996 at the Zeppelin Observatory and a marked seasonal cycle was found with a minimum at or below 100 ppt in summer and a maximum in March/ April (~day-of-year 90) at levels ≈400 ppt (Fig. 21). This is consistent with the expected behaviour of reactive nitrogen at high latitudes. During summer, concentrations are kept low by thermal breakdown and a multitude of photochemical reaction chains. During the cold and dark winter, thermal decomposition is low,

and there is no local photochemical activity. The slightly increased winter concentrations can be attributed to long-range transport of well-mixed air masses. During the spring season, light intensity and photochemical processes increase sharply,





while thermal decomposition is still low, and we see multiple short-lived episodes of high PAN and PPN concentrations, merging into a spring maximum with a duration of 1 to 3 months.

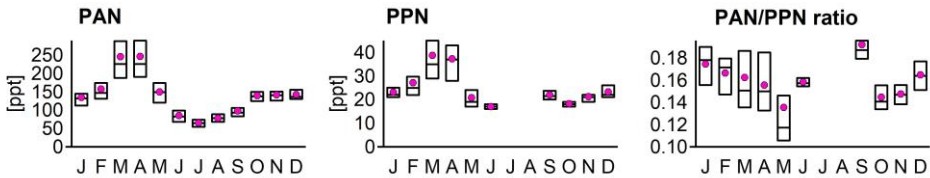

**Figure 21: Boxplot of monthly variation (box top and bottom: 75th and 50th quartiles, respectively, horizontal bar: median) and monthly averages (pink) of peroxyacetyl nitrate (PAN), peroxypropyonyl nitrate (PPN), and the PAN/PPN ratio at the Zeppelin Observatory in 1994 to 1996.**

A close correlation with the seasonal cycle of surface ozone was found, except during low ozone episodes (LOEs), indicating that PAN was not part of these local events (Beine and Krognes, 2000). In spring, PAN levels approached the level of $NO_Y$, confirming that PAN constitutes a major fraction of odd nitrogen species in the Arctic in this season (Solberg et al., 1997). Note that unfortunately, substantial uncertainties ($\approx 20\%$) in both the $NO_Y$ and the PAN measurements hindered a precise calculation of the fractionation of the individual NOy species. Based on the temperatures recorded at the Zeppelin Observatory it was concluded that PAN was too stable in the Arctic atmosphere in spring to contribute to local $NO_2$ formation and subsequent ozone formation, but it was not ruled out that such processes could occur during air mass transport to the Arctic (Beine and Krognes, 2000).

Along with the PAN measurements, PPN (Peroxypropyonyl nitrate) was measured with the same instrument. A PPN/PAN ratio of 0.1 to 0.2 was found through the year (Fig. 21). Compared to studies of reactive nitrogen at lower latitudes (e.g. Singh and Salas, 1989;Shepson et al., 1992), this indicates that this ratio may have been overestimated by a factor of 2.

As for many other trace gases measured at the Zeppelin Observatory, a clear link between air mass origin and PAN/PPN concentrations was found, with highest levels linked to transport from the Russian sector and lowest levels linked to marine, Atlantic air (Solberg et al., 1997). The latter reflects that Atlantic air normally carries cleaner air masses, but it was also speculated that heterogeneous reactions involving PAN and other oxidised nitrogen species could reduce the species in humid air as a clear link between humidity and PAN levels was found (Beine et al., 2000).

In addition to PAN/PPN, $NO_X$ was measured at the observatory in 1994 along with the $NO_2$ photolysis rate (Beine et al., 1996, Beine et al., 1999). Unlike most other trace gases, a seasonal cycle with maximum in spring was not found for NOx. Instead, the monitoring data showed levels of ppt on average without any systematic pattern through the year. Based on the NOx data it was indicated that the Zeppelin Observatory was influenced by local emissions from Ny-Ålesund a mere 6% of the time, but it should be noted that the measurement period was short and that detailed information of the local wind field around the Mt. Zeppelin was not available.



## 6 The future of the Zeppelin Observatory

Many trends in atmospheric research have shifted over the decades. The original emphasis on establishing a global background, e.g. (Junge, 1972) (Sect. 1) has been replaced by an understanding that there is no longer an atmosphere, anywhere on Earth, unperturbed by humans. Thus, the focus of many of the programmes and measurements described above is now on understanding the balance of atmospheric and Earth-systems processes, with an emphasis on understanding the present and future impact of anthropogenic activities.

We identify three areas important for the future of the observatory and for Arctic atmospheric research: 1) a broad need to maintain and strengthen the position of the Zeppelin Observatory as a leading global measurement platform, 2) examining the effects of rapid climate change particularly for aerosols and the carbon cycle where there is potential for feedbacks and tipping points, and 3) monitoring and study of new and emerging atmospheric trace constituents of relevance to health and climate, e.g. emerging contaminants such as POP-like chemicals of concern and CFC/HFC/HCFCs with very high global warming potentials.

### 6.1 Securing the standing of the Zeppelin Observatory

Global background sites offer unique opportunities for monitoring and research and Zeppelin Observatory's location ensures that it will remain at the forefront of atmospheric science for years to come. The partners at the Zeppelin Observatory (NPI, SU, NILU) are actively engaged in securing this future and a new strategic plan for Zeppelin has been published (Steen et al., 2021). Many changes in human society occur over decadal timescales, if not longer, as do many atmospheric processes linked e.g., to the long lifetimes of $CO_2$ and CFCs, hence background monitoring sites should maintain as many time series as possible which are compatible backwards in time, while at the same time introducing new measurements of emerging pollutants like airborne microplastics that need study. One must prepare for surprises, where a site like the Zeppelin Observatory can add a lot of information. Hence, the strategic plan includes ensuring data quality via traceable references (good metadata), deploying state-of-the-art instrumentation, and monitoring the parameters that are relevant for understanding anthropogenic influences.

To maintain data quality, continued minimal local contaminants levels must be ensured. As the region rapidly changes, alongside monitoring activities, it is important survey the effects of local emissions on the measured constituents in all ongoing monitoring programmes. Ensuring minimal local contamination, linked to the activities at the observatory, as well as actively seeking to reduce emissions in the Ny-Ålesund settlement, is essential, as is logging local emissions. The interaction of mixing processes on the local, meso and regional scales needs to be under permanent surveillance to assess how measured constituent levels are impacted by them. Examples of such work are the studies of (Eckhardt et al., 2013;Dekhtyareva et al., 2018), demonstrating an influence of cruise ship emissions, now largely mitigated by the 2015 heavy fuel oil ban for ships close to the shoreline around Svalbard. Furthermore, the changing climate in Svalbard is likely to impact on the local dispersion characteristics and increase the frequency of local dust and sandstorms (e.g., by decreased glaciation) which would undoutably influence aerosol distributions, particle number and metal concentrations both locally and on the regional scale and their might



be associated climate feedbacks. It will remain important to be able to distinguish between the impact of both local and regional dust.

Further steps outlined by (Steen et al., 2021) to maintain the leading position of the observatory include maintaining open and accessible data, following FAIR principles (Wilkinson et al., 2016) and making meta-data and the physical data available in open databases promptly after reporting. Long term funding, good management routines, trained staff and stable and adequate infrastructure are also essential.

## 6.2 Arctic change

Arctic amplification (Sect. 1) has been linked to surface albedo feedbacks (Serreze et al., 2009; Hall, 2004), an increase in solar radiation absorbance due to loss of snow and ice, increased heat transfer to the atmospheric surface layer from the ocean following sea ice loss (Serreze et al., 2009), vapour feedbacks due to increaseing atmopsheric water vapour (Graversen and Wang, 2009), and the Planck feedback (Pithan and Mauritsen, 2014), whereby a given increase in emitted radiation (R) requires a larger temperature (T) increase when the background temperature is low, following $R = \varepsilon \times \sigma \times T^4$ where $\varepsilon$ is the surface emissivity and $\sigma$ the Stefan–Boltzmann constant (Planck, 1978). Rapid Arctic warming has changed atmopsheric transport patterns: the polar front has moved southwards by 2.5° while the polar vortex (strong westerly winds in winter which limit movement of air between higher and lower latitudes) has grown weaker, due to increased/ earlier undulations (so-called 'Rossby waves', (Mitchell et al., 2012)). This phenomenon results in very cold weather events at lower latitudes and increased heat transport northwards. Thus, global warming, enhanced by the Arctic amplification is shifting the polar climate of Svalbard towards a maritime, Atlantic climate, with consequences for the natural biogeochemical exchanges between the atmosphere, ocean, ice and eventually permafrost. The frequency of important transport pathways of pollution to Ny Ålesund will also change with the retreat of the ice cover and rapidly increasing lower troposphere temperatures. The variability and trends in the cycling of water through the Arctic atmosphere as observed at the Zeppelin Observatory is one obvious theme worth more focus due to its significance in climate change.

Shifts in aerosol properties are likely to follow these regional changes with e.g. increased biological activity (Myers-Smith et al., 2020), increases in mineral dust from areas recently free of ice, increased wildfires (both forests and tundra scrub, Hu et al., 2015), and societal changes due to easier access to Arctic oil and gas extraction (Harsem et al., 2011) and the opening of new shipping routes (Humpert and Raspotnik, 2012). So far, these changes to Arctic aerosol, and hence to regional and global climate, are not well constrained. For example, Arctic mineral dust is hardly accounted for in global models (Groot Zwaaftink et al., 2016), while ongoing research shows elevated levels of biogenic secondary organic aerosol (BSOA, e.g., 2-methyltetrols) from isoprene at Svalbard in summer of local origin (Yttri et al., in prep.).

Changes in the Arctic aerosol burden will in turn influence climate via direct and indirect aerosol effects, i.e. via increased absorption and scattering, and changes in CCN and ice nucleating particles (INP), respectively. Marine and terrestrial sources both act as INP (Hartmann et al., 2020). Primary biological aerosol particles (PBAP) are particularly important, both per se and as a coating on sea salt aerosol and mineral dust enabling activation at higher temperatures than sea salt aerosol or mineral





dust alone. Thawing permafrost can mobilise biological INP precursors into the atmosphere and, via lakes and rivers, to the
ocean, and Arctic greening can be a source of INP-active PBAPs. Essential information on polar INP is lacking, including on
activation temperature, composition, sources, origin, and seasonality (e.g., Creamean et al., 2018; 2019; 2020; Hartmann et
al., 2019; 2020). Meanwhile, most previous studies of Arctic CCN and cloud properties are based on short term campaigns,
carried out predominantly in summer. Only a handful of studies cover seasonal cycles and interannual variability (Jung et al.,

2018), while a recent study highlighted the importance of studying localized chemical composition for cloud formation in the
high Arctic, finding that oceanic iodine-driven new particle formation potentially increases cloud formation (Baccarini et al.,
2020). Long-term measurements of CCN/INP are therefore required alongside detailed information on aerosol chemical
composition.

Understanding how changing Arctic aerosol composition will influence climate requires 1) knowledge of changes in the Arctic

aerosol burden and 2) how these relate to changes in CCN and INP properties. Addressing 1) requires better knowledge in
several areas. For example, investigation of the influence of atmospheric transport patterns (e.g. physical processes and the
connection between land use and the composition of air and aerosols moving next to the ground) coupled to historical transport
patterns and future projections based on Earth system models. Improved knowledge of sources themselves, including mineral
dust, carbonaceous aerosol (e.g., BSOA, biomass burning organic aerosol, BBOA and BC) and how they evolve due to

regional/ global change is also required. Addressing 2) requires coupling of aerosol properties to observed INP and CCN
properties including concentrations and ice nucleating / cloud forming potential. This might be achieved through comparison
of INP and CCN properties to aerosol composition either via comparison to pre-existing long term data sets or dedicated
measurement campaigns incorporating state-of-the-art instrumentation including high time resolution measurements of
composition. Source apportionment techniques, such as cluster analysis or positive matrix factorization (PMF) would also

yield better link between CCN/INP properties and the factors contributing to their formation e.g. BBOA, mineral dust etc. to
better predict the impact of future changes in these sources.





There is some evidence that levels of mineral dust are increasing at the Zeppelin Observatory For example, Mn, Cr, Ni, and to

a      lesser      extent      Zn      and      Cd,      have      increased      since      ~2007      (

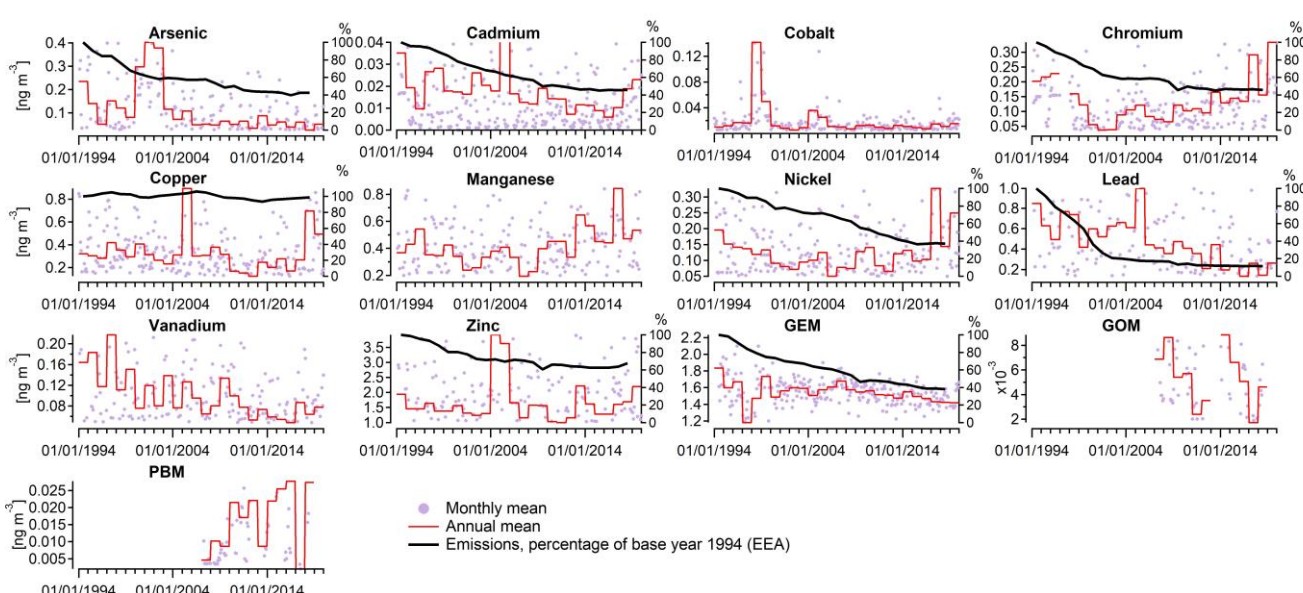

Figure **18**). While these elements have anthropogenic sources e.g., heavy industry, they are also crustal elements. It is therefore

interesting to note that at the same time as levels appear to have increased, their estimated anthropogenic emissions have

decreased, according to the European Environment Agency. This suggests an increase in local or regional emissions, possibly

from increased erosion and levels of mineral dust in an increasingly ice-free Arctic. In the case of Mn, ocean spray is a

significant source along with mineral dust (Howe et al., 2004), and the increasing Mn levels might be linked to declining sea

ice. A third possible explanation is changing weather patterns. Since mineral dust will influence climate both via aerosol effects

and on snow, it is important to elucidate the reason for the increasing levels, e.g. by transport modelling of emissions and/or

by extending the range of HMs analysed to include more crustal elements to distinguish between sources.

Although high time resolution measurements of eBC are performed at several Arctic sites (Hirdman et al., 2010a), regular OC

(and EC) measurements are generally lacking, as are studies of the Arctic organic aerosol. In a rapidly changing Arctic

environment with increased temperature and precipitation, retracting sea ice and changed circulation patterns and changes in

natural and anthropogenic emissions are likely to affect the carbonaceous aerosol, its speciation and sources (Yttri et al in

prep.). We recommend an increased focus on the Arctic carbonaceous aerosol, reflecting the current mismatch of importance

and knowledge.

As well as changes in aerosol composition, rapid climate change will have a profound influence on the carbon cycle in the

Arctic due to changes in biogeochemistry and the state of carbon reservoirs (see also Sect. 5.3). The complexity of both the

natural and anthropogenic components of the carbon cycle is therefore another crucial topic. For example, the role of Arctic



CH$_4$ in future climate is particularly important since levels have been rising unexpectedly since 2007 with negative implications for the Paris Agreement goals (see Sect. 4.3), hence future study of the intermittency of CH$_4$ concentrations and its isotopic composition is of particular importance.

The proximity of Ny-Ålesund to major carbon reservoirs on land and on the coast of Siberia is another aspect that is likely to ensure future relevance of measurements at the Zeppelin Observatory. For example, the Arctic seabed hosts a vast CH$_4$ reservoir, from 0.28 to 512 Gt carbon (Marín-Moreno et al., 2016, and refs therein), in the form of gas hydrates (GH). While previous work has demonstrated that the low CH$_4$ fluxes to the atmosphere from seaps and GHs is due to the capacity of methanotrophic bacteria to rapidly convert CH$_4$ to CO$_2$ in the water column e.g. (Silyakova et al., 2020), (Puglini et al., 2020)

demonstrated that 'sudden' sea floor CH$_4$ releases yield a 'window of opportunity' for emissions before microbial communities can react to changing water column CH$_4$.

Increased wildfires in peat beds, forests, and tundra scrub, also result from rising temperatures (Hu et al., 2015), as do changes in CH$_4$ release from anaerobic methanogenic microbial communities in high-latitude wetland soils, which also respond to changes in precipitation, i.e., anoxic conditions (Valentine et al., 1994). Thawing permafrost emits CH$_4$ directly and also causes

indirect CH$_4$ emissions, and is a potential climate feedback (Schuur et al., 2015). The direct CH$_4$ emissions result from release of trapped CH$_4$ while the more important indirect effect is due increased release of organic carbon coupled with hydrological changes, increasing the activity of aforementioned methanogenic microbes (McCalley et al., 2014). Another important non-CO$_2$ greenhouse gas is N$_2$O, with a global warming potential 265–298 times that of CO$_2$. N$_2$O is also released from anoxic soils and the changes in wetland soils and microbial communities are also relevant to this species.

Top-down CH$_4$ and N$_2$O estimates from these sources, have been assessed with atmospheric inversion frameworks (e.g. FLEXINVERT, (Thompson and Stohl, 2014)) and the Community Inversion Framework (CIF, (Berchet et al., 2020)). Such inversions are based on combining observations with an atmospheric transport model (e.g. FLEXPART) to relate changes in concentrations to changes in fluxes. The approach uses Bayesian statistics and optimizes (posterior) fluxes by minimising a 'cost-function', accounting for uncertainties in prior flux estimates and observations. Understanding developments in the CH$_4$

and N$_2$O budgets requires better integration of atmospheric chemistry (e.g. Cl oxidation) and land surface models (e.g. FLUXNET-CH$_4$, Knox et al., 2019) with top-down approaches. Furthermore, inclusion of more observational data is needed at high latitudes and in the Arctic and would reduce errors in posterior flux estimates. This might be achieved via integration of satellite data fields (such as Sentinel 5P) into inversion models, which would require not only streamlined algorithms to reduce computation times, but also careful validation of the satellite data. As one of only a handful of Arctic sites, the Zeppelin

Observatory would play a key role.

Finally, the IPCC estimates that carbon emissions must be cut by 45% by 2030 to prevent warming beyond 1.5°C, thus the next 10 years are crucial for the state of the Earth's climate from a political perspective. Several nationally determined contributions (NDCs) towards meeting the goal of limiting average warming to 2°C come into effect. For example, Norway submitted an enhanced climate target under the Paris agreement with the target to reduce GHG emissions by at least 50 %, and

towards 55 % by 2030 (Norwegian Climate Law). The EU has committed to a 40% reduction by 2030 (European Commission,



2019), and a more ambitious EU plan to cut emissions by 55% was presented in September 2020. These legal requirements are likely to see considerable focus on GHG emission compliance, and in-situ observations at Zeppelin Observatory will play a key role at the national level (Zeppelin is one of only two ICOS atmospheric observatories on Norwegian territory as of 2020) and the international level as a global background site. This focus on GHG emissions, together with an understanding of the importance of the Arctic for climate is an opportunity to gain political support to establish a pan-Arctic observational capability, crucial to examining the impacts of the rapidly altering regional land and marine conditions on the Arctic rim states. The institutional support of the Zeppelin Observatory should be discussed first at the national level in Norway/ Sweden where weather, marine and ecosystem research should align their objectives and capabilities, and then an international initiative could be undertaken to further develop a pan-Arctic earth system observing capability involving all the Arctic rim states.

### 6.3 Emerging environmental concerns

Many emerging pollutants like airborne microplastics (Evangeliou et al., 2020) will need study. The backdrop provided by the long-time series from the Zeppelin Observatory forms a unique opportunity for process oriented or basic research experiments. Zeppelin will undoubtedly be a primary location for this in years to come. Surprise events with environmental effects can be followed up, radioactivity was for example detected at the Zeppelin Observatory 10 days after the Fukushima nuclear incident (Paatero et al., 2012) demonstrating a different long-term justification for the observatory.

The long-term POP monitoring programme at Zeppelin documents a general decline for most regulated POPs. However, the concentrations of some of these POPs decline only very slowly, or even show occasional increases, such as for HCB and PCBs (Sect. 5.5). These examples highlight the need for sustained monitoring at Zeppelin to ensure that global chemical management strategies remain effective. Attention should be given to the legacy POPs which remain to be of ecotoxicological concern, and for which contemporary emissions remain poorly characterised. The example discussed for HCB furthermore illustrates the utility of the FLEXPART model to identify regional and global source regions when these are poorly constrained.

At the same time, new organic chemicals are continuously entering the market, either as substitutes to replace the regulated POPs or to fulfil new demands. Some of these chemicals may have similar impacts on ecosystems as the legacy POPs while some may fulfil persistence and mobility criteria but do not necessarily bioaccumulate. The latter do, however, need to be put on an equivalent level of concern to traditional POPs. To support and improve regulatory actions, there is a need to gather proofs of persistence, long-range transport and impact of new chemicals. In the Norwegian Arctic the detection and prioritisation can be achieved of new potential chemicals of emerging Arctic concern (CEACs). The detection of a chemical in Arctic air is a good indication for its persistence and long-range transport, after local sources have been excluded.

Targeted screening projects aiming at identifying CECs in various environmental matrices are important for prioritizing CECs to include in Arctic monitoring programmes. The results of such studies have provided the evidence needed to include cVMS, chlorinated paraffins, novel flame retardants, dechloranes and a broader set of PFAS in the routine monitoring programme at Zeppelin. A complementary approach for identifying potential CECs for targeted analysis is to use in silico tools for screening large lists of chemicals in commerce to identify chemicals that can be transported into the Arctic (Brown and Wania,





2008;Howard and Muir, 2010). While the targeted approaches used for monitoring of POPs apply very selective sample clean-up and analytical methods only allowing for detection and analysis of a very limited number of target compounds, the non-target methods allow for detection also of unselected chemicals. Recent instrumental developments will allow a much broader analytical approach by using suspect and non-target screening. Röhler et al. (2020) identified previously not detected compounds in Arctic air by using non-target and suspect screening methods on high-volume air samples. This shows that combining air sampling with new analytical methodologies can be a tool for early identification and an early-warning system for airborne CECs.

Lastly, beyond $CH_4$ and $N_2O$, there are several other non-$CO_2$ climate gases monitored at the Zeppelin Observatory with extremely high global warming potentials (GWP) compared to that of $CO_2$. These include CFCs such as CFC-11 (GWP=4660), CFC-12 (GWP=10200), and CFC-113 (GWP=13100), as well their replacement HFCs e.g. HFC-23 (GWP=12400) and $SF_6$ (GWP=23500). Accordingly, the CFC/HCFC/HFC family accounts for 12% of the increase in radiative forcing since 1750, despite mixing ratios 2 to 3 orders of magnitude lower than that of $CO_2$ (Myhre et al., 2013). Presently, the contribution to global warming posed by CFC/HCFC/HFC is very limited, since concentrations are extremely low. However, since levels of many of these compounds are increasing rapidly their development must be carefully followed (Myhre et al., 2020). Of particular concern as shown in recent studies is the slowing down of the rate of decline in CFC-11 by ≈50% after 2012, both globally and at Zeppelin (Montzka et al., 2018). This is probably related to unreported emissions in China (Rigby et al., 2019), though this emissions source has now been stopped Park et al. (2021).

**7 Conclusions**

With continuous measurements of a range of atmospheric trace gas components since 1989, the Zeppelin Observatory is a cornerstone of national and international monitoring programs and Arctic atmospheric research. The construction of the observatory was motivated by the need to monitor the global background levels of aerosols, gaseous species related to climate change, ozone layer depletion, Arctic haze, changes in the oxidizing capacity of the global atmosphere, accumulation of persistent organic species in the food chain, heavy metals and in particular mercury, and eventually related to earth system dynamics and changes. While the observatory at its inception was primarily focussed on national monitoring, the Zeppelin Observatory now host measurements from 17 institutions in13 countries. Although mostly long-term measurements, Zeppelin Observatory regularly host instruments for short-term (1 to 3 years) campaigns. Measurement capabilities have been continuously improved to include state-of-the-art instrumentation.

The location of the observatory was selected to minimise local influences and surface exchange, based on measurements of $CO_2$ and sulfate aerosol. Subsequent analysis with the FLEXPART model confirmed that the Zeppelin Observatory receives air mostly from above 500 m a.s.l. due to frequent temperature inversions and from the unpolluted wider Arctic with little influence from the Ny-Ålesund settlement and a minor influence from cruise ships. Because of this, the site experiences some of the lowest levels of particulate matter in Europe. Aerosol levels are influenced by the formation of Arctic haze with high





levels of EC and OC in the Arctic spring, with a second peak in OC seen in August/ September, a result of biogenic emissions of PBAP and BSOA formation. Meanwhile, overall declines in sulfate and nitrate reflect the success of the Gothenburg protocol.

The Zeppelin Observatory is now an ICOS class 1 site, making an important contribution to Norwegian national monitoring and international monitoring of greenhouse gases. Time series of $CO_2$ and $CH_4$, dating back to 1989 and 1994, respectively (pre-dating ICOS), reflect the global trend of long term increases in $CO_2$ and recent increases since ~2007 for $CH_4$. Similarly, CFC/HCFC/HFC monitoring is undertaken at Zeppelin as part of AGAGE and is a key station for the monitoring of these species as ozone depleting substances for the Montreal protocol and towards meeting the Paris Agreement as species with high global warning potentials. We have shown how Arctic climate change is driving rapidly evolving capabilities to study the Earth System as a seamless, integrated whole, providing new opportunities and responsibilities for the Zeppelin Observatory agenda and for Norwegian authorities and research institutions. The backdrop of the long-time series provides a unique opportunity for both process-oriented and basic research experiments. The Zeppelin Observatory will undoubtedly be one of the primary locations for this for years to come.

## 8 Data availability

Most data are publicly available on ebas.nilu.no or else on request via the responsible institutions listed in Tables 1-4.

## 9 Author contributions

Compiling co-author input and leading the manuscript preparation: SP, Kjetil T, ØH. Preparation of figures: SP, KAP, KB, SE, H-CH, RK, PN, SS, AS, JS, KEY. Data analysis: SP, SS, SE, RK, DH-R, OH, TK, CL, NS, PBN, TS, KEY, PZ. Writing manuscript sections, Introduction and historical aspects: SP, ØH, SS, CAP, JH, KH, SH, SL, GH; Atmospheric transport: NE, AS, SE; Aerosol chemical composition: KE, KEY, MF, YJY, K-TP, WAa; Aerosol physical properties: MF, H-CH, DH-R, RK, JS, PZ, YJY, K-TP; Climate gases: SP, OH, TS, CL, CLM, EN, RF, DL, TR, CvdV; Non-methane hydrocarbons: SS, NS, CLM; Persistent organic pollutants: KB, SE, PBN; Heavy metals and mercury: KAP, KEY, TB; Surface ozone: SS; Reactive nitrogen: TK,SS; Future of the observatory: SP, KEY, WAa, ØH, CAP, KH. Data collection and station operations: OH, CL, NS, Kjersti T, RK, H-CH, DH-R, RK, JS, PZ, CAP, KH. Review of manuscript: all.

## 10 Competing interests

The authors declare that they have no conflict of interest.



## 11 Acknowledgements

The Ministry of Climate and Environment in Norway and the Swedish EPA have provided the funding for the facilities on the Zeppelin Mountain. The individual time series have been funded through various national or international programs and projects, and we refer to the list of References for details. For the Norwegian monitoring activities, the Norwegian Environment Agency is the major source of funding.

The authors acknowledge NPI, SU, and NILU staff/engineers that have worked every day of the week to maintain and operate all the instruments at Zeppelin Observatory over 30 years. Without them, we would not have had all the high-quality long term timeseries without data gaps.

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
