# Peer review of "Atmospheric composition in the European Arctic and 30 years of the Zeppelin Observatory, Ny-Ålesund"

_Atmospheric Chemistry and Physics, 2021_

## Referee Comment (RC2)

Title: Atmospheric composition in the European Arctic and 30 years of the Zeppelin Observatory, Ny-Ålesund

Authors: Stephen M. Platt et al.

**REFEREE COMMENTS**

**General comments**

The paper provides an overview on The Zeppelin Observatory, a history of the station, measurements and trends and a review of the current state of the European Arctic atmosphere. The paper is well structured and written. However, it is quite long and the authors could consider shorten the text and add some summarizing figures or statistics. The abstract could include short statement on the most scientific significant result(s). For demonstrating the scientific impact of the observatory there could be also some statistics of the published papers, number international projects over the years when / where Zeppelin dataset has been used.

**Specific / minor comments**

- line 53: how the decision of the Swedish parliament accepted for a Swedish monitoring program in connected to the Norwegian approach. Please clarify.
- line 842: History; you could add the time line figure demonstrating the different atmospheric composition measurements at different locations. This would also provide a general overview on the development and availability of long term measurements and address the role of Zeppelin measurements.
- line 113: add "full stop" after parenthesis.
- lines 195-199: please add references for the location climate / vegetation classification.
- line 224: please add a reference if possible.
- line 255: you could use "/sios-svalbard.org/"
- line 285: open the acronym FLEXPART particle dispersion model:>>> "FLEXible PARTicle dispersion model". Please re-check all the acronyms in the text and open the acronym when mentioned for the first time
- Fig 3: improve quality (resolution) of the figure
- line 316: Aitken mode particles, please add size
- Fig 4 please improve quality (resolution) of the figure
- line 340: you could add a short overall (meta) description / table of all measurements which would better describe the overall measurement capacity of the station. And give some general statistics of the measurements.
- line 350-365: (4.1) this is very detailed description of the samplings and filters, you could consider a schematic figure of the process or an annex.
- line 366: add the reference for Mann-Kendall Test/Sen's slope.
- Fig 6., 7., 8. technical quality of the fig should be improved
- line 439: "With this set-up the Zeppelin Observatory is now one of the first global aerosol observatories with semi-continuous in-situ cloud sampling". What are the other stations ?
- line 839, 865: some error with the reference (*Error! Not a valid bookmark self-reference*)
- line(s) 925 & 962 please check, add the reference Petäjä et al.2020 Overview: Integrative and Comprehensive Understanding on Polar Environments (iCUPE) – concept and initial results, Atmos. Chem. Phys., 20, 8551–8592, https://doi.org/10.5194/acp-20-8551-2020, 2020.
- line 1074: please add some specification for the acronyms "POLARCAT" "TOPSE"

Title: Atmospheric composition in the European Arctic and 30 years of the Zeppelin Observatory, Ny-Ålesund

Authors: Stephen M. Platt et al.

- line 1172: "strengthen the position of the Zeppelin Observatory as a leading global measurement platform, perhaps" one of the ? / Arctic ?
- line 1200: Refer to "open access", how is the data access to Zeppelin measurements currently organized ?
- line 1227: "Changes in the Arctic aerosol burden will in turn influence climate via direct and indirect aerosol effects, i.e. via increased absorption and scattering, and changes in CCN and ice nucleating particles (INP), respectively. " - add reference
- line 1228: "Another important non-CO2greenhouse gas is $N_2O$, with a global warming potential 265–298 times that of CO2." - add reference

---

## Author Comment (AC1)

Response to anonymous reviewer #1

We thank the reviewer for their careful consideration of our manuscript and their helpful comments.

We address the review comments (blue text) point-by-point as follows:

Overview of the obtained results (section 5) has several scientific problems, as outlined below in more detail, and these problems need to be fixed before the paper can be accepted for publication. Concerning section 6, I am not fully in favor of putting strategic/political (section 6.1) and scientific (sections 6.2 and 6.3) aims side by side, but it is up to the editor to decide whether this requires some restructuring of the text.

We address the points raised for Section 5 in the remaining comments below. The discussion on strategic goals has been moved to a separate section 7 to distinguish between political and scientific issues.

Section 5.2.

I do not understand the first statement of section 5.1 (lines 652-653). Based on figure 6, it is impossible to see whether OC and EC resemble each other at any time (their concentrations levels certainly do not, and resemblance of concentrations ratios is also questionable). Overall, I do not see how Figure 6 could tell anything about the similarities in OC and EC source regions.

We have deleted the first statement of section 5.1 as this requires a more thorough discussion of a specific issue which is then beyond the broad scope of our article. To show that the time series of OC and EC resemble each other, we provide a scatter plot in a second panel in Figure 6, showing a Pearson's $R^2$ correlation of 0.96 in October to May 2019, and 0.77 for June to September 2019.

The statement on lines 575-677 is unclear. I suppose that the authors mean that the CF conversion factor typical for aged aerosols should be applicable for Zeppelin because of its remote location from main sources. The wording (complies well with) gives an impression that this thing has somehow confirmed for Zeppelin.

The conversion factor we use is that suggested for non-urban sites by Turpin and Lim, 2001. We have changed the text to "A conversion factor (CF) of 1.9 to 2.2 is suggested for conversion of OC to organic matter (OM) for non-urban aerosol (Turpin and Lim, 2001), such as at the Zeppelin Observatory, where most aerosol particles are long-range transported."

Russian is a very large territory. Do the authors have more detailed information on the main source areas for high sulfur episodes, e.g. the Kola Peninsula area discussed a lot in previous literature?

Indeed, the Kola Peninsula (particularly smelters in the area) is a known major source, also discussed in the original reference by Aas et al. This is included in the revised manuscript: "These episodes of sulfur pollution occur due to the arrival of air masses from Russia, for example the Kola Peninsula, due to the presence of heavy industry including non-ferrous metal smelters (Aas et al., 2020)."

When discussing about past trends of inorganic ion concentrations in atmospheric aerosols (lines 686-696), I wonder why the authors refer to targeted emission reductions during 1990-2010, not the real emission reduction that took place. Data on actual emission reductions during that period is certainly available.

The next statement in the manuscript is in fact a comparison to the real-world emissions. For clarity, we specify this in the revised manuscript as follows "This is significantly lower than the real-world

reductions seen on the Norwegian mainland, e.g. SO₂: 95%, sea salt corrected $SO_4^{2-}$: 74% at Birkenes, South Norway (Aas et al., 2020)."

It is important to note that these inorganic ion measurements are connected to the need to monitor these species in the GP, and so the progress in real-world compared to the targeted emissions is highly relevant.

This section is about aerosol physical and optical properties (the word optical could be included into the title), so why do the author start the section by mentioning CCN and cloud properties which are not discussed at all in this section?

We note that several of the papers covering aerosol physical properties in the Arctic are linked to the influence of such properties on CCN, but that this does not need to be explicitly stated here. For clarity we have changed the opening sentence to "There are large number of atmospheric aerosol and cloud studies in polar regions based on short term campaigns, carried out predominantly in the polar day season."

lines 702-704: The authors assume implicitly here that the particle size is some sort of proxy for its ageing. This is probably true but should be explained for readers not familiar with combined effects of aerosol sources and aerosol dynamics taking place during atmospheric transportation.

A clarification of what is observed is added, as well as an explanation of the processes driving growth in these particles, which is commonly called ageing. We state this explicitly in the revised manuscript as follows: "These particles, when observed in remote areas, are usually formed through atmospheric processes such as condensation, coagulation and cloud processing, and are commonly referred to as 'aged' particles (See Fig. 7). "

Please explain in more detail what is meant by "light period" and "summer" (lines 710 and 711), and whether "sunlit period" (line 724) means something else.

The text has been revised to clarify that nucleation is highest during the lightest periods of the summer, specified as follows: "Particle nucleation events followed by particle growth are observed frequently during the lightest part of the year, i.e. April to September, and dominate the particle concentration during the summer, i.e. June to August (see Fig. 8 and Tunved et al., 2013)"

The discussion about aerosol optical properties is vague (lines 732-745). The authors try to relate changes in optical properties to those in particle concentrations (number or mass, not explained?), but the relevance or purpose of this exercise has not been explained. I do not understand what the authors mean be stating the particle concentrations increase through the year (line 740). This whole paragraph needs to be rewritten.

It is a misunderstanding that we try to relate the optical properties to the size distributions given in figure 7 to 9. It is not written anywhere. In this paper we only give examples on how different types of observations have given a better understanding of the Arctic atmosphere and its composition. The optical observations are made over 18 years to compare with 10 years for the particle size measurements supporting a more rigid trend analysis. There is no such trend analysis presented for the particle size measurements yet. Thus, no comparison is made.  However, we have rewritten the section on the optical properties to make it as clear as possible. A more detailed explanation on what influences the optical properties has been added:  "Recently, 18 years of nephelometer measurements at the Zeppelin Observatory were used to evaluate the trends of particle light scattering properties

(see Fig. 10 and Heslin-Rees et al. 2020). An increase in particle light scattering indicates either an increase of particle concentrations or an increase in particle size; the latter is supported by a decreasing scattering Ångström exponent, showing a shift to larger particles in the particle size distribution. The increase in particle size and particle light scattering coefficient seen throughout the 18 years most likely corresponds to an increased contribution from larger particles such as sea spray. Heslin-Rees et al. (2020) argue the observed long-term changes are due to changes in atmospheric circulation, i.e., an increased frequency of long-range transport from the open northern Atlantic. However, new particle formation (NPF) events at the Zeppelin Observatory have been shown to be anti-correlated with sea ice extent indicating a dependence on more open sea Dall'Osto et al. (2017). This is also supported by number of recent studies linking ocean biological activity with biogenic sulfur variability and abundance in the Arctic atmosphere (Jang et al, 2021) and related aerosol properties and cloud condensational nuclei variability (Choi et al, 2019,  Park et al, 2021).  Naturally driven NPF dominates the summertime Arctic atmospheric aerosol, even though the detailed physiochemical process pathway is not known and is a subject of ongoing research."

The discussion on lines 755-764 is rather general and appears to be loosely connected with other contents of section 5.2.

This section belongs in the aerosol measurement methodology in 4.2 and has been moved accordingly

Section 5.3

What is the point of bringing up CO2 concentration in 2019 and its increase from the previous year? The CO2 increase is a well-known fact, while its annual increase rate varies from year to year. Data from one single year provide little insight on this matter (lines 766-768).

The comparison to the previous year has been removed and the sentence is now "$CO_2$ is increasing with a long-term trend of 2.5 ppm per year (Table 5) and has increased by ≈15% since 1989 levels (357 ppm)."

What is the basis for stating that the CO2 concentration increase rata is exponential? (line769)

This statement has been rephrased: "It should be noted that the growth is positive in all years, highlighting the challenge in meeting emissions reductions needed to meet the Paris Agreement goal of keeping the global annual average temperature increases below 2℃."

line 793: any explanation for the stated pause of CH4 mixing ratio?

The reason for this pause is discussed in detail in the section that immediately follows and we do not feel that any further detail is needed.

There is repetition of text between the lines 838-844 and lines 858-869. Also figure 12 appears twice in the paper.

The repetitions in text and the extra figure have been deleted in the revised manuscript at the second occurrence.

Section 5.6:

Based on measurements of just one site in Arctic and one site outside Arctic,it is impossible to make any general statement about differences between Article areas

and those outside Arctic (lines 992-994).

This general statement has been removed from the main text.

Section 6.2:

There is much new scientific work and findings on arctic amplification and related issues that seem to be missing in the introductory part of this section (lines

A thorough review of Arctic amplification is beyond the scope of this article. Meanwhile, the observation of rapid Arctic warming alone, rather than a discussion of its causes, is more relevant to the section. Hence, we have removed the discussion of Arctic amplification from the revised manuscript.

Technical and minor scientific issues

line 227: something is missing from here (e.g. … during 1971 to 1980)

'From' has been added. "This conclusion was based on extensive climatological tabulations of the meteorological observations in Ny-Ålesund from 1971 to 1980."

line 435: INP should be in parenthesis

This has been corrected in the revised manuscript.

line 440 (and later line 765): The term "climate gases" is not commonly in use. Please consider modifying the titles.

We have modified this to the more explicit "Atmospheric trace gases of high relevance to global climate change"

line 711: Figure 8 does not tell anything about nucleation and particle growth, so it should not be referred to here but later in the text.

Particle growth and nucleation directly impact particle mass, and these are the main causes of seasonal variability in Fig. 8. We do not feel a change is needed here.

line 795: is it possible to measure the CH4 concentration with a 5-digit accuracy?

It is indeed possible to measure with such precision: the G2401 manual states that the precision is at least +/- 0.3 ppb for 1 hour averaging (we actually average over longer for the daily means). See https://www.picarro.com/support/library/documents/g2401_analyzer_datasheet

line 801: an increase from 28 to 32 is not consistent with 25% increase.

This has been deleted in the revised manuscript.

lines 811 and 815: suggest à suggested

Since the change is recent and ongoing (at least until 2020) these statements seem grammatically correct with respect to the use of present and present continuous tenses.

line 1008: MC or MCM?

This has been changed to MCM at the first instance

lines 1013-1014: it is enough to explain GEM one time.

This has been corrected in the revised manuscript.

line 1037: … Asia, including China, contribute…

The comma has been added to the revised manuscript

line 1092 vs. line 1143: please use only a single term for LOE (episodes of very low ozone vs. low ozone episodes).

We have opted to use 'low ozone episodes' for consistency

lines 1119-1121: Unclear sentence, please modify.

We have split this sentence in the revised manuscript: "Furthermore, links between climate change in other regions and the frequency of Arctic LOEs have been proposed. For example. (Koo et al., 2014) found correlations between so-called 'teleconnection patterns', i.e. weather patterns in other regions, such as the Western Pacific, and the frequency of LOEs in the Arctic."

lines 1166-1168: the first sentence of section 6 is unclear. Please re-write.

We have changed this to "The main focus of atmospheric research has shifted over the decades."

line 1184: please correct the grammar (that need study)

This has been changed to "requiring study".

line 1125: a paper in preparation is not a proper reference.

This reference and accompanying statement have been removed for the revised manuscript. Instead, we provide a broader statement with a finished paper reference: "A shift in the natural aerosol baseline within the Arctic is evident and an improved knowledge of the individual processes is needed to better constrain the future development of the Arctic climate (Schmale et al. 2021)."

line 1316: please correct the grammar (will need study)

This has been changed to "require study".

lines 1332 and 1334: CEAC or CEC?

CEAC has been changed to "CECs in the Arctic" for the revised manuscript.

lines 1363-1364: unclear sentence, please modify

This has been corrected to "While the observatory at its inception was primarily focused on national monitoring, the Zeppelin Observatory now hosts measurements from 17 institutions in 13 countries.

Although these are mostly long-term measurements, the Zeppelin Observatory regularly hosts instruments for short-term (1 to 3 years) campaigns. Measurement capabilities have been continuously improved to include state-of-the-art instrumentation."

Some of the figures (figs. 13, 14 and 20) are of poor technical quality.

We have improved the figure quality for the revised manuscript.

Figures 12 and 18 appear twice in the text.

Duplicate figures have been removed from the revised manuscript.

---

## Author Comment (AC2)

Response to anonymous reviewer #2

We thank the reviewer for providing helpful comments on our manuscript. We address all issues raised (blue text) as follows:

*The paper is well structured and written. However, it is quite long and the authors could consider shorten the text and add some summarizing figures or statistics.*

This article is long because of the breadth of material we cover. We have structured the article in a consistent, accessible way, so that individual subject areas e.g., historical aspects, or mercury etc. can be found by specialist readers. Meanwhile, we have kept each individual section concise. We do not believe that adding additional figures, metadata, and tables will shorten the manuscript. Shortening the article would require removing sections, which would then mean the paper was no longer comprehensive. We also note that reviewer #1 suggests the length is justified for this type of article.

We do agree that some statistics on the impact of the site would be useful and add these to the revised manuscript in the introduction, e.g. "The Zeppelin Observatory is now a leading global background measurement site (Tørseth et al., 2012). Google scholar finds at least 280 publications including the search term "Zeppelin Observatory" and, for example, greenhouse gas data/ meta data ($CO_2$, CO, $CH_4$) have been downloaded at least 4439 times from the ICOS carbon portal (https://www.icos-cp.eu/observations/carbon-portal)."

*line 53: how the decision of the Swedish parliament accepted for a Swedish monitoring program in connected to the Norwegian approach. Please clarify.*

We reformulate this sentence to "Meanwhile in Sweden in the same year, the Swedish parliament accepted the proposal for a Swedish monitoring programme ('program för övervakning av miljökvalitet', PMK) one part of which was to be long-term monitoring at Ny-Ålesund by Stockholm University of changes in atmospheric composition with an emphasis on aerosols and carbon dioxide in collaboration with NILU in Norway."

*line 842: History; you could add the time line figure demonstrating the different atmospheric composition measurements at different locations. This would also provide a general overview on the development and availability of long term measurements and address the role of Zeppelin measurements.*

We have added a timeline figure to the revised manuscript.

*line 113: add "full stop" after parenthesis.*

This has been corrected in the revised manuscript.

*lines 195-199: please add references for the location climate / vegetation classification.*

A reference to (Bliss, 2000) has been added for 'Arctic Tundra Zone'. 'Northern' does not need a reference as the high latitude has been stated previously in the manuscript but should not be capitalized, i.e. "northern Arctic Tundra Zone". This has now been corrected.

*line 224: please add a reference if possible.*

A reference to (Hov and Holtet, 1987) has been added.

*line 255: you could use "/sios-svalbard.org/"*

This has been corrected in the revised manuscript.

line 285: open the acronym FLEXPART particle dispersion model:>>> "FLEXible PARTicle dispersion model". Please re-check all the acronyms in the text and open the acronym when mentioned for the first time

This, and other undefined acronyms in the text have been defined in the revised manuscript.

Fig 3: improve quality (resolution) of the figure

The figure quality has been improved in the revised manuscript.

line 316: Aitken mode particles, please add size

This has been added to the revised manuscript: "At the same time there is usually a shift in the aerosol size distribution from dominant accumulation mode towards smaller Aitken mode particles (particle diameter typically <60 nm, e.g. Tunved et al., 2013) indicating a very different origin of the chemical load observed. "

Fig 4 please improve quality (resolution) of the figure

The figure quality has been improved in the revised manuscript.

line 340: you could add a short overall (meta) description / table of all measurements which would better describe the overall measurement capacity of the station. And give some general statistics of the measurements.

Tables 1-4 already list the instruments and capability of the Zeppelin Observatory. General numbers are given in the main text where relevant to the discussion, while we feel that adding more complexity to the tables without context (which would otherwise increase the length of the paper) would not provide the reader with a better overview of the history of the station or long-term developments in the Arctic atmosphere.

line 350-365: (4.1) this is very detailed description of the samplings and filters, you could consider a schematic figure of the process or an annex.

A small part of the discussion on filter sampling has been removed to keep this method description more consistent in length with the other methods sections.

line 366: add the reference for Mann-Kendall Test/Sen's slope.

The references (Mann, 1945;Kendall, 1948;Sen, 1968) have been added to the revised manuscript at appropriate locations.

Fig 6., 7., 8. technical quality of the fig should be improved

The figure resolutions have been improved in the revised manuscript.

line 439: "With this set-up the Zeppelin Observatory is now one of the first global aerosol observatories with semi-continuous in-situ cloud sampling". What are the other stations ?

We have modified this sentence as follows: "With the current set up, including cloud condensation nuclei counters (CCNC), cloud residual properties and cloud and precipitation microphysical properties the Zeppelin Observatory is now (to the best of our knowledge) the first global aerosol observatory with continuous in-situ observations of atmospheric aerosol, cloud residuals, clouds and precipitation (e.g., Koike et al., 2019)."

line 839, 865: some error with the reference (Error! Not a valid bookmark self-reference)

This has been corrected in the revised manuscript.

line(s) 925 & 962 please check, add the reference Petäjä et al.2020 Overview: Integrative and Comprehensive Understanding on Polar Environments (iCUPE) – concept and initial results, Atmos. Chem. Phys., 20, 8551–8592, https://doi.org/10.5194/acp-20-8551-2020, 2020.

This reference has been added to the revised manuscript.

line 1074: please add some specification for the acronyms "POLARCAT" "TOPSE"

The acronyms Polar Study using Aircraft, Remote Sensing, Surface Measurements and Models, of Climate, Chemistry, Aerosols, and Transport (POLARCAT) and Tropospheric Ozone Production about the Spring Equinox (TOPSE) have been added to the revised manuscript

line 1172: "strengthen the position of the Zeppelin Observatory as a leading global measurement platform, perhaps" one of the ? / Arctic ?

Since the Zeppelin observatory is defined as a global background site, e.g. (Tørseth et al., 2012), we have changed this to "strengthening the position of the Zeppelin Observatory as a leading global background measurement site"

line 1200: Refer to "open access", how is the data access to Zeppelin measurements currently organized ?

Data access is already described in Section 9. "Most data are publicly available on ebas.nilu.no or else on request via the responsible institutions listed in Tables 1-4."

line 1227: "Changes in the Arctic aerosol burden will in turn influence climate via direct and indirect aerosol effects, i.e. via increased absorption and scattering, and changes in CCN and ice nucleating particles (INP), respectively. " - add reference

We have added a reference to (Schmale et al., 2021) in the revised manuscript.

line 1228: "Another important non-$CO_2$ greenhouse gas is $N_2O$, with a global warming potential 265–298 times that of $CO_2$." - add reference

References to Montzka et al. 2011 and Hodnebrog et al., 2013 have been added to the revised mansucript.

**References**

Bliss, L. C.: Arctic tundra and polar desert biome, North American terrestrial vegetation, 2, 1-40, 2000.
Hov, Ø., and Holtet, J., A.: Prosjektering av atmosfærekjemisk forskningsstasjon I Ny-Ålesund på Svalbard. NILU OR 67/87. Lillestrøm, Norway., 1987.
Kendall, M. G.: Rank correlation methods, 1948.
Koike, M., Ukita, J., Ström, J., Tunved, P., Shiobara, M., Vitale, V., Lupi, A., Baumgardner, D., Ritter, C., and Hermansen, O.: Year-round in situ measurements of Arctic low-level clouds: Microphysical properties and their relationships with aerosols, Journal of Geophysical Research: Atmospheres, 124, 1798-1822, 2019.
Mann, H. B.: Nonparametric tests against trend, Econometrica: Journal of the econometric society, 245-259, 1945.
Schmale, J., Zieger, P., and Ekman, A. M.: Aerosols in current and future Arctic climate, Nature Climate Change, 11, 95-105, 2021.
Sen, P. K.: Estimates of the regression coefficient based on Kendall's tau, Journal of the American statistical association, 63, 1379-1389, 1968.

Tørseth, K., Aas, W., Breivik, K., Fjæraa, A. M., Fiebig, M., Hjellbrekke, A.-G., Lund Myhre, C., Solberg, S., and Yttri, K. E.: Introduction to the European Monitoring and Evaluation Programme (EMEP) and observed atmospheric composition change during 1972–2009, Atmospheric Chemistry and Physics, 12, 5447-5481, 2012.